# Robust single-cell matching and multimodal analysis using shared and distinct features

**Bokai Zhu**[1,2,9], **Shuxiao Chen**[3,9], **Yunhao Bai**[2,4], **Han Chen**[2], **Guanrui Liao**[5], **Nilanjan Mukherjee**[2], **Gustavo Vazquez**[2], **David R. McIlwain**[2], **Alexandar Tzankov**[6], **Ivan T. Lee**[2], **Matthias S. Matter**[6], **Yury Goltsev**[2], **Zongming Ma**[3,10] ✉, **Garry P. Nolan**[2,10] ✉ & **Sizun Jiang** ◉ [5,7,8,10] ✉

The ability to align individual cellular information from multiple experimental sources is fundamental for a systems-level understanding of biological processes. However, currently available tools are mainly designed for single-cell transcriptomics matching and integration, and generally rely on a large number of shared features across datasets for cell matching. This approach underperforms when applied to single-cell proteomic datasets due to the limited number of parameters simultaneously accessed and lack of shared markers across these experiments. Here, we introduce a cell-matching algorithm, matching with partial overlap (MARIO) that accounts for both shared and distinct features, while consisting of vital filtering steps to avoid suboptimal matching. MARIO accurately matches and integrates data from different single-cell proteomic and multimodal methods, including spatial techniques and has cross-species capabilities. MARIO robustly matched tissue macrophages identified from COVID-19 lung autopsies via codetection by indexing imaging to macrophages recovered from COVID-19 bronchoalveolar lavage fluid by cellular indexing of transcriptomes and epitopes by sequencing, revealing unique immune responses within the lung microenvironment of patients with COVID.

The rapid developments in single-cell technologies have fundamentally transformed the investigation of complex biological systems. The ability to individually measure the genomic[1], epigenomic[2], transcriptomic[3], and proteomic[4] states at the single-cell level marks an exciting era in biology. Single-cell transcriptomics and targeted proteomics are the two main approaches commonly used to delineate cell populations and infer functionality or disease states. Single-cell transcriptomics is theoretically able to assess the entire transcriptome of a target cell, with 5,000–10,000 unique gene transcripts captured on average for each cell. A key drawback of this method is the relative sparseness of the data generated, particularly for less abundant genes. On the other hand, antibody-based single-cell proteomics has gradually progressed over the years, from the initial detection of a handful of protein targets[5,6], to about 40 targets via mass cytometry[7], over 100 protein targets via sequencing[8,9] and, most recently, more than 40 protein targets spatially resolved in their native tissue context[10–13]. Emerging sequencing-based approaches such as cellular indexing of transcriptomes and epitopes by sequencing (CITE-seq) and RNA expression and protein sequencing

[1]Department of Microbiology and Immunology, Stanford University, Stanford, CA, USA. [2]Department of Pathology, Stanford University, Stanford, CA, USA. [3]Department of Statistics and Data Science, The Wharton School, University of Pennsylvania, PA, USA. [4]Department of Chemistry, Stanford University, Stanford, CA, USA. [5]Center for Virology and Vaccine Research, Beth Israel Deaconess Medical Center, Boston, MA, USA. [6]Pathology, Institute of Medical Genetics and Pathology, University Hospital Basel, University of Basel, Basel, Switzerland. [7]Department of Pathology, Dana Farber Cancer Institute, Boston, MA, USA. [8]Broad Institute of Harvard and MIT, Cambridge, MA, USA. [9]These authors contributed equally: Bokai Zhu, Shuxiao Chen. [10]These authors jointly supervised this work: Zongming Ma, Garry P. Nolan, Sizun Jiang. ✉e-mail: zongming@wharton.upenn.edu; gnolan@stanford.edu; sjiang3@bidmc.harvard.edu

assay can simultaneously probe the RNA and protein levels for each single cell, albeit with the tradeoff of dissociating cells from their original spatial location.

Given the frequent overlap in proteins measured across dissociated single cells via sequencing, and intact tissues via antibody-imaging, an orthogonal approach would leverage information from one modality to inform the other. Such an effort would use biological measurements obtained on one modality (for example, CITE-seq) to inform cells measured using another modality (for example, codetection by indexing or CODEX) for a comprehensive assessment of the localization of both proteins and RNAs within tissue samples, hence it is vital to have the ability to align individual cells across these experiments.

Several computational approaches for integrative analysis of single-cell data across multiple modalities currently exist[14–18]. However, most of these methods are tailored toward single-cell sequencing-based analysis, such as single-cell RNA-sequencing (scRNA-seq) and single-cell assay for transposase-accessible chromatin sequencing, and are not designed for protein-based assays as the limited shared features across proteomic datasets are orders of magnitude smaller than those in single-cell sequencing datasets, and the signals within these limited shared features alone are typically insufficient to produce high-quality and interpretable pairwise cell-matching results. In addition, the intrinsically greedy (and thus at most locally optimal) nature of the mutual nearest neighborhood (mNN) matching algorithm routinely used in available methods limits the ability to fully use the correlation structure within the distinct protein features. Thus, there is an unmet need for a new strategy specifically designed for matching and integrating single-cell datasets based on limited but robust proteomic parameters.

To meet this need, we have developed matching with partial overlap (MARIO): the matching process leverages both shared and distinct features between datasets, and is nongreedy by optimizing a global objective. We additionally developed two quality control steps, the matchability test and joint regularized filtering, to avoid suboptimal matching and prevent uninterpretable over-integration. Benchmarking of MARIO across various single-cell proteomic data generated from different modalities (cytometry by time of flight (CyTOF), CITE-seq, and CODEX) and from different species (human and nonhuman primates (NHPs)) demonstrated consistent outperformance of cell–cell-matching accuracy over available methods. Finally, we matched macrophages from a CODEX multiplex immunofluorescence lung autopsy dataset to CITE-seq bronchoalveolar lavage fluid (BALF) macrophage cells using MARIO to uncover a spatially orchestrated immune conditioning by complement-expressing macrophages and neutrophils in COVID-19. To make MARIO freely available to the public, we implemented the algorithm in the Python package MARIO, along with an R version available online at https://github.com/shuxiaoc/mario-py.

## Results

### Matching single cells using partially shared features

There are unique challenges in the implementation of a cell-matching algorithm using proteomic information. First, each study is often bespoke and rarely shares identical antibody panels, although a portion of the proteins measured is generally the same. Thus, the matching process must be able to achieve stable pairing of cells with the limited number of features; this is in contrast to transcriptomics data where often several hundreds to thousands of shared features are available[16,17]. Second, underlying correlations between shared and distinct features often exist within and between datasets as a result of panel design and fundamental biological principles. It is therefore pertinent to incorporate information from both shared and distinct protein features. Third, the matching problem corresponds to a well-defined objective function. The mNN-type algorithms can be thought of solving this objective function in a greedy fashion, but often a global optimum is unattainable (see the Methods for mathematical details). As such,

the matching problem should be solved to attain the global optimum rather than a local optimum. Finally, key quality control steps are crucial to ensure the accuracy and interpretability of the postulated cell–cell-matching results.

To address these challenges, we developed MARIO: a robust framework that accurately matches cells across single-cell proteomic datasets for downstream analysis (Fig. 1a,b). MARIO first performs a pairwise cell matching using shared features. To do this, after proper transformation, normalization and batching, we use singular value decomposition on shared features to construct a cross-data distance matrix based on the Pearson correlation of the reduced matrix. An initial cell–cell pairing is then obtained by solving a linear assignment problem that searches for a distance-minimizing injective map between the two collections of cells. The two datasets are next aligned using this initial matching, and both shared and distinct features of the two datasets are projected onto a common subspace using canonical correlation analysis (CCA)[19], as it incorporates the hidden correlations between different proteomic features not shared between the datasets. A cross-dataset distance is then obtained using the canonical scores, and a refined matching is obtained via linear assignment on the new distance. By taking the means of the top sample canonical correlations as a proxy of matching quality, MARIO then finds the best convex combination weight to interpolate the initial and refined matchings. This allows users to data-adaptively backtrack toward the initial matching when the refined matching becomes unreliable (Fig. 1c).

After obtaining the interpolated matching, MARIO next performs a matchability test to determine whether or not the datasets selected for integration by the user are suitable for such a joint analysis. The matchability test is performed by flipping the sign of each row of the two datasets with some flipping probability, so that most of underlying inter-dataset correlations (if these exist) are abrogated. This process is repeated several times to build a distribution of the background canonical correlations of the samples with a low underlying correlation. Comparison of the deviation of the sample canonical correlations from the background distribution reveals whether strong underlying information exists to connect the datasets (Fig. 1d).

Although datasets passing the matchability test are highly correlated, the matching at the individual cell level could still be erroneous. To address this problem, we developed a process termed jointly regularized filtering to automatically filter out low-quality matches without a priori biological knowledge. The filtering process is carried out by optimizing a regularized $k$-means objective. This objective is a superposition of two parts, where the first part contains individual $k$-means clustering objectives for both datasets and the second part penalizes the Hamming distance between the two individual cluster label vectors and a hypothesized 'global' label vector. Use of such a strategy stems from our hypothesis that although the populations being measured in two different experiments may contain modality-specific characteristics (thus the existence of 'individual' cluster labels), both originate from a biologically analogous population (thus the existence of a global cluster label that is close to the two individual cluster labels). If, for a matched pair of cells, the individual labels obtained by joint regularized clustering are not the same, this matched pair is likely spurious and thus disregarded (Fig. 1d). After this filtering step, the resulting individually matched cells are subjected to CCA, and the canonical scores are used as the reduced components in the final embeddings. We implemented generalized CCA to achieve joint embedding of more than two datasets, and subsequently used the gCCA canonical scores as dimensionally reduced components in the final embeddings (Fig. 1e). Mathematical details can be found in the Methods.

To verify the merit of MARIO in a 'ground-truth' setting, we tested the matching performance on simulated data with high-granularity cell types. We used Symsim[20] to simulate single-cell epitome-like datasets[21]: data with 20 cell populations from two different modalities, with a total of 60 features generated. To mimic scenarios of different antibody

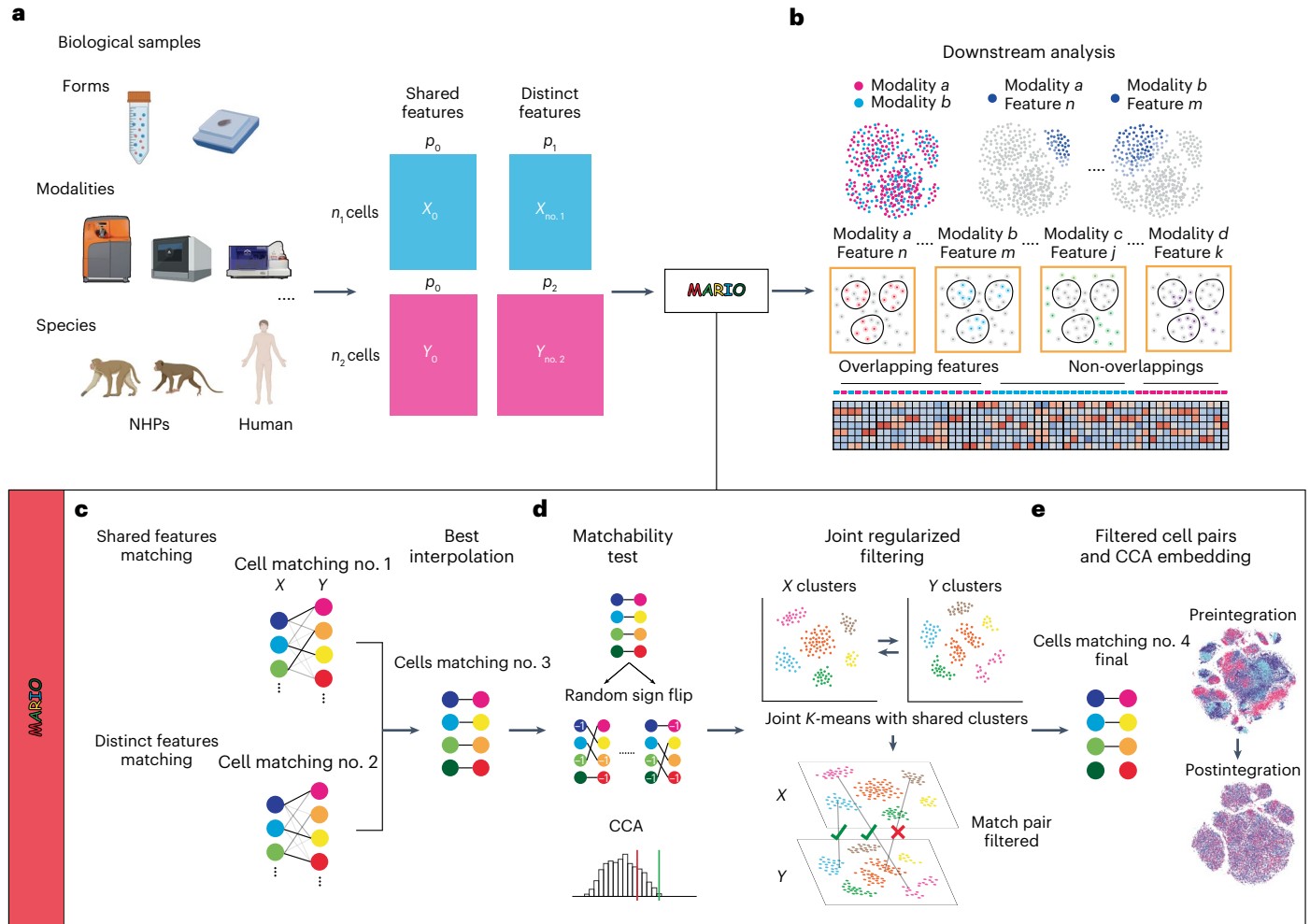

**Fig. 1 | Schematic of the MARIO analysis pipeline. a**, Single-cell proteomic datasets can be acquired using various modalities, including CyTOF, CITE-seq and CODEX, on different biological samples or species (for example, human or NHP) with shared underlying biological information. Protein markers are divided into two classes: (1) features captured within both datasets (shared features), and (2) markers not shared between the datasets (distinct features). Both classes of protein expression matrices serve as inputs to the MARIO algorithm. **b**, After the MARIO pipeline, further downstream analysis can be conducted using the combined information integrated across multiple individual experiments.

**c**, In the first step of MARIO, individual cells are first subjected to matching using the distance matrix constructed using the shared features described in **a**, before further match refinement using the distance matrix constructed from the distinct features such that all features are included. Thereafter, the best interpolation of initial and refined matching will be performed. **d**, In the second step of MARIO, the dataset then undergoes quality control steps (matchability test and joint regularized filtering). **e**, In the MARIO third step, with the matching information, cells across both datasets are jointly embedded into a CCA subspace.

panel setup, 20 out of 60 features were shared across two datasets, with a further 20 distinct features each. Indeed, MARIO showed improved matching capability in the simulated ground-truth case with limited protein features available and shared across datasets (MARIO 81.54%, second best Scanorama 63.3%) (Extended Data Fig. 1).

### Matching and integration of multimodal single-cell protein datasets

We evaluated the performance of MARIO on two distinctive datasets generated using individual cells isolated from healthy human bone marrow. The first is a sequencing-based CITE-seq dataset consisting of 29,007 cells stained with an antibody panel of 29 markers[17], and the second is a mass cytometry-based CyTOF dataset consisting of 102,977 cells stained with an antibody panel of 32 markers[22]. Twelve markers were common to both datasets. MARIO successfully matched and aligned these two datasets as shown by visual inspection (Fig. 2a). The intricate data structures were preserved post-MARIO integration, with clear separation of cells belonging to phenotypically distinctive populations in dimensionally reduced t-distributed stochastic neighbor

embedding (t-SNE) plots (Fig. 2b). The original cell-type annotations based on the shared low-level annotation (Fig. 2b, top left), and on preexisting annotations from each dataset (Fig. 2b, top right and bottom left) were highly conserved after MARIO integration. Subsequent joint clustering of the post-MARIO integrated data using the canonical correlation scores also corroborated highly accurate cell-type delineation (Fig. 2b, bottom right).

We next designed three different scenarios to further characterize the integration performance of MARIO and to compare its performance against the single-cell integration methods Seurat[17], fastMNN[14] and Scanorama[15]. In the first case, shared protein markers were removed from each dataset individually (in an accumulative fashion and in alphabetical order) to simulate the distinctive antibody panel designs across datasets. MARIO consistently outperformed other methods in terms of matching accuracy, independently of the excluded protein targets (full 12-shared panel total accuracy MARIO, 96.01%; second beset Scanorama, 91.46%; dropping eight shared antibodies MARIO, 91.45%; second best Scanorama, 71.22%) (Fig. 2c and Extended Data Fig. 2a).

We additionally evaluated the integration quality among these methods, using metrics, including Structure alignment score, Silhouette F1 score, adjusted Rand index (ARI) F1, Cluster mixing score, and lower dimensional embedding, based on each method's post-integration latent space scores (Extended Data Fig. 2a,b and Supplementary Figs. 1 and 2). In addition, we removed shared protein markers as previously tested, but in the order of importance score (Methods), where less important markers were dropped first. This process mimics the natural logic of building antibody panels, and in such a dropping scheme, MARIO still consistently outperformed other methods in terms of matching accuracy (Supplementary Figs. 3 and 4).

In the second test, random noise was gradually spiked into the datasets to simulate the variability of intrinsic signal-noise in real world data. The matchability test implemented in MARIO was able to detect and alert the user when data quality was insufficient for confident matching (Fig. 2d). In contrast, the elevated noise resulted in an increase in the number of cells being forcefully paired in other tested methods (reaching close to 100%), albeit with low accuracy (ranging from 50 to 80% in accuracy).

In the third scenario, an entire group of cell types was removed from the destination dataset (that is, the set being matched to) to mimic fluctuations of cell-type composition between datasets. MARIO outperformed all other tested methods by successfully suppressing the incorrect matching of these missing cell types (Fig. 2e).

Given that the matching accuracy for CyTOF to CITE-seq cell pairs among all the main cell types with MARIO was consistently high (Supplementary Fig. 5a); this allowed confident inference of the transcriptome within the single cells measured using CyTOF from their CITE-seq counterparts. We confirmed that the expression patterns of cell type-specific markers were in good agreement between CyTOF proteins, CITE-seq proteins and CITE-seq RNA transcripts (Fig. 2f,g and Supplementary Fig. 5b,c). Moreover, the expression patterns of CD45RO protein and *S100A4* and *CCR7* RNAs from CITE-seq assisted the delineation of memory and naive CD4 T cell subtypes in the integrated dataset, which was individually unavailable for manual annotation in the CyTOF dataset alone. Therefore, this integrated analysis better defines cell states than do these modalities individually.

We subsequently evaluated the performance of MARIO on two healthy human peripheral blood mononuclear cell (PBMC) datasets measured using CITE-seq and CyTOF. Fifteen proteins were common across these two datasets. MARIO successfully integrated the two datasets (Extended Data Fig. 3a) with high accuracy (Extended Data Fig. 3b). Our results reveal that the expression of key genes on both protein (CyTOF and CITE-seq) and RNA (CITE-seq) levels are in high agreement with their corresponding phenotypic cell-of-origin assignments (Extended Data Fig. 3c). Further benchmarking using the three cases described above showed similar superior matching accuracy for MARIO regardless of antibody panel setup (Extended Data Fig. 4a; for full 15-antibody shared panel total accuracy, MARIO at 90.62% and second best, Seurat 87.55%; for dropping eight shared antibodies, for total accuracy, MARIO 86.34% and second best, Scanorama 81.03%). In evaluation of suppression of over-integration due to poor quality data, mNN methods force matched almost all cells with accuracy below 70%, whereas MARIO alerted the user of poor data quality (Extended Data Fig. 4b). Third, integration with MARIO, but not with mNN methods, was robust even with extensive cell-type composition changes (Extended Data Fig. 4c,d and Supplementary Fig. 6).

## Cross-species analysis reveals species and stimuli-specific responses

We performed MARIO matching of four CyTOF datasets from studies in which (1) human whole blood cells were isolated from individuals challenged with H1N1 virus[23], (2) human whole blood cells were stimulated with IFNγ[24], (3) rhesus macaque whole blood cells were stimulated with IFNγ and (4) cynomolgus monkey whole blood cells were stimulated with IFNγ (Fig. 3a). Dataset 1 was generated using 42 markers, and datasets 2–4 were generated using 39 markers. We observed a high degree of concordance between cell types when visualizing the human–human and human-NHP datasets via *t*-SNE using MARIO integrated canonical scores (Fig. 3a). In contrast, datasets without MARIO integration process showed an unhomogenized pattern in the *t*-SNE visualization, indicating the necessity of performing MARIO integration for robust cross-comparisons across these four datasets (Fig. 3b). MARIO cell-type assignment was accurate among different cell types (Supplementary Fig. 7a). There were minimal differences, as measured using Euclidean distance, between paired cells calculated by canonical scores (Supplementary Fig. 7b).

Successful application of MARIO for robust matching and integration across three species and two stimulation conditions granted the opportunity to visually observe subtle changes in expression patterns across different cell types and datasets (Fig. 3c and Supplementary Fig. 7c). We observed an increase in proliferation of CD4 T cells in human blood cells after both influenza viral challenge and IFNγ stimulation, as marked by the upregulation of Ki-67, but no increase in proliferation was detected after stimulation of NHP blood cells. We also observed the upregulation of pSTAT3 in the natural killer cell population within human and NHP samples treated with IFNγ compared to human participants challenged with influenza, although overall pSTAT3 expression was higher in the influenza group. These results are consistent with previous observations[25–27]. Finally, there was an increased p38 expression in all cell types across all samples, reflective of the conserved functionality of p38 during cell inflammatory and stress responses[28,29]. In contrast, using the *t*-SNE plots from preintegration data proved hard to visually identify such an effect (Supplementary Fig. 7d).

Our benchmarking results showed superior matching accuracy using MARIO regardless of antibody panel setup. When using 39 shared antibodies, the total accuracy was 93.26% for MARIO and 86.20% for the second best method (Seurat); when eight shared antibodies were dropped, the total accuracy for IFNγ treatment was 86.79% for MARIO and 82.23% for the second best method (Scanorama) (Extended Data Fig. 5). The mNN methods forced matching of almost 100% of the cells with an accuracy less than 70% with increased spike-in noise, whereas MARIO alerted the user of insufficient information for matching (Supplementary Fig. 8a). MARIO, unlike the mNN methods we tested, was robust in resisting cell-type composition changes (Supplementary Figs. 8b, 9 and 10). Additionally, we removed shared protein markers as previously tested in the order of their importance score, with MARIO consistently outperforming other methods in matching accuracy (Supplementary Figs. 11 and 12).

---

**Fig. 2 | Matching and Integration of CyTOF and CITE-seq bone marrow data.**
**a**, *t*-SNE plots of individual cells colored by assay modality, either preintegration or MARIO integration. IFNG, interferon gamma. **b**, *t*-SNE plots of MARIO integrated cells colored by clustering results from (top left to bottom right): high concordance in shared cell types based on annotations from both original datasets; annotation from Levine et al.; annotation from Stuart et al. and the clustering result based on CCA scores from MARIO high cell-type resolution using information from both assays. **c–e**, Benchmarking results of MARIO against other mNN-based methods. **c**, The matching accuracy (left) and the proportion of cells being matched (right) are tested by sequentially dropping protein features. **d**, The matching accuracy (left) and the proportions of cells being matched (right) are measured with increasingly spiked-in noise. **e**, The error avoidance score (higher is better) is calculated after dropping each cell type sequentially from the dataset. **f**, Heatmap of cross-modality protein expression levels for the matched cells. **g**, *t*-SNE plots of the matched cells with protein or RNA expression levels overlaid based on each of the assays.

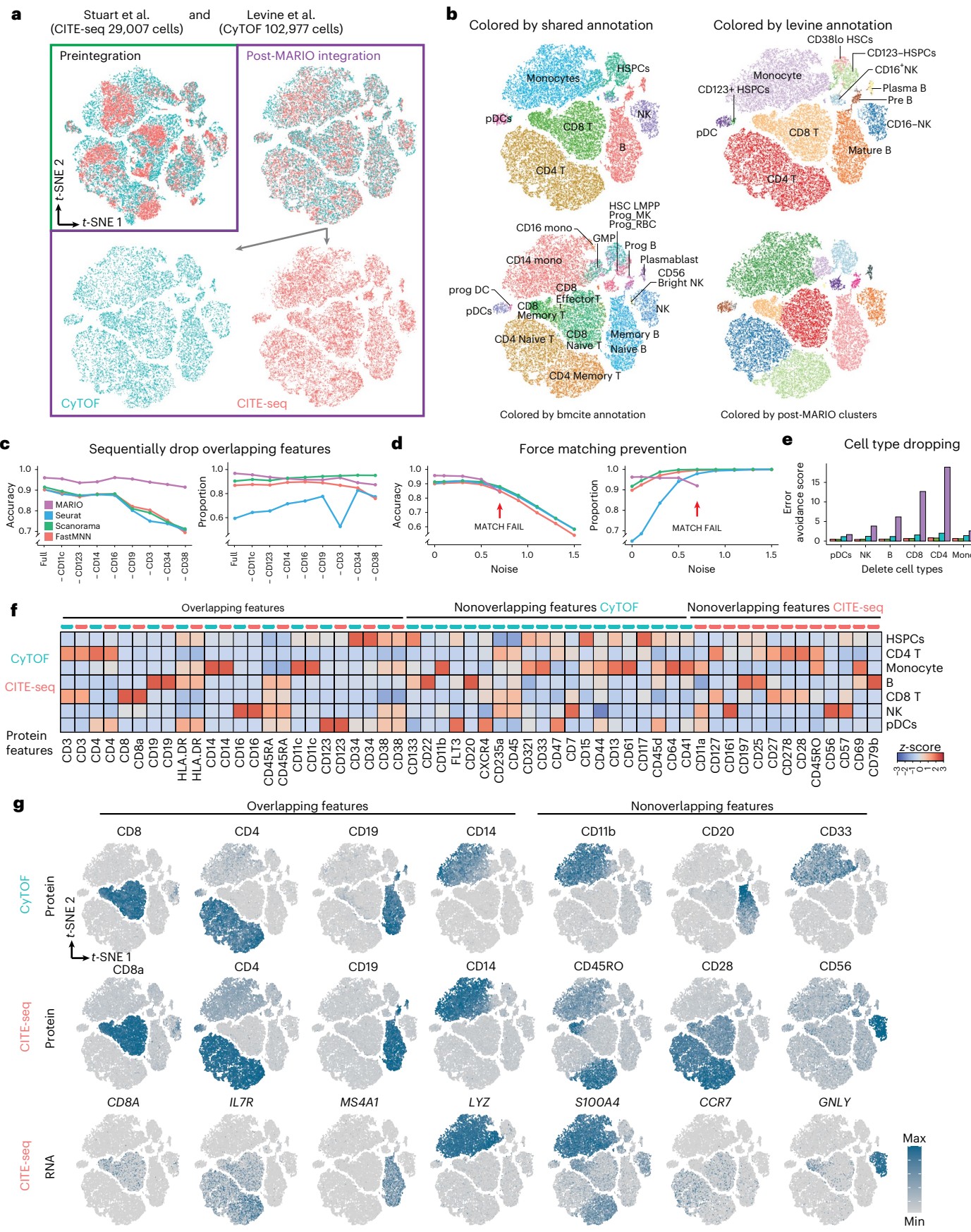

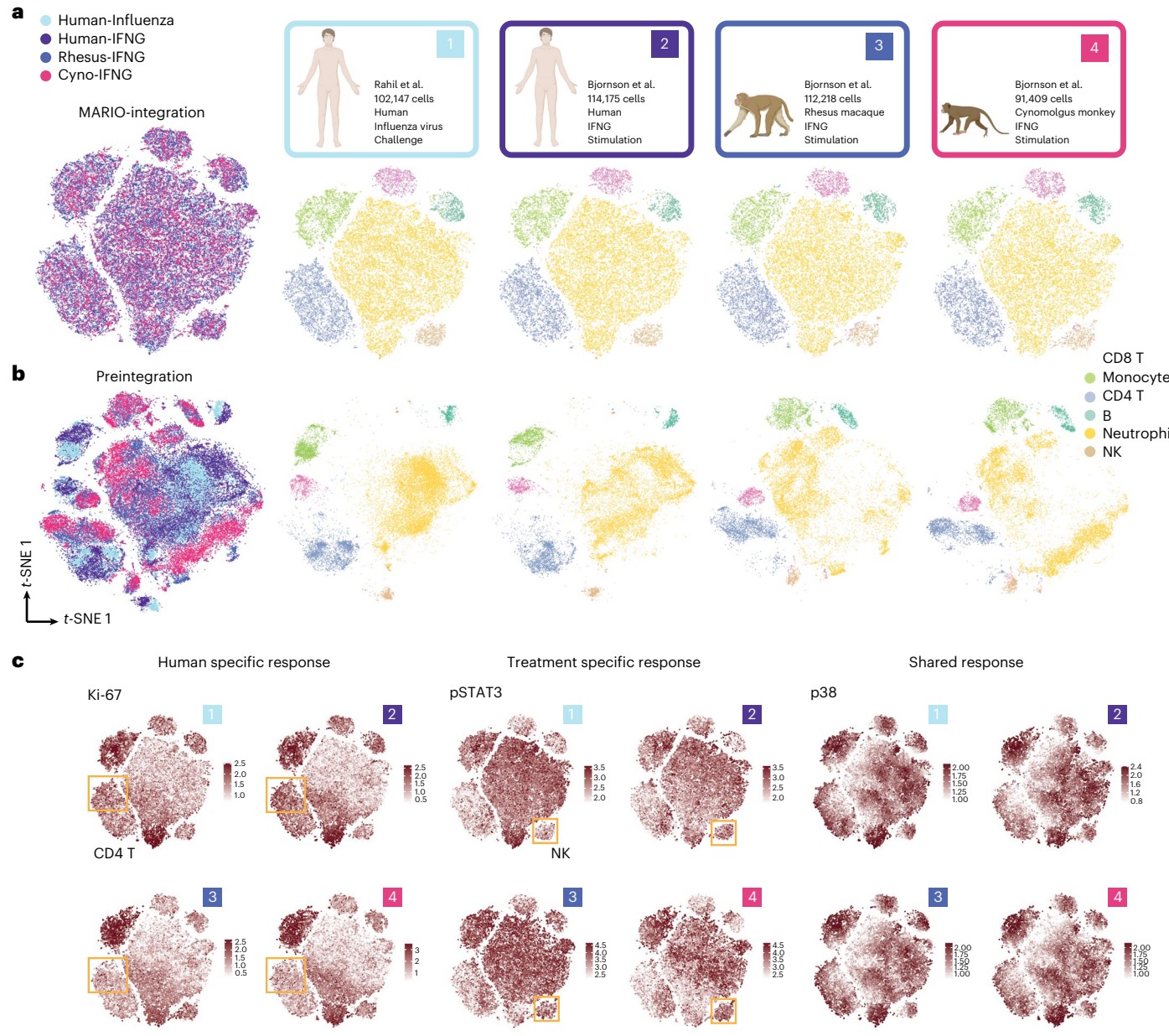

**Fig. 3 | MARIO enables cross-species and stimuli integrative analysis.** **a**, *t*-SNE plots of post-MARIO integration data, colored by their origin (left) or by each individual dataset, colored by cell type (right). **b**, *t*-SNE plots of pre-MARIO integration data, colored by their origin (left) or by each individual dataset, colored by cell type (right). **c**, *t*-SNE plots of post-MARIO integration data, with expression levels of Ki-67, pSTAT1 and p38 across the four datasets.

We similarly applied this strategy to data from IL-4-stimulated human and NHP whole blood cells, and compared them to human influenza viral challenge blood cells (Supplementary Fig. 13a,b). On IL-4 stimulation, we saw an upregulation of Ki-67 in human CD4 T cells but not NHP cells, much akin to IFNγ stimulation, and high expression of pSTAT3 in the natural killer of IL-4-stimulated blood cells, but not in PBMCs from humans challenged with influenza (Supplementary Fig. 13c). In line with IFNγ stimulation, the p38 response was consistent across species and treatments. Our results consistently showed that, regardless of antibody panel setup, MARIO had superior matching accuracy (Supplementary Fig. 14), prevented over-integration (Supplementary Fig. 15a), was robust to cell-type composition changes (Supplementary Fig. 15b) and generated accurate lower dimensional embedding (Supplementary Figs. 15c and 16).

**Accurate tissue architectural reconstruction via matching**

Matching cells from sequencing modalities on to multiplex proteomics imaging data has been previously attempted using existing integration algorithms (for example, STvEA using Seurat v.2)[30]. We reasoned that a highly accurate cell matching and integration from MARIO could infer the spatial localization of transcripts within individual cells. We performed MARIO on spatially resolved data from murine splenic cells collected using antibody-based CODEX imaging (28 protein markers)[13] and data from dissociated murine splenic cells assayed using CITE-seq (206 protein markers)[31]; 28 protein markers (all the markers in the CODEX dataset) were shared.

We first visually verified successful MARIO matching and integration using dimensionally reduced *t*-SNE plots (Fig. 4a). Cell–cell matching was accurate across different cell types (Extended Data Fig. 6a). This enabled accurate single-cell information transfer between

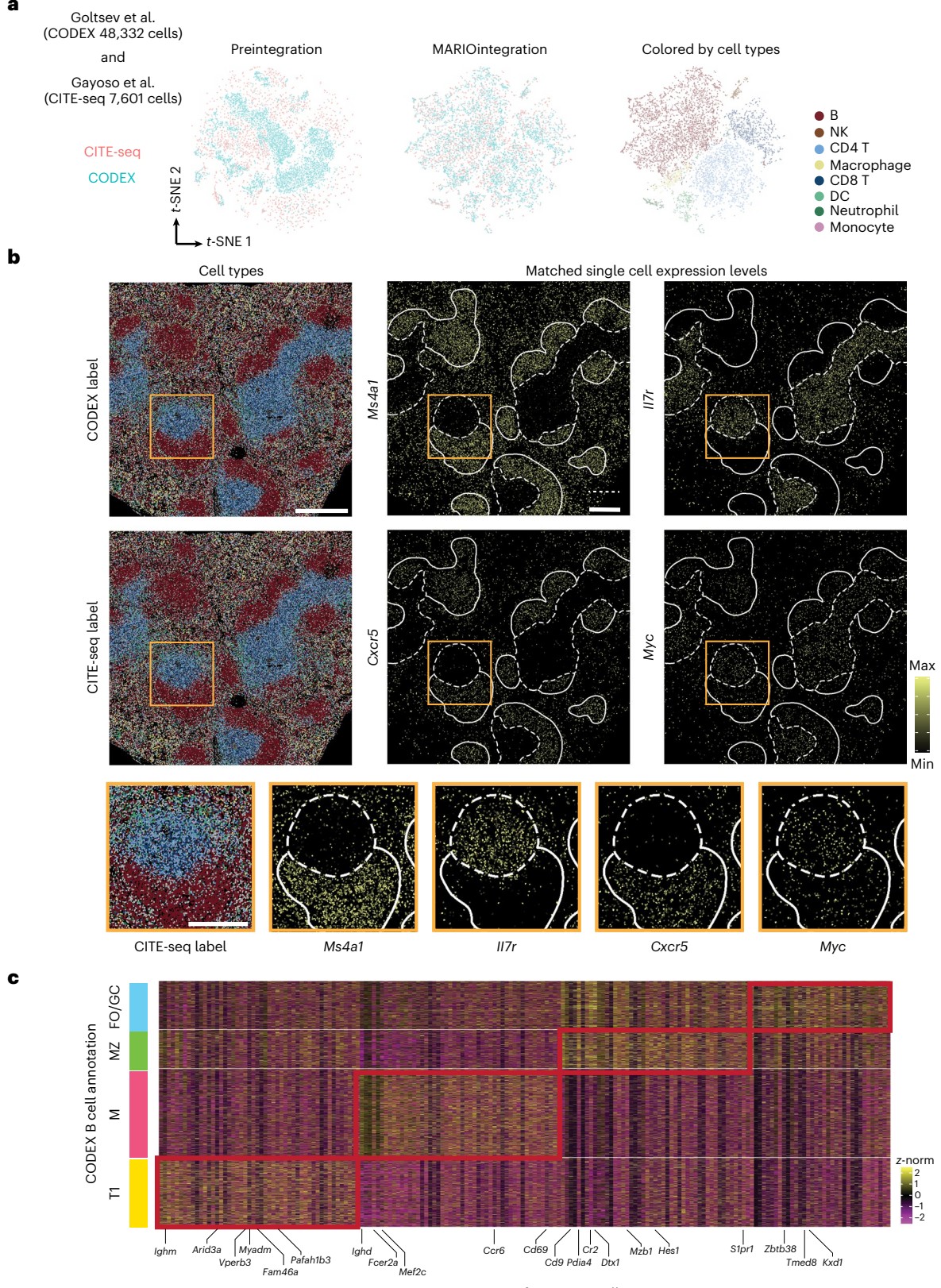

**Fig. 4 | Spatial multi-omics in murine spleen enabled by MARIO matching.**
**a**, *t*-SNE plots of murine spleen CITE-seq and CODEX cells, preintegration and MARIO integration, colored by the dataset of origin (left and middle) or colored by cell-type annotation (right). **b**, A murine spleen section colored by the cell-type annotation from CODEX (top left) or the label transferred annotation from CITE-seq (middle left). Examples of RNA transcripts ((*Il7r*, *Ms4a1*, *Cxcr5* and *Myc*) and their tissue-specific localization are inferred through MARIO integrative analysis (middle

and right columns). An enlarged view of the tissue region demarcated by the orange box is shown in the bottom row. Scale bars, 400 μm (upper) and 200 μm (lower). Dotted line, T cell zone; white line, B cell zone. **c**, Heatmap of differentially expressed genes (from matched CITE-seq cells) among subpopulations of CODEX B cells, gated based on CODEX proteins. Four subpopulations of B cells were identified: transitional type 1 B cells (T1), marginal zone B cells (MZ), mature B cells (M), and follicular/germinal center B cells (FO/GC).

cells measured using CITE-seq and CODEX spatially resolved cells. We visually observed highly concordant spatial organization of cell types annotated using CODEX or CITE-seq information and further observed a clear distribution pattern of single-cell transcript expression levels (based on matched individual CITE-seq cells for CODEX cells) corresponding to their expected spatial localization in the spleen (Fig. 4b and Extended Data Fig. 6b). For example, *Il7r* is concentrated in the T cell zone as expected[32]; *Myc* and *Cxcr5* are localized to activated and proliferating T and B cells within the germinal center[33,34]; *Ms4a1* and *Bhlhe41* are highly expressed in the B cell zone and B cells in the red pulp region[35–38] and *Il1b* is expressed outside the B cell zone[39]. *t*-SNE (visualized using CODEX proteins alone) overlays of the matched protein and RNA expression confirmed expected RNA expression profiles within given cell types (Extended Data Fig. 6c).

We next sought to further refine cells from the B lymphocyte lineage by gating the B cell population from the CODEX dataset. Four subpopulations of B cells were identified: transitional type 1 B cells, marginal zone B cells, mature B cells and follicular/germinal center B cells. Visual inspection of the spatial location of these four subtypes of B cell confirmed localization within mouse spleens consistent with previous observations (Extended Data Fig. 6d)[40,41]. MARIO matching thus enabled a detailed examination of the differentially expressed transcripts within these B cell subtypes resolved by CODEX, revealing a distinctive transcriptional program reflective of their phenotype (Fig. 4c)[42–44]. These genes were significantly upregulated (*P* adjusted < 0.05, Wilcoxon Test, two-sided) in the corresponding gated populations of CODEX B cells. In addition, we also confirmed the B cell subtypes as originally annotated by transcriptomic information from Gayaso et al. successfully localized to corresponding spatial niches after MARIO matching (Extended Data Fig. 6e).

For this CODEX to CITE-seq matching, MARIO had matching accuracy superior to mNN methods (Extended Data Fig. 7a). For the full 28-antibody shared panel, the total accuracy for MARIO was 87.76% and the second best method (fastMNN) was 87.40%. Dropping eight shared antibodies resulted in total accuracies of 85.31% for MARIO and the second best method (fastMNN) was 82.01%. MARIO prevented over-integration due to poor quality data, whereas the mNN methods forced matching (Extended Data Fig. 7b). MARIO was also robust in resisting changes to cell-type composition (Extended Data Fig. 7c,d and Supplementary Fig. 17). Additionally, we removed shared protein markers as previously tested, but in the order of importance score, and MARIO still consistently outperformed other methods in matching accuracy (Supplementary Fig. 18).

### Generation of a multi-omic COVID-19 lung molecular atlas

We reasoned that the ability to perform integrative and inferential analysis across biologically analogous clinical cohorts, measured at different institutions with varying technologies, would further our understanding of the facets of COVID-19 biology, including a study in which BALF samples were subjected to CITE-seq[45]. We additionally profiled 76 lung tissue regions from 23 individuals who succumbed to COVID-19 using CODEX imaging with 54 markers (Supplementary Table 1) and observed the abundance of macrophages in both CITE-seq (15.8% of total cells) and CODEX (31.3% of total cells) cohorts. The large overlap in antibody panels of both studies allowed the robust matching and subsequent functional interrogation of macrophages with high granularity (Fig. 5a).

We were able to stratify the macrophages into two populations based on their transcriptional signatures of complement pathway activity (Fig. 5b; *C1Q* low and high). Such stratification is challenging without using MARIO matching and solely relying on macrophage-related protein markers, including canonical M1 and M2 markers (Supplementary Fig. 19). However, protein expression of these two classes of macrophages partially corresponded to an M1 phenotype for *C1Q* low macrophages, and an immunosuppressive M2 phenotype for *C1Q* high macrophages (Fig. 5b). We further observed that the *C1Q* high transcriptional program was enriched in antigen processing and presentation, whereas that of the *C1Q* low population consisted of several immune chemotaxis and migration pathways including that of neutrophil chemoattractants (Extended Data Fig. 8a). The top differentially expressed transcripts included *CXCL8*, *CCL7*, and *TMEM176B*, with previously described roles in regulating neutrophil recruitment and migration[46–48]. The roles of proteins encoded by *IL1B*, *S100A8*, and *CCL2* in the recruitment of aberrant neutrophils have been recently elucidated in NHP and mice models of SARS-CoV-2 lung pathology[49], and are also reflected by elevated transcript levels in *C1Q* low macrophages (Extended Data Fig. 8b).

In the five previously established functional clusters of interferon stimulated genes (ISG)[50,51], we observed distinctive ISG transcriptional programs in *C1Q* low and high macrophages (Extended Data Fig. 8c; *P* adjusted <0.05, Wilcoxon Test, two-sided) across all clusters (Extended Data Fig. 8c, ISG clusters 1-05). Our results indicate the activation of the innate immunological pathway, including several previously characterized genes (*SERPINB9*, *CKAP4*, *CCL2* and *SPHK1*)[52–55], in C1Q low macrophages to inhibit SARS-CoV-2 replication and entry. The failure to subsequently regulate and dampen this innate response resulted in unchecked host immune responses and collateral tissue damage for C1Q low macrophages, while C1Q high macrophages have elevated complement cascade activation (for example, *LGALS3BP*[56]) and express genes correlating with mild rather than severe COVID-19 symptoms (for example, *SIGLEC1*, ref. 57).

In line with the transcriptional signatures for aberrant neutrophil infiltration, we noted a correlation between the presence of *C1Q* low macrophages and increased infiltrating neutrophils (Fig. 5c–e; $\rho = -0.453$, $P < 0.01$). This elevated neutrophil presence was also confirmed visually (Fig. 5f,g and Extended Data Fig. 9a). Spatial cell–cell

**Fig. 5 | Spatial multi-omic analysis of macrophages in patients with COVID-19.** **a**, Schematic of the experimental and MARIO analysis on BALF and lung tissues from patients with COVID-19 were measured from two independent studies via CITE-seq (from VIB/Ghent) and CODEX (University Hospital Basel/Stanford). Macrophages from the CODEX were matched to those in CITE-seq. **b**, Heatmaps of *C1Q* high and low macrophages identified from CITE-seq, and their matched CITE-seq and CODEX expression patterns. **c**, A ranked plot (median ± 1.5 IQR) for macrophages from each patient (*n* = 23) in the CODEX data, as a percentage of *C1Q* high proportions. **d**, Proportion of neutrophils (of all cell types) in each patient from the CODEX data, ranked by the same sequence as in **c** (*P* value and correlation calculated by a two-sided Spearman-ranked test). **e**, A dot plot with 95% CI showing the relationship between *C1Q* high macrophages (*y* axis) and neutrophil percentage (*x* axis). Each dot represents a tissue core (*n* = 76). Colors represent patients. **f**, A representative pseudo image of two tissue cores colored with the locations of *C1Q* high and low macrophages. **g**, The CODEX images of the same two tissue cores in **f**, with CD163, CD68 and CD15 antibody staining.

Scale bars, 400 μm (left) and 100 μm (right). **h**, An experimental schematic of PANINI to validate the spatial localization of *C1Q* macrophages on Basel/Stanford COVID-19 tissues. Slides were costained with probes detecting *C1QA* mRNA and antibodies targeting CD15 and CD68 proteins. **i,j**, A dot plot with 95% CI showing the relationship between the proportion of *C1QA* high macrophages (to all macrophages) from the PANINI validation (*y* axis) versus the MARIO prediction (*x* axis) per patient (**i**) or per tissue core (**j**). *P* values and correlations were calculated using a two-sided Spearman-ranked test. **k**, Anchor plots of neutrophils as a function of distance from *C1QA* high or low macrophages in MARIO-predicted (above) or PANINI-validated (below) experiments. **l,m**, A representative tissue core with MARIO-predicted *C1QA* expression levels in macrophages (left), and PANINI-validated *C1QA* and CD68 signals (right). Scale bar, 400 μm. **n**, Dot plot with 95% CI of spatial-correlations of C1QA between validation and prediction experiments (*P* value and correlation calculated by a two-sided Spearman-ranked test). Each dot represents a region in the core.

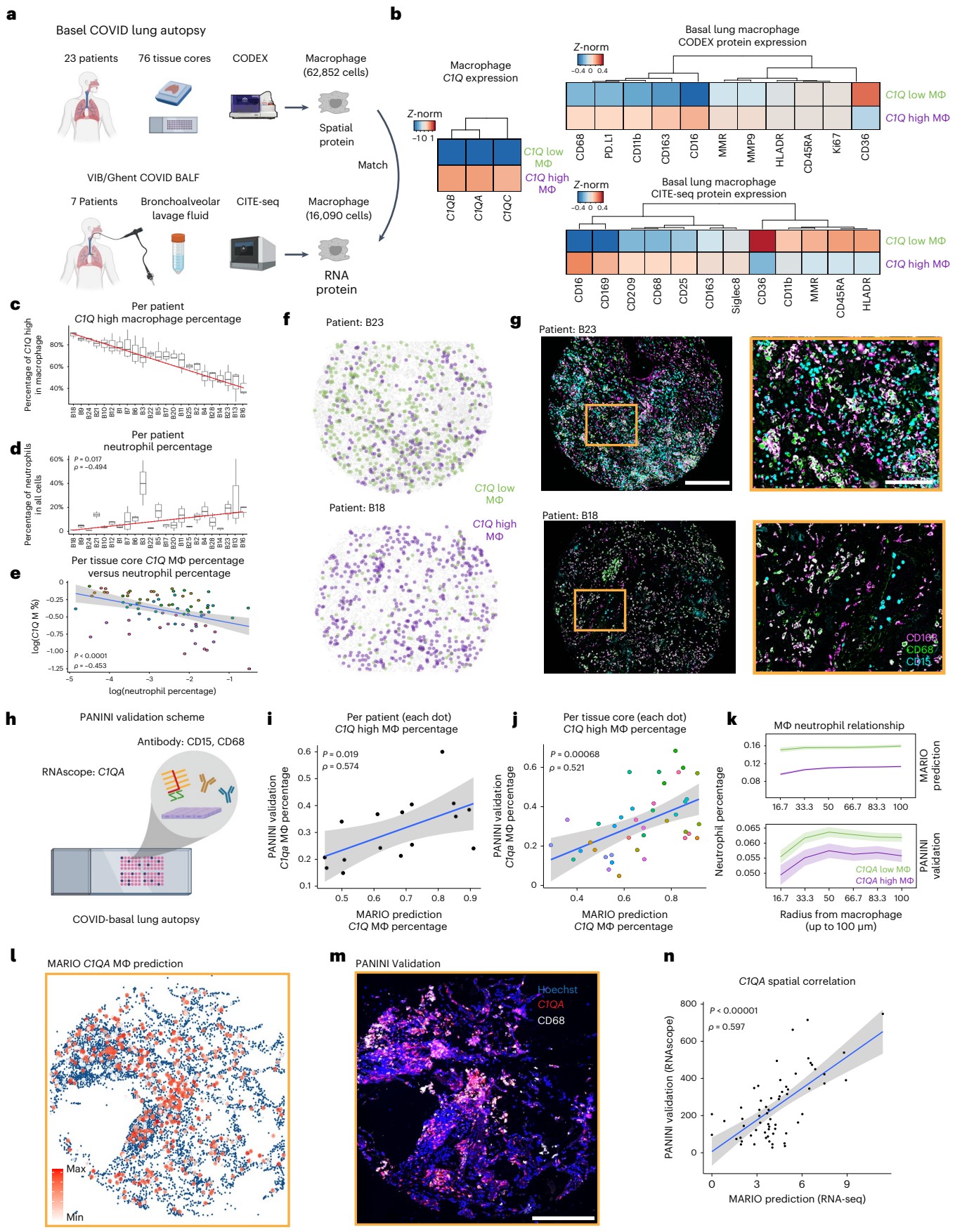

interaction analysis showed differences in these two subclasses of macrophages and their proximity to other cell types, such as high frequency of *C1Q* high macrophages proximal to CD4 and CD8 T cells, B cells, myeloid cells and other macrophages (Extended Data Fig. 9b). We next centered *C1Q* high and low macrophages for an anchor analysis[58] to understand the microenvironment as a function of distance around these two groups of macrophages. Our analysis confirmed the distinctive microenvironments around these macrophages, as evident from the differential organization of macrophages, plasma cells, vasculature and CD8 T cells (Extended Data Fig. 9c).

We finally performed protein and nucleic acid in situ imaging (PANINI)[58] to visualize the messenger RNA of a complement marker, *C1QA*, the neutrophil marker CD15 and the macrophage marker CD68 on COVID-19 tissue microarray sections to experimentally validate the spatially resolved gene-expression patterns predicted by MARIO (Fig. 5h). We confirmed the robust expression patterns of *C1QA* mRNA, CD68 and CD15 proteins in the tissue sections (Extended Data Fig. 9d). We observed a significant correlation between the percentages of experimentally validated *C1Q* High macrophages and MARIO-predicted *C1Q* High macrophages percentage, both at the patient level ($P = 0.019$, $\rho = 0.574$) and at the per tissue core level ($P < 0.01$, $\rho = 0.521$, Spearman-ranked test, Fig. 5i,j). In line with anchor analysis from MARIO-inferred data, we confirmed a significantly decreased neutrophil density around *C1Q* high macrophages in the PANINI validation experiment (Fig. 5k). The RNA spatial pattern from our PANINI experiment, performed on a separate, nonadjacent section of the same patient tissue core, recapitulated the prediction from the MARIO-matched data (Fig. 5l,m). The spatial correlation between MARIO-predicted and PANINI-validated expression levels of *C1QA* in macrophages was highly consistent (*C1QA* signal per region $P < 0.01$, $\rho = 0.597$, Spearman-ranked test, Fig. 5n). This $\rho$ value was close to the maximum possible spatial correlation of the tissue structure as determined using cell density per region ($P < 0.01$, $\rho = 0.602$, Extended Data Fig. 9e), validating the highly accurate inferential capabilities of MARIO.

### Parameter choices, computational resource usage and algorithmic alternatives

MARIO is generally highly robust with respect to different parameter choices for running (Supplementary Figs. 20 and 21). Given the globally optimal nature of the core matching algorithm implemented in MARIO, the time required to run the MARIO pipeline is cubically related to the number of cells. To circumvent this, in the actual implementation of the pipeline matching is automatically performed in batches, thus the time and memory usage is linear rather than cubic, in relation to the dataset size. We also further developed a sparsification technique that reduces the search space to accelerate the matching process. Empirically, we found that MARIO can be run on datasets with moderate sample sizes within reasonable time and memory usage (Supplementary Fig. 22). We also observed that the distance matrix constructed in MARIO (using Pearson correlation) is computationally efficient and generally produces better matching outcome compared to more complicated distance matrices (Extended Data Fig. 10). We also tested alternative algorithms, such as optimal transport (SpaOTsc[59]) as another potential approach for matching of cells beyond the scope of this work (Supplementary Fig. 23).

### Discussion

MARIO is a powerful matching and integration framework for single cells that allows the retention of distinct features. It is particularly suitable for the integration of single-cell proteomic datasets with limited antibody panel overlap. The analysis pipeline builds on several rigorous mathematical advances. First, the matching is constructed by globally (rather than locally) optimizing over a new distance matrix that incorporates both the explicit correlations in shared features and the hidden correlations among distinct features. Second, the accuracy and robustness of the matching are ensured by two theoretically principled quality control processes: the matchability test and jointly regularized filtering[60]. Third, the integrated embeddings are obtained via CCA or gCCA, which incorporates the information in both the shared and distinct features.

In spite of the clear advantages of MARIO, it has some technical limitations. First, the accuracy and robustness may come at the cost of longer analysis times compared to mNN-based approaches. Second, the prerequisite of performing such matching across datasets is that these datasets should be very similar, thus if certain cell types or cell states are missing in one modality, the matching and integration performance can potentially be affected. Third, although in all the benchmarking scenarios tested in the paper MARIO showed better tolerance of the antibody panel difference between datasets being matched comparing to mNN-based methods, the matching accuracy will still eventually drop below a biologically relevant level when too little information is shared across datasets. Thus, the exact minimal requirement for matching will depend on each dataset itself, marker panels and the biological goal the user wants to achieve. Fourth, while the distance matrix constructed in MARIO defaults to using Pearson correlation, to better accommodate specific requirements from future users, we supplied the option to use nonlinear kernels (Laplacian) instead of Pearson to construct the distance matrix, per user's choice. Last, linear assignment (MARIO), mNN (for example, Seurat, Scanorama, fastMNN and more) and various other recent methods (for example, SpaOTsc) are all capable of matching cells across modalities. Future iterations of these approaches will be of broad interest to the field.

The need to study biological processes within their tissue context is increasingly evident, with direct relevance to the physiological context of health and disease. The ability to match similar biological samples measured using distinctive single-cell assays will be paramount for hypothesis generation and guidance for experimental design. We are confident that MARIO will serve as a useful methodology and resource for the community with direct applications to a plethora of experimental platforms and biological contexts.

### Online content

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

## Methods

Complete methods, including details of the data analysis process and extensions of the method summarized below, are available in the Supplementary Notes.

### MARIO pipeline

Before the input of MARIO, data were encouraged to go through standard preprocessing pipeline (for example, normalization and scaling) suggested by their originated modality. Suppose the two datasets are denoted as $X$ and $Y$, where $X \in \mathbb{R}^{n_x \times (p_{share} + p_x)}$ consists of $n_x$ cells and $(p_{share} + p_x)$ features and $Y \in \mathbb{R}^{n_y \times (p_{share} + p_y)}$ consists of $n_y$ cells and $(p_{share} + p_y)$ features. The matching implemented by MARIO is a linear assignment problem, therefore requires $n_x \leq n_y$. If data size input does not fulfill such a requirement, $X$ can be randomly segmented into equal-sized batches, and matching will be performed on each batch, as per the user's request. Among all the features, $n_{share}$ features are shared across both datasets, whereas the rest of the features are distinct to either $X$ or $Y$. Thus, we can write both datasets as horizontal concatenations of a shared part and a distinct part:

$$X = (X_{share} \quad X_{dist}), \qquad Y = (Y_{share} \quad Y_{dist}).$$

The cell matching between $X$ and $Y$ is defined as an injective map $\Pi$, represented as a binary matrix of dimension $n_x \times n_y$, such that $\Pi_{i,i'} = 1$ if and only if the $i$th cell in $X$ shares a similar biological state to the $i'$th cell in $Y$.

**Initial matching with shared features.** We first construct an initial estimator of $\Pi$ using shared features alone. The procedure starts by denoising the shared parts via thresholding their singular values. Consider the singular value decomposition of the vertical concatenation of $X_{share}$ and $Y_{share}$:

$$\begin{pmatrix} X_{share} \\ Y_{share} \end{pmatrix} = \begin{pmatrix} \hat{U}_{share} \\ \tilde{U}_{share} \end{pmatrix} \hat{D}_{share} \hat{V}_{share}^{\top},$$

where the vertical concatenation of $\hat{U}_{share} \in \mathbb{R}^{n_x \times p_{share}}$ and $\tilde{U}_{share} \in \mathbb{R}^{n_y \times p_{share}}$ collects the left singular vectors, $\hat{D}_{share} \in \mathbb{R}^{p_{share} \times p_{share}}$ is a diagonal matrix that collects the singular values in descending order, and $\hat{V}_{share}$ collects the right singular vectors. Let $\hat{r}_{share} \leq p_{share}$ be the number of components to keep. We then compute the denoised version of $X_{share}$ and $Y_{share}$ by

$$\hat{X}_{share} = (\hat{U}_{share})_{.,1:\hat{r}_{share}} (\hat{D}_{share})_{1:\hat{r}_{share}} (\hat{V}_{share})_{.,1:\hat{r}_{share}}^{\top},$$

$$\hat{Y}_{share} = (\tilde{U}_{share})_{.,1:\hat{r}_{share}} (\hat{D}_{share})_{1:\hat{r}_{share}} (\hat{V}_{share})_{.,1:\hat{r}_{share}}^{\top},$$

respectively, where for a matrix $A$, we let $A_{.,1:r}$ denote its first $r$ columns and for a diagonal matrix $D$, we let $D_{1:r}$ denote the submatrix formed by taking its first $r$ rows and columns. We then construct a cross-data distance matrix $\mathscr{D}_{share} \in \mathbb{R}^{n_x \times n_y}$, whose entries are given by

$$(\mathscr{D}_{share})_{i,i'} = 1 - \mathrm{cor}[(\hat{X}_{share})_{i,.}, (\hat{Y}_{share})_{i',.}],$$

where $\mathrm{cor}[(\hat{X}_{share})_{i,.}, (\hat{Y}_{share})_{i',.}]$ is the Pearson correlation coefficient between the $i$th row of $\hat{X}_{share}$ and the $i'$th row of $\hat{Y}_{share}$. The initial estimator of $\Pi$ is given by:

$$\hat{\Pi}_{share} \in \underset{\Pi}{\mathrm{argmin}} \langle \Pi, \mathscr{D}_{share} \rangle$$
$$\text{subject to } \Pi \in \{0,1\}^{n_x \times n_y}, \Pi \mathbf{1}_{n_y} = \mathbf{1}_{n_x}, \tag{1}$$

where for two matrices $A$ and $B$, we let $\langle A, B \rangle = \sum_{i,i'} A_{i,i'} B_{i,i'}$ denote the Frobenius inner product. This optimization problem is an instance of minimal weight bipartite matching (also known as rectangular linear

assignment problem) in the literature[61]. We refer readers to ref. [62] for the optimality of this procedure.

**Refined matching with distinct features.** Given the initial matching $\hat{\Pi}_{share}$, we can approximately align cells in $X$ and $Y$: the rows of $X$ and $\hat{\Pi}_{share} Y$ correspond to pairs of cells with similar biological states, up to mismatches induced by the estimation error of $\hat{\Pi}_{share}$. Such an approximate alignment opens up the possibility of estimating the latent representations of $X$ and $Y$ by CCA.

Let $1 \leq \hat{r}_{all} \leq p_{share} + \min(p_x, p_y)$ be the number of components to keep. Collecting top $\hat{r}_{all}$ sample canonical vectors into matrices

$$\hat{W}_x = \begin{pmatrix} \hat{w}_x^{(1)} & \cdots & \hat{w}_x^{(\hat{r}_{all})} \end{pmatrix},$$
$$\hat{W}_y = \begin{pmatrix} \hat{w}_y^{(1)} & \cdots & \hat{w}_y^{(\hat{r}_{all})} \end{pmatrix},$$

the latent representation of $X$ can be estimated by $X\hat{W}_x$, the sample canonical scores of $X$. The same projection can be done on $Y$ data by computing $Y\hat{W}_y$.

We can now compute the cross-data distance matrix $\mathscr{D}_{all}$ directly on the latent space, whose entries are

$$(\mathscr{D}_{all})_{i,i'} = 1 - \mathrm{cor}[(X\hat{W}_x)_{i,.}, (Y\hat{W}_y)_{i,.}].$$

We finally solve for a refined matching by

$$\hat{\Pi}_{all} \in \underset{\Pi}{\mathrm{argmin}} \langle \Pi, \mathscr{D}_{all} \rangle$$
$$\text{subject to } \Pi \in \{0,1\}^{n_x \times n_y}, \quad \Pi \mathbf{1}_{n_y} = \mathbf{1}_{n_x}. \tag{2}$$

**Interpolation of initial and refined matchings.** The quality of the refined matching $\hat{\Pi}_{all}$ is highly contingent on the quality of the distinct features. If the distinct features are extremely noisy, incorporation of them may hurt the performance, in which case it is more desirable to revert back to the initial matching $\hat{\Pi}_{share}$. We developed a data-adaptive way of deciding how much distinct information shall be incorporated when we estimate the matching from the data.

To start with, we cut the unit interval $[0, 1]$ into grids (for example, $\{0, 0.1, \ldots, 0.9, 1\}$). For each $\lambda$ on the grid, we interpolate the two kinds of distance matrix by taking their convex combination

$$\mathscr{D}_\lambda = (1 - \lambda)\mathscr{D}_{share} + \lambda \mathscr{D}_{all},$$

from which we can solve for the $\lambda$-interpolated matching

$$\hat{\Pi}_\lambda \in \underset{\Pi}{\mathrm{argmin}} \langle \Pi, \mathscr{D}_\lambda \rangle$$
$$\text{subject to } \Pi \in \{0,1\}^{n_x \times n_y}, \quad \Pi \mathbf{1}_{n_y} = \mathbf{1}_{n_x}. \tag{3}$$

Note that $\hat{\Pi}_{\lambda=0} = \hat{\Pi}_{share}$ and $\hat{\Pi}_{\lambda=1} = \hat{\Pi}_{all}$. After aligning $X$ and $Y$ using $\hat{\Pi}_\lambda$, we compute top $k$-sample canonical correlations (in the MARIO package, defaulted to 10), whose mean is taken as a proxy of the quality of $\hat{\Pi}_\lambda$. We then select the best $\hat{\lambda}$ according to this quality measure and use $\hat{\Pi}_{\hat{\lambda}}$ afterward.

### Quality control

**Test of matchability.** In extreme cases, the two datasets $X$ and $Y$ may not have any correlation at all, and thus any attempt to integrate both datasets would give unreliable results. For example, some methods, when applied to uncorrelated datasets, would pick up the spurious correlations and hence resulting in over-integration. A robust procedure should be able to warn the users when the resulting matching estimator might be of low quality. We develop a rigorous hypothesis test, termed matchability test, for this purpose.

The matchability test starts by repeatedly drawing $B$ independent and identically distributed copies of $n_x$-dimensional (potentially asymmetric) Rademacher random vectors $\{\varepsilon_x^{(b)}\}_{b=1}^B$ and another $B$ independent and identically distributed copies of $n_y$-dimensional Rademacher random vectors $\{\varepsilon_y^{(b)}\}_{b=1}^B$. That is, for each $1 \le b \le B$, we have $\varepsilon_*^{(b)} = (\varepsilon_{*,1}^{(b)}, \dots, \varepsilon_{*,n_*}^b)$, and $\varepsilon_{*,i}^{(b)}$ is $+1$ with probability $1 - p_{\text{flip}}$ and is $-1$ otherwise for any $1 \le i \le n_*$, where $*$ is the placeholder for either mathttx or mathtty. The parameter $p_{\text{flip}}$ (denoted as flip_prob in MARIO package and defaulted to 0.2) controls the 'sensitivity' of the test—a lower value of $p_{\text{flip}}$ means that a more accurate matching is needed to pass the matchability test. For every $b$, we generate a fake pair of datasets by flipping the signs of each row of $X$ and $Y$:

$$X^{(b)} = \text{diag}(\varepsilon_x^{(b)})X, \qquad Y^{(b)} = \text{diag}(\varepsilon_y^{(b)})Y.$$

After such a sign-flipping procedure, most of the correlation between $X$ and $Y$ (if exists) is destroyed, but the intra-dataset covariance structures of both $X$ and $Y$ are preserved. As a result, if we run any matching algorithm with $X^{(b)}$ and $Y^{(b)}$ as the input, the resulting estimator $\hat{\Pi}^{(b)}$ would be of low quality, in the sense that if we align $X^{(b)}$, $Y^{(b)}$ using $\hat{\Pi}^{(b)}$ and run CCA, the resulting sample canonical correlations will be small. In our implementation, we calculate the mean of top_$k$ (and defaulted to 10), which we denote as $\{\text{côr}^{(b)}\}_{b=1}^B$.

The matchability test proceeds by running the same algorithm on the real datasets $X$, $Y$, aligning them using the estimator $\hat{\Pi}$, and calculates the mean of top_$k$ sample canonical correlations, which we denote as côr. The final $P$ value for testing the null that $X$ and $Y$ are uncorrelated is given by the proportion of $\{\text{côr}^{(b)}\}_{b=1}^B$ that are larger than the observed côr.

**Jointly regularized filtering of low-quality matched pairs.** Even if the two datasets $X$ and $Y$ are highly correlated (and thus the matchability test gives a small $P$ value), the estimated matching $\hat{\Pi}$ might still be error-prone. For example, consider the case where some cell type exists in $X$ but is completely absent in $Y$. We developed an algorithm that automatically filters out the low-quality matched pairs in $\hat{\Pi}$.

Assume there are $K$ cell types present in either $X$ or $Y$. In the MARIO package we default $K$ to 10. Let $z_x, z_y \in \{1, \dots, K\}^{n_x}$ be the unknown ground-truth cell-type labels of $X$ and $\hat{\Pi}Y$, respectively. The fact that $X$ and $Y$ have passed the matchability test tells that $z_x$ and $z_y$ should agree on most coordinates. However, it is possible that there exists a sparse subset of $\{1, \dots, n_x\}$ on which $z_x$ and $z_y$ disagree, and our goal is to detect this sparse subset and disregard them in downstream analyses. To achieve this goal, we consider the following regularized $k$-means clustering objective:

$$
\begin{aligned}
(\hat{z}_\star, \hat{z}_x, \hat{z}_y) = \mathop{\text{argmin}}_{\substack{\{\mu_k\}_{k=1}^K \subset \mathbb{R}^{p_{\text{share}}+p_x} \\ \{v_k\}_{k=1}^K \subset \mathbb{R}^{n_{\text{share}}+n_y} \\ z_\star, z_x, z_y \in \{1, \dots K\}^{n_x}}} \\
\frac{1}{2} \sum_{i=1}^{n_x} \left( \| X_{i,\cdot} - \mu_{z_{x,i}} \|_2^2 + \| Y_{i,\cdot} - v_{z_{y,i}} \|_2^2 \right) \\
+ \log\left(\frac{1-\rho}{\rho/(K-1)}\right) \sum_{i=1}^{n_x} \left( \mathbb{1}\{z_{x,i} \ne z_{\star,i}\} + \mathbb{1}\{z_{y,i} \ne z_{\star,i}\} \right),
\end{aligned}
$$

where $\| \cdot \|_2$ is the $\ell_2$ norm and $\mathbb{1}\{\cdot\}$ is the indicator function. The above objective function is composed of two parts. The first part is the classical $k$-means objective for $X$ and $Y$, and the second part is a regularization term that penalizes when the estimated $X$ label $\hat{z}_x$ and $Y$ label $\hat{z}_y$ are too far away from a global label $\hat{z}_\star$.

After solving the above objective function, if $\hat{z}_{x,i} \ne \hat{z}_{y,i}$, then there is evidence that the matched pair $(X_{i,\cdot}, (\hat{\Pi}Y)_{i,\cdot})$ is spurious, and is thus disregarded in the downstream analyses. The parameter $\rho$ controls the strength of regularization: if $\rho = 1 - 1/K$, then there is no regularization at all, whereas if $\rho = 0$, we effectively require $\hat{z}_\star = \hat{z}_x = \hat{z}_y$. Thus, we can naturally control the 'intensity' of such a filtering procedure by choosing a suitable $\rho$. Under a hierarchical Bayesian model, the parameter $\rho$ has a rather intuitive interpretation as the probability of disagreement between individual labels and global labels[60].

We solve the regularized $k$-means clustering objective via a warm-started block coordinate descent algorithm. The algorithm starts by computing initial estimators $\hat{z}_x^{(0)}, \hat{z}_y^{(0)}$ of $z_x, z_y$ via spectral clustering[63]: we compute the sample canonical scores of $X$ and $\hat{\Pi}Y$, average them, and apply the classical $k$-means clustering on top $K$ eigenvectors of the averaged score to get $\bar{z} \in \{1, \dots, K\}^{n_x}$. We then let $\hat{z}_x^{(0)} = \hat{z}_y^{(0)} = \bar{z}$.

Suppose at iteration $t$, the current estimators of $z_x, z_y$ are given by $\hat{z}_x^{(t)}, \hat{z}_y^{(t)}$, respectively. We run block coordinate descent as follows:

(1) Given $\hat{z}_x^{(t)}, \hat{z}_y^{(t)}$, the current estimators of $\{\mu_k\}$, $\{v_k\}$ are given by

$$
\hat{\mu}_k^{(t)} = \frac{1}{\sum_{i=1}^{n_x} \mathbb{1}\{\hat{z}_{x,i}^{(t)}=k\}} \sum_{i=1}^{n_x} \mathbb{1}\left\{\hat{z}_{x,i}^{(t)} = k\right\} \times X_{i,\cdot},
$$

$$
\hat{v}_k^{(t)} = \frac{1}{\sum_{i=1}^{n_x} \mathbb{1}\{\hat{z}_{y,i}^{(t)}=k\}} \sum_{i=1}^{n_x} \mathbb{1}\left\{\hat{z}_{y,i}^{(t)} = k\right\} \times Y_{i,\cdot},
$$

for any $1 \le k \le K$.
(2) Given $\{\hat{\mu}_k^{(t)}\}, \{\hat{v}_k^{(t)}\}$, the next estimators of $z_\star, z_x, z_y$ are given by

$$
\begin{aligned}
(\hat{z}_{\star,i}^{(t+1)}, \hat{z}_{x,i}^{(t+1)}, \hat{z}_{y,i}^{(t+1)}) = \mathop{\text{argmin}}_{z_\star, z_x, z_y \in \{1, \dots K\}^{n_x}} \\
\frac{1}{2}\left( \| X_{i,\cdot} - \hat{\mu}_{z_{x,i}}^{(t)} \|_2^2 + \| Y_{i,\cdot} - \hat{v}_{z_{y,i}}^{(t)} \|_2^2 \right) \\
+ \log\left(\frac{1-\rho}{\rho/(K-1)}\right)\left( \mathbb{1}\{z_{x,i} \ne z_{\star,i}\} + \mathbb{1}\{z_{y,i} \ne z_{\star,i}\} \right)
\end{aligned}
$$

for any $1 \le i \le n_x$. The above problem is solved via an enumeration procedure. We first hypothesize that $\hat{z}_{\star,i}^{(t+1)} = k$ for some $1 \le k \le K$. We then solve for the best $\hat{z}_{x,i}^{(t+1)}$ by enumerating all $K$ possible choices of labels. The same thing can be done to solve for the best $\hat{z}_{y,i}^{(t+1)}$. Hence, we can compute the best value of the above objective function under the hypothesis that $\hat{z}_{\star,i}^{(t+1)} = k$. We can then solve for the global optimal $\hat{z}_{\star,i}^{(t+1)}$ by enumerating and comparing the objective values under every possible hypothesized value of $\hat{z}_{\star,i}^{(t+1)} = 1, \dots, K$. Given the global optimal $\hat{z}_{\star,i}^{(t+1)}$, the global optimal $\hat{z}_x^{(t+1)}$ and $\hat{z}_y^{(t+1)}$ can be extracted.

In our implementation, we run the above procedure for 20 iterations.

**The objective function of MARIO.** In this subsection, we formulate the whole MARIO pipeline into a single optimization problem. Let $X$ and $Y$ be the two data matrices (rows as cells and columns as features). Without loss of generality, we assume that $X$ has at most as many rows as $Y$. Thus, there are more or as many cells in $Y$ compared with $X$. Suppose there are $n$ rows in $X$ and $m$ rows in $Y$, then $n \le m$. MARIO is an algorithm aimed at solving the following optimization problem:

$$
\begin{aligned}
\text{maximize} \quad & \text{Tr}(A^\top X^\top \Pi HYB) \\
\text{subject to} \quad & \Pi \in S(n, m), \\
& A^\top X^\top XA = I, B^\top Y^\top YB = I.
\end{aligned}
$$

Here $H = I_m - \frac{1}{m}\mathbf{1}_m\mathbf{1}_m^T$ is the centering matrix, and $S(n, m)$ is the collection of all binary $n$-by-$m$ matrices such that there are $(m - n)$ zero columns and each of the remaining $m$ columns has one and only one entry equal to one, and each row has one and only one entry equal to one. That is, MARIO aims at simultaneously finding the cell–cell

correspondence matrix and two linear transformations $A$ and $B$ such that after projecting the data matrices $X$ and $Y$ to a common latent space using $A$ and $B$, and selecting a subset of rows of $YB$ and matching them to the rows of $XA$ in a one-to-one fashion, the trace inner product between $XA$ and $YB$ is maximized. By the definition of $S(n, m)$, the matrix selects $n$ rows of $YB$ and then finds a bijection between the selected rows of $YB$ and rows of $XA$.

Suppose both $A$ and $B$ are of rank $k$. The objective function of the optimization problem is a combination of the top $k$ CCA objective function and the (unbalanced) linear assignment problem objective function:

(1) When given, solving for optimal $A$ and $B$ means simultaneously solving for top $k$ canonical correlation loading vectors for the pair $(X, Y)$;

(2) When $A$ and $B$ are given, solving for means exactly solving a linear assignment problem.

### Downstream analysis after cell matching

**Joint embedding.** After running jointly regularized filtering on the best interpolated estimator $\hat{\Pi}_{\hat{\lambda}}$, we get a pair of aligned datasets $X^\star \in \mathbb{R}^{n \times (p_{\mathrm{share}} + p_x)}$, $Y^\star \in \mathbb{R}^{n \times (p_{\mathrm{share}} + p_y)}$, whose rows correspond to cells of similar types and $n$ is the number of remaining cell–cell pairs after filtering. Then, we run CCA on $X^\star$, $Y^\star$ and collect the first $n$ pairs of sample canonical scores (scaled within dataset) as the final embeddings. Note that other standard methods for joint embedding that take row-wise aligned datasets can also be applied.

**Label transfer via $k$-nearest-neighbors matching.** The interpolated distance $\mathscr{D}_{\hat{\lambda}}$ can be used to do label transfer via $k$-nearest-neighbors. Suppose we know the cell types for all cells in $Y$ but the corresponding labels in $X$ are missing. Then for the $i$th cell in $X$, we can predict its label by finding the $k$-nearest cells in $Y$ according to $\mathscr{D}_{\hat{\lambda}}$ and taking the majority vote.

### Systematic benchmarks

**Benchmarking on the matching quality.** Three scenarios were tested during the benchmarking process:

(1) Sequentially dropping shared features between datasets, to test the robustness of the algorithm regardless of the antibody panel design. There are two dropping sequence used: first scheme is dropping antibodies based on their names, in alphabetical order (for example, CD1c is dropped before CD3); the second scheme is dropping by importance score, where less important antibodies were dropped first, mimicking real world antibody panel design. To roughly assess the order of importance of the antibodies at distinguishing cell states, a random forest model for each dataset was trained to predict cell types from marker expression, with the function randomForest in R package randomForest, with default parameters. Then a permutation feature importance test was performed to determine the effects of variables in the model, using function varImp with default parameters in R package caret, hence getting a score for each protein. The importance scores for the protein (shared) were then averaged between the datasets for matching, and ranked from low to high. For the cross-species and murine spleen dataset, only the top 50% important shared markers were considered.

(2) Stimulating poor quality data by adding increasing levels of random noise to both datasets, to test the robustness of the algorithm in terms of over-integration. Gaussian random noise with mean 0 and standard deviations of 0.1, 0.3, 0.5, 0.7, 0.9, 1.1, 1.3 and 1.5 were added to the normalized values of all protein channels.

(3) Intentionally dropping cell types in the dataset being matched against, to test the robustness of the algorithm regardless of the cell-type composition difference between datasets.

In all three scenarios described above, all other compared methods used the exact same set of cells tested by MARIO. For cross-species data (related to Fig. 3 and Supplementary Fig. 7) only H1N1 challenged human and X-species cynomolgus monkey were benchmarked.

The following metrics were used in the benchmarking process:

(1) Matching accuracy: this was calculated by the percentage of cells in $X$ that have paired correctly with the same cell type in $Y$, based on the individual dataset's cell-type annotations.

(2) Matching proportion: this was calculated by the percentage of cells in $X$ that had a match in $Y$ after quality control steps.

(3) Structure alignment score: this measures how much structural information is preserved after data integration. Let $D_{\mathrm{full}}$ be the matrix whose $(i, j)$th entry is the Euclidean distance between the $i$th row and the $j$th row of $X$. Similarly, let $D_{\mathrm{partial}}$ be the matrix whose $(i, j)$th entry is the Euclidean distance between the $i$th row and the $j$th row of the embedding of $X$. The structure alignment score for the $i$th cell in $X$ is defined as the Pearson correlation between the $i$th row of $D_{\mathrm{full}}$ and the $i$th row of $D_{\mathrm{partial}}$. The structure alignment score for $X$ is then defined as the average of the scores over all cells in $X$. The structure alignment score for $Y$ can be similarly obtained. The final structure alignment score is the average of the scores for $X$ and $Y$.

(4) Silhouette F1 score: this has been described in ref. [64] and is an integrated measure of the quality of dataset mixing and information preservation. In brief, two preliminary scores slt_mix and slt_clust were obtained, and the Silhouette F1 score was calculated as $2 \cdot \mathrm{slt\_mix} \cdot \mathrm{slt\_clust} / (\mathrm{slt\_mix} + \mathrm{slt\_clust})$. Here, slt_mix is a measure of dataset mixing and is defined as one minus normalized Silhouette width with the label being dataset index, this is a measure of mixing; slt_clust is a measure of information preservation and is defined as the normalized Silhouette width with label being cell-type annotations. All Silhouette widths were computed using the silhouette() function from R package cluster.

(5) ARI F1 score: this is an integrated measure of the quality of dataset mixing and information preservation[64]. The definition is similar to that of Silhouette F1 score, except that we compute ARI instead of the Silhouette width. All ARI scores were computed using the function adjustedRandIndex() in R package mclust.

(6) Average mixing score: this is a measure of dataset mixing based on the Kolmogorov–Smirnov statistic. For each cluster, the subsets of cells corresponding to that cluster were extracted from the embeddings of $X$ and $Y$, respectively. For each coordinate of the embeddings, one minus the Kolmogorov–Smirnov statistic was computed. The mixing score for that cluster was then computed by taking the median of one minus the Kolmogorov–Smirnov statistic for each coordinate. The average mixing score is defined as the average of mixing scores over all clusters.

(7) Error avoidance score: this measures the performance of the quality control process and is specific to the benchmarking scenario 3 (intentionally dropping cell types). For each cell type dropped, the corresponding error avoidance score is defined as $\sqrt{a/b}$, where $a$ is the number of cells in $X$ that are of that type and have survived the quality control process (that is, a match involving that cell type has occurred), and $b$ is the total number of cells of that type $X$. A higher value of this score indicates that erroneous matching toward deleted cell types has been better avoided.

During benchmarking, all datasets were downsampled. The bone marrow dataset (Fig. 2) was downsampled to 40,000 cells (8,000 and 32,000 for $X$ and $Y$); the PBMC dataset (Supplementary Fig. 3) was downsampled to 25,000 cells (5,000 and 20,000 for $X$ and $Y$);

# Article

the X-Species H1N1/IFN-gamma dataset (Fig. 3) was downsampled to 40,000 cells (8,000 and 32,000 for $X$ and $Y$); the X-Species H1N1/IL-4 dataset (Supplementary Fig. 7) was downsampled to 40,000 cells (8,000 and 32,000 for $X$ and $Y$) and the murine spleen dataset (Fig. 4) was downsampled to 25,000 cells (5,000 and 20,000 for $X$ and $Y$). All methods used the same set of cells.

Parameters used for benchmarking are as follows. For benchmarking of MARIO, we used a consistent set of parameters across all datasets: n_components_ovlp = 10 (or the maximum number available); n_components_all = 20 (or the maximum available), sparsity = 5,000, bad_prop = 0.1 or 0.2, n_batch = 1. For other methods, the input of data were all values normalized per feature within each dataset (Same as MARIO input data, except Liger/UINMF where their own custom normalization is required). Only mNN-based methods (Scanorma, Seurat, fastMNN) were included in the comparison of matching accuracy and matching proportion. For Seurat, three versions were compared (principal components analysis (PCA), CCA and reciprocal PCA). For computation of SAM, ASW, ARI and avgMix, the first 20 (or maximum available) components of MARIO CCA scores or reduced values from other methods were used. For visualization, $t$-SNE plots were produced using the first ten components for all methods. In some rare cases, certain methods produced NAs (Not Avaliable) in the integrated values for limited number of cells, which were replaced with 0 for downstream analysis. Detailed information of the benchmarking process can be retrieved from the deposited code in our GitHub repository.

### Reporting summary

Further information on research design is available in the Nature Portfolio Reporting Summary linked to this article.

## Data availability

Publicly available datasets used were: Levine et al. Human BMC CYTOF at: https://github.com/lmweber/benchmark-data-Levine-32-dim; Stuart et al. Human BMC CITE-seq (from the R package SeuratData, 'bmcite') at https://satijalab.org/seurat/articles/weighted_nearest_neighbor_analysis.html; Zainab et al. Human H1N1 challenged whole blood CYTOF at flow repository 'FR-FCM-Z2NZ'; Bjornson et al. Human and NHP whole blood CYTOF at flow repository 'FRFCM-Z2ZY'; Goltsev et al. Murine Spleen CODEX at https://data.mendeley.com/datasets/zjnpwh8m5b/1 (raw images per reasonable request from the Nolan Laboratory); Gayoso et al. Murine Spleen CITE-seq at https://github.com/YosefLab/totalVI_reproducibility/tree/master/data; COVID-19 Cell Atlas. COVID-19 patient BALF CITE-seq (VIB/Ghent) at https://www.covid19cellatlas.org/index.patient.html; Hartmann et al. Human PBMC CyTOF at flow repository 'FR-FCM-Z249', HD06_run1; 10X Genomics. Human PBMC CITE-seq at https://support.10xgenomics.com/single-cell-gene-expression/datasets/3.0.2/5k_pbmc_protein_v3?. Newly generated data used came from COVID-Lung CODEX imaging expression files (macrophage related) at: https://github.com/shuxiaoc/mario-py/tree/main/Manuscript_Archive_Code/data/COVID-19. Full dataset information, including raw images of the CODEX and PANINI validation experiments, is available on reasonable request. All data mentioned above are also summarized and deposited (with related preprocessing scripts) at https://github.com/shuxiaoc/mario-py.

## Code availability

MARIO and related tutorials are freely available to the public at GitHub https://github.com/shuxiaoc/mario-py. For reproducibility, code to regenerate the main and supplementary figures have also been deposited to GitHub repository.

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

## Acknowledgements

We thank S. Bendall, S. Rodig and members of the Nolan and Jiang laboratories for helpful discussions. B.Z. is supported by a Stanford Graduate Fellowship. This work was funded in part by grants from the National Institutes of Health nos. R01AI149672 (S.J. and G.P.N.), the Bill & Melinda Gates Foundation grant no. INV-002704 (S.J. and G.P.N.), grant no. OPP1113682 (G.P.N.), COVID-19 Pilot Award (S.J., D.R.M. and G.P.N.), the Fast Grant Funding for COVID-19 Science (S.J., D.R.M. and G.P.N.), the Botnar Research Centre for Child Health Emergency Response to COVID-19 grant (S.J., D.R.M., G.P.N., M.S.M. and A.T.), the Hope Foundation to G.P.N., the US Food and Drug Administration Medical Countermeasures Initiative contract nos. HHSF223201610018C and 75F40120C00176 (G.P.N.), the Parker Institute for Cancer Immunotherapy (G.P.N.), the Rachford and Carlota A. Harris Endowed Professorship (G.P.N.), the National Institute Of Allergy And Infectious Diseases of the National Institutes of Health under award number DP2AI171139 (S.J.) and the Gilead Research Scholar in Hematologic Malignancies (S.J.). This article reflects the views of the authors and should not be construed as representing the views or policies of the FDA, NIH, BMGF, Botnar Foundation or other institutions that provided funding.

## Author contributions

Conceptualization was done by B.Z., S.C., Z.M., G.P.N. and S.J. Algorithm development and implementation were carried out by S.C., B.Z. and Z.M. Analysis was done by B.Z., S.C., Y.B., H.C., G.L., I.T.L., Y.G. and S.J. Contribution of key reagents and tools was from N.M., G.V., D.R.M., A.T. and M.S.M. The project was supervised by S.J., G.P.N. and Z.M. Both B.Z. and S.C. contributed equally and have the right to list their name first in their CV.

## Competing interests

G.P.N. received research grants from Pfizer, Inc., Vaxart, Inc., Celgene, Inc. and Juno Therapeutics, Inc. during the course of this work. G.P.N. and Y.G. have equity in Akoya Biosciences, Inc. G.P.N. is a scientific advisory board member of Akoya Biosciences, Inc.

## Additional information

**Extended data** is available for this paper at https://doi.org/10.1038/s41592-022-01709-7.

**Correspondence and requests for materials** should be addressed to Zongming Ma, Garry P. Nolan or Sizun Jiang.

# Simulated High Granularity Groundtruth Data Benchmark

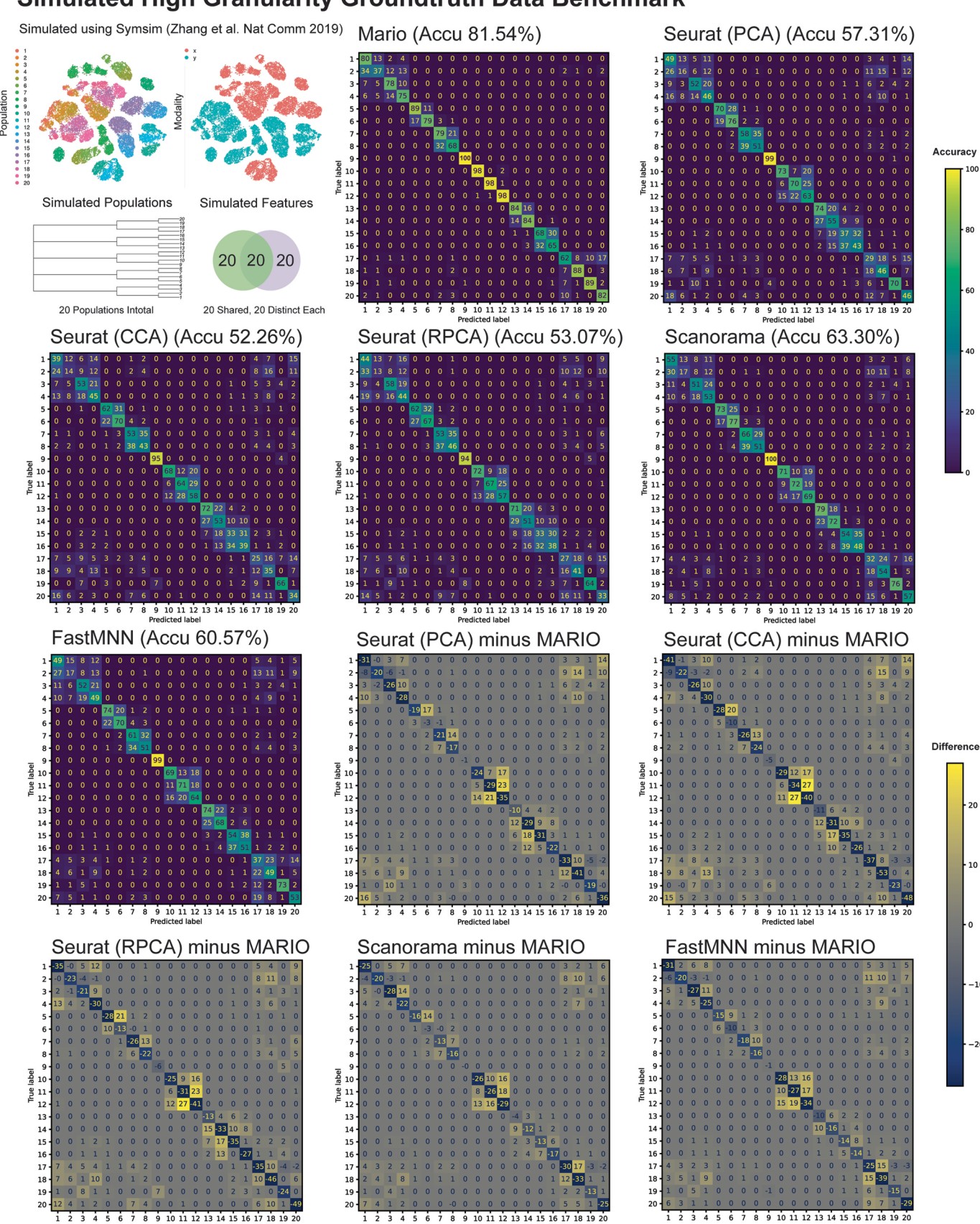

**Extended Data Fig. 1 | MARIO Benchmarking on Simulated Ground Truth Data using Symsim.** A total of 20 cell populations were simulated from two modalities, each with 20 shared features and 20 distinct features, using Symsim. Matching accuracy (cell type) was compared across methods (Mario, Seurat-PCA/CCA/RPCA, Scanorama, and FastMNN).

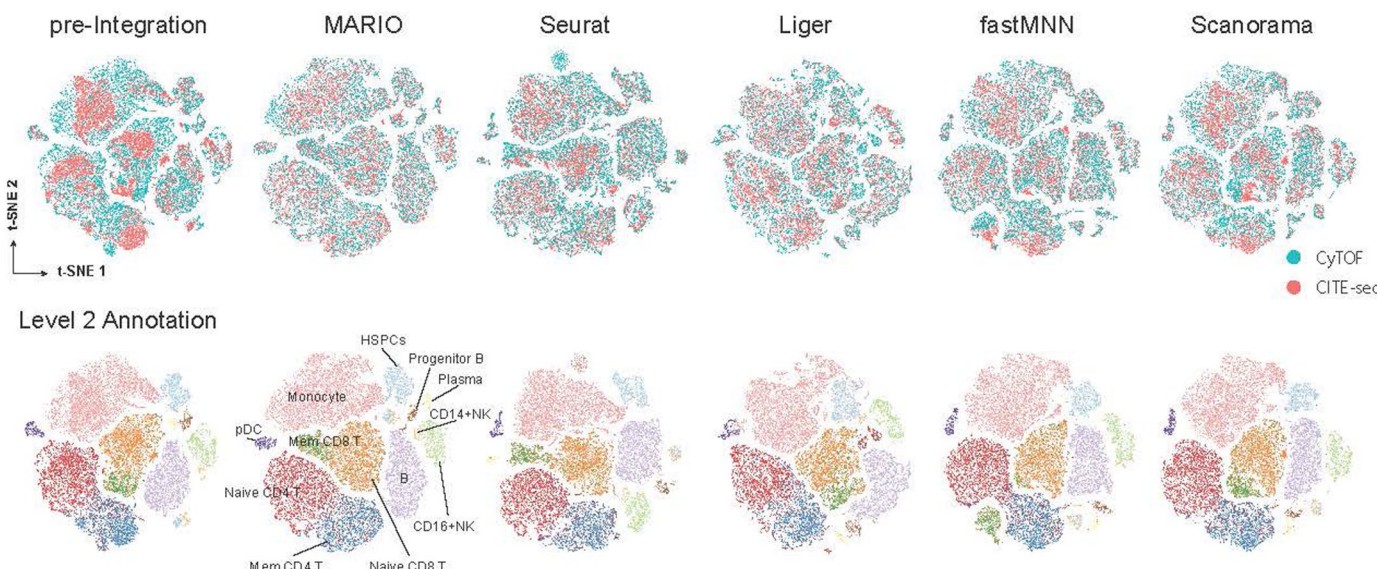

## A  Sequentially Deleting Overlapping Protein Features (Alphabetical)

**Extended Data Fig. 2 | Performance of matching and integration on bone marrow cells. (A)** Performance of matching and integration during the sequential dropping of shared protein features (by alphabetical order) **(B)** t-SNE plots visualizing pre-integration and post-integration results with the various methods. For methods other than MARIO, only shared features were used during integration. All cells are colored by modality or cell type annotation (level2).

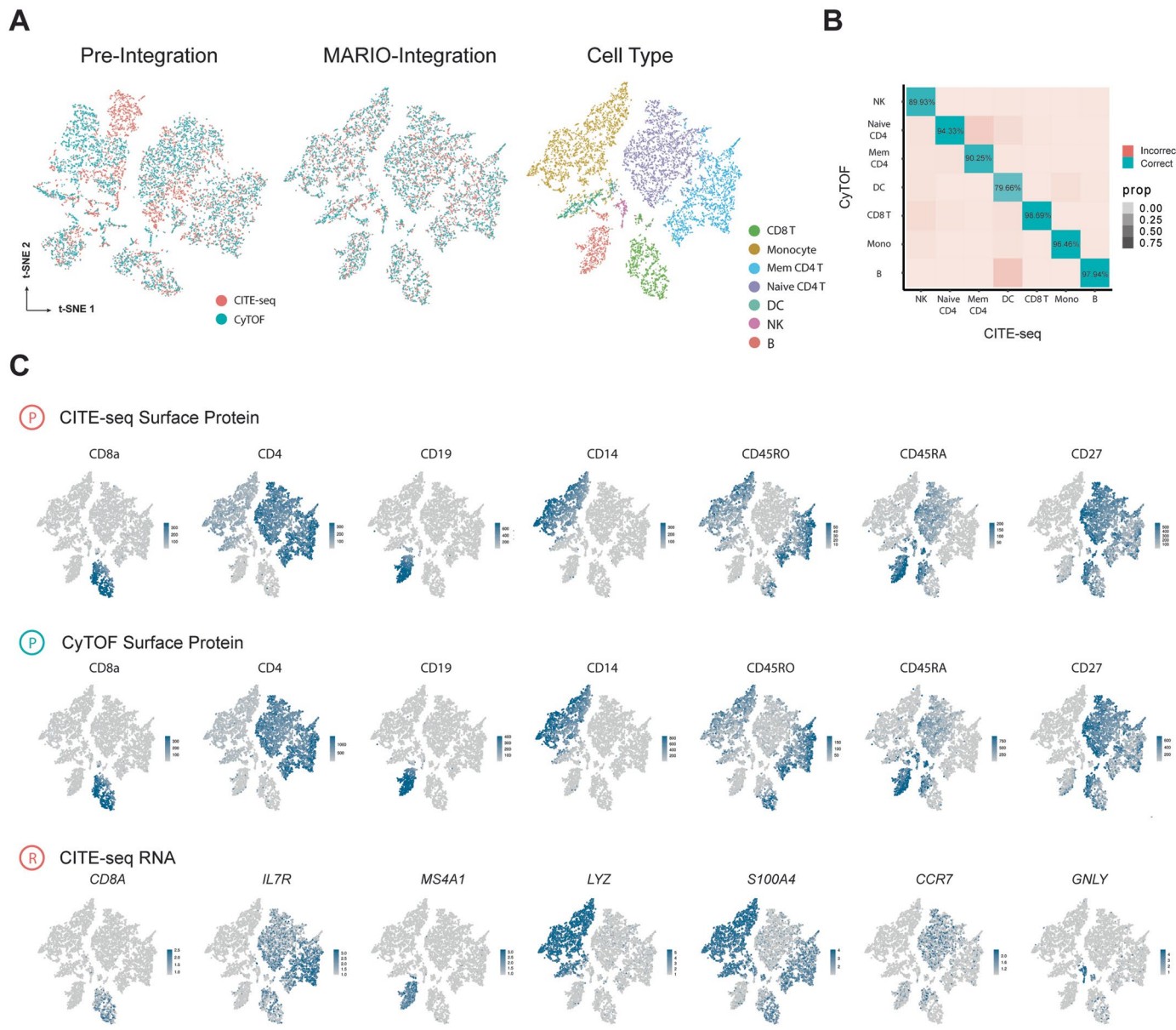

**Extended Data Fig. 3 | Matching and integration of cross-modality CyTOF and CITE-seq PBMC data.** MARIO integration of human PBMCs as measured by CyTOF and CITE-seq. **(A)** t-SNE plots of the PBMC CITE-seq and CODEX cells, pre-integration (left) and MARIO integrated (middle and right), colored by dataset of origin (left and middle) or colored by cell types (right). **(B)** Confusion matrix with MARIO cell-cell matching accuracy (balanced accuracy) across cell types. **(C)** t-SNE plots of the matched cells with protein or RNA expression levels overlaid.

## A Sequentially Deleting Overlapping Protein Features (Alphabetical)

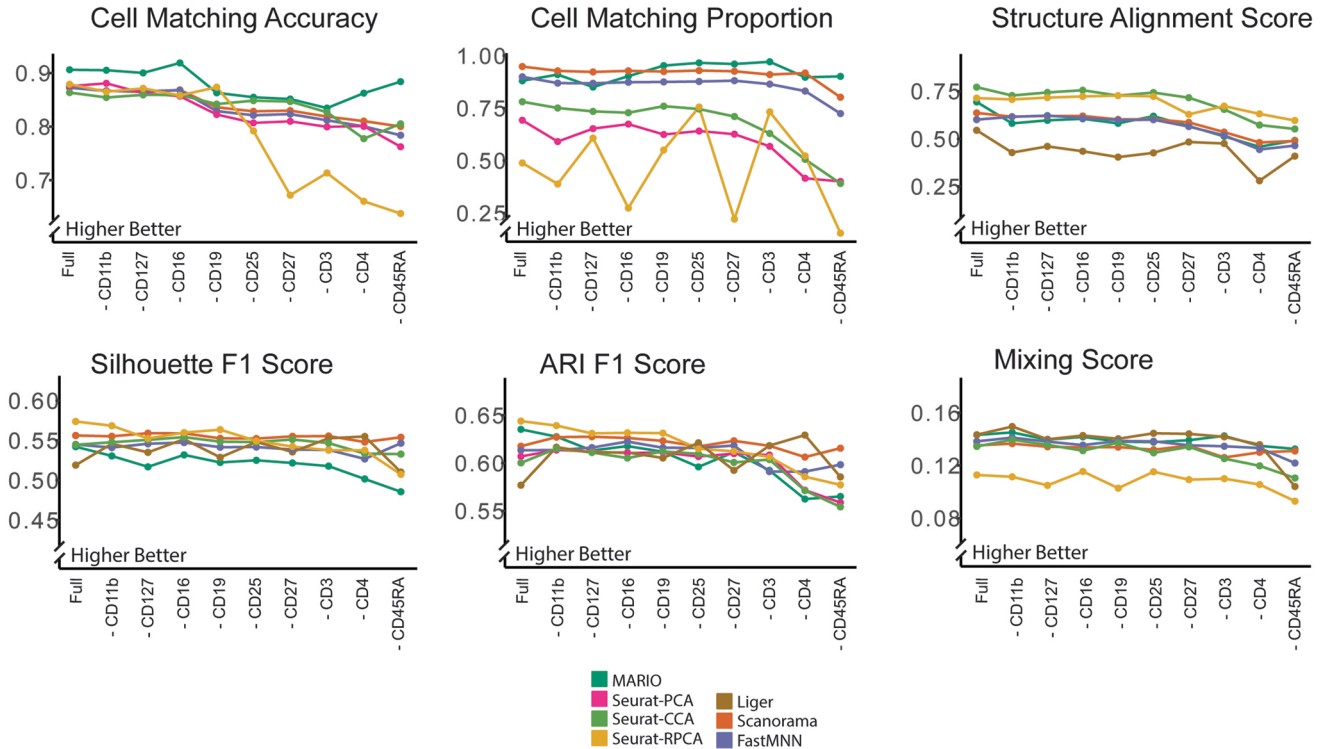

## B Simulated Contaminated Data

## C Delete Specific Cell Type

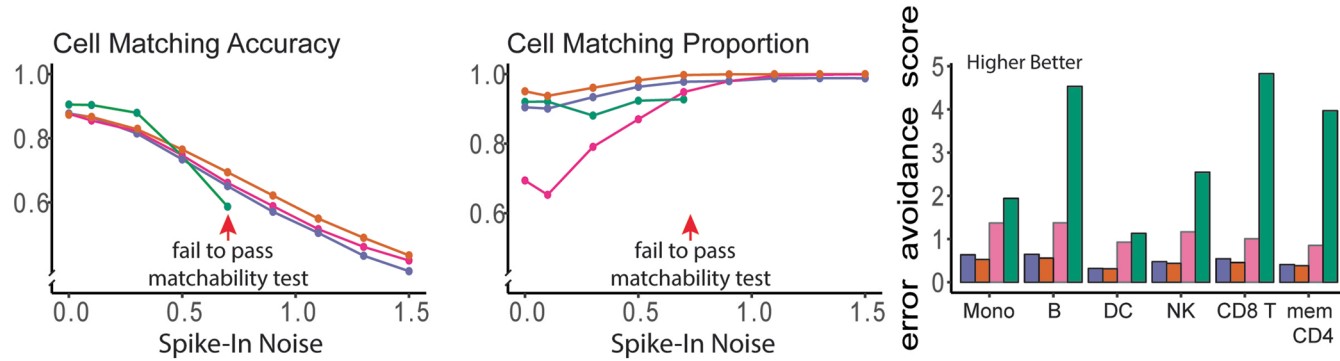

## D t-SNE Dimension Reduction Visualisation of Different Methods

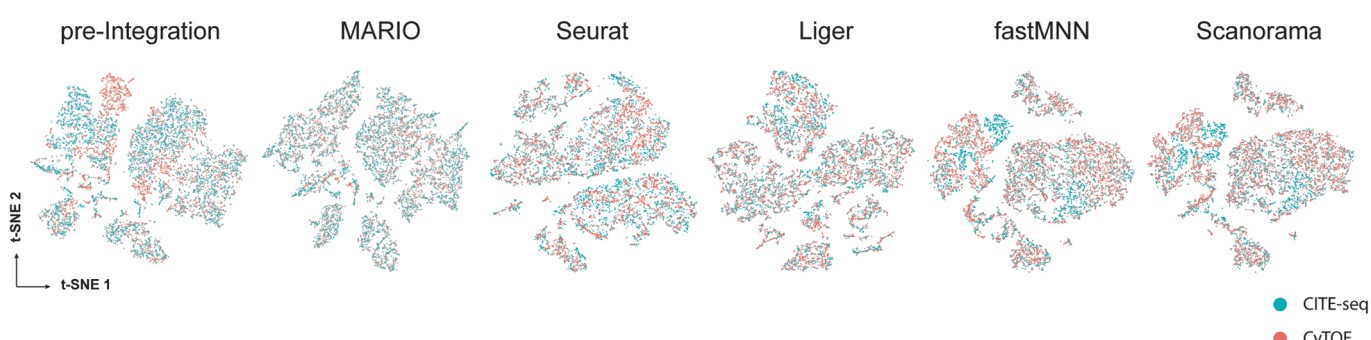

**Extended Data Fig. 4 | See next page for caption.**

**Extended Data Fig. 4 | Matching and integration of cross-modality CyTOF and CITE-seq PBMC data.** MARIO integration of human PBMCs as measured by CyTOF and CITE-seq. **(A)** Performance of matching and integration during the sequential dropping of shared protein features. **(B)** Testing algorithm stringency between different methods. Increasing amounts of random spike-in noise was added to the data, and the matching accuracy and proportion of cells matched to X were quantified. **(C)** Testing algorithm stringency among different methods. Single-cell types in Y were completely removed before matching to X. The proportion of cells belonging to the deleted cell type in matched X cells was used to calculate the erroneous avoidance score. **(D)** t-SNE plots visualizing pre-integration and post-integration results across different methods.

## Sequentially Deleting Overlapping Protein Features (Alphabetical)

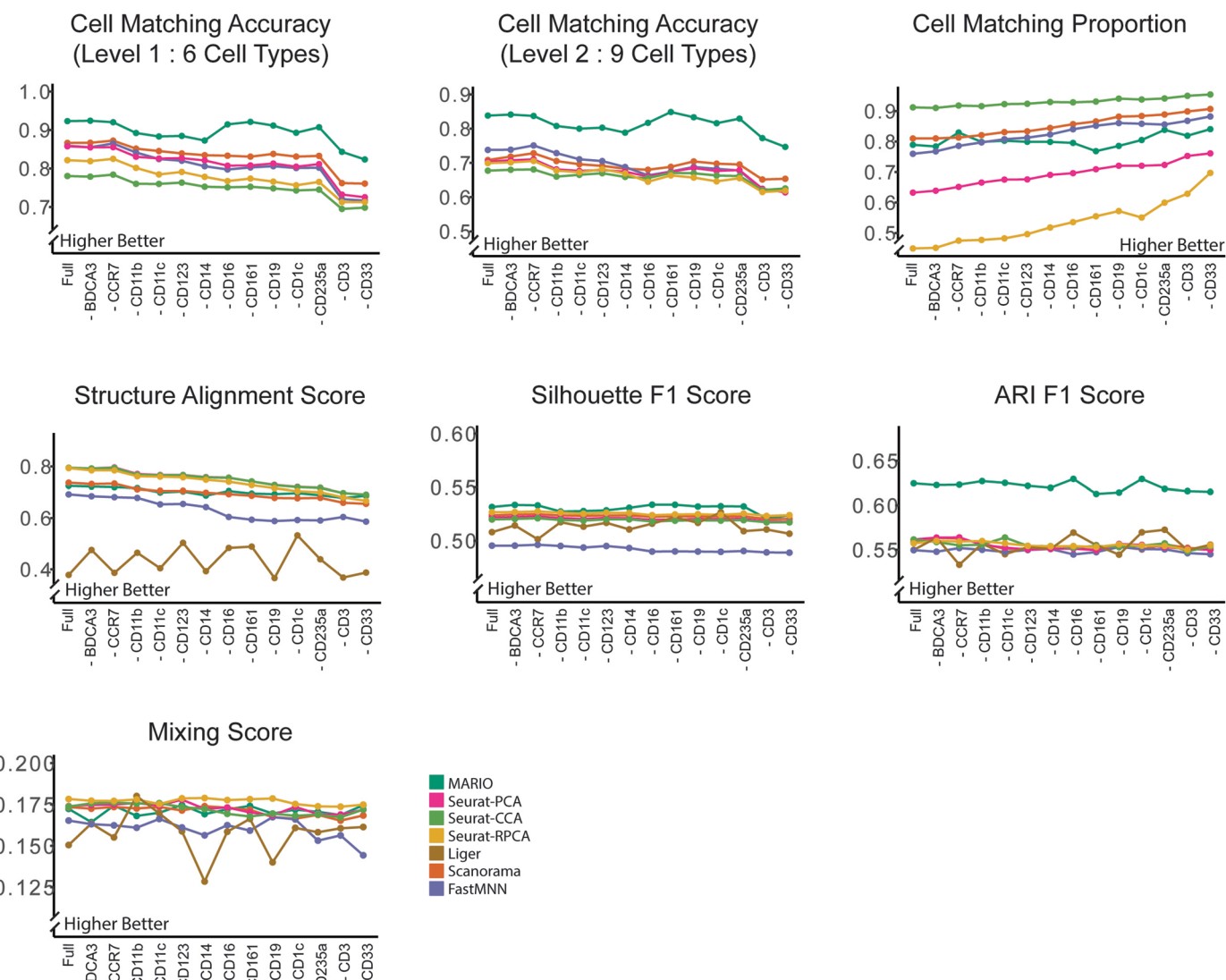

**Extended Data Fig. 5 | Performance of matching and integration on cross-species whole blood cells CyTOF data.** Performance of matching and integration during sequentially dropping of shared protein features.

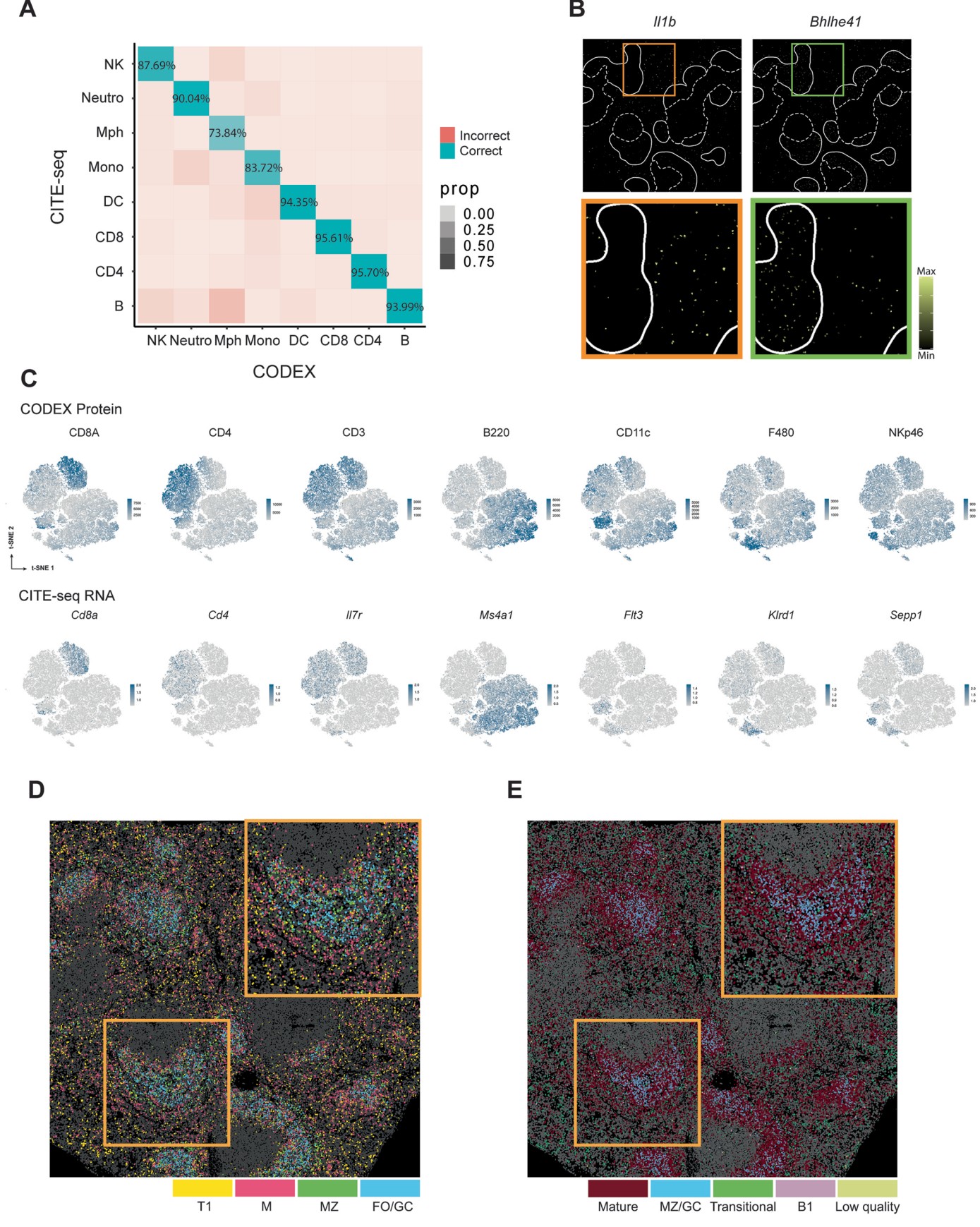

**Extended Data Fig. 6 | See next page for caption.**

**Extended Data Fig. 6 | MARIO integrative analysis of CODEX and CITE-seq for spatial multi-omics. (A)** Confusion matrix with MARIO cell-cell matching accuracy (balanced accuracy) across cell types for matched CITE-seq or CODEX cells. **(B)** A pseudo-colored murine spleen section showing the localization of transcripts (*Il1b* and *Bhlhe41*) inferred from CITE-seq. The white outline demarcates the white pulp. **(C)** t-SNE plots (calculated from CODEX protein alone) of MARIO integrated murine spleen CITE-seq and CODEX cells, overlaid with matched CODEX protein and CITE-seq RNA expression levels. **(D)** A pseudo-colored murine spleen section colored by the annotation of CODEX B cell subpopulations, via expert manual gating. **(E)** A pseudo-colored murine spleen section colored by MARIO label transferred annotation from the CITE-seq dataset annotation.

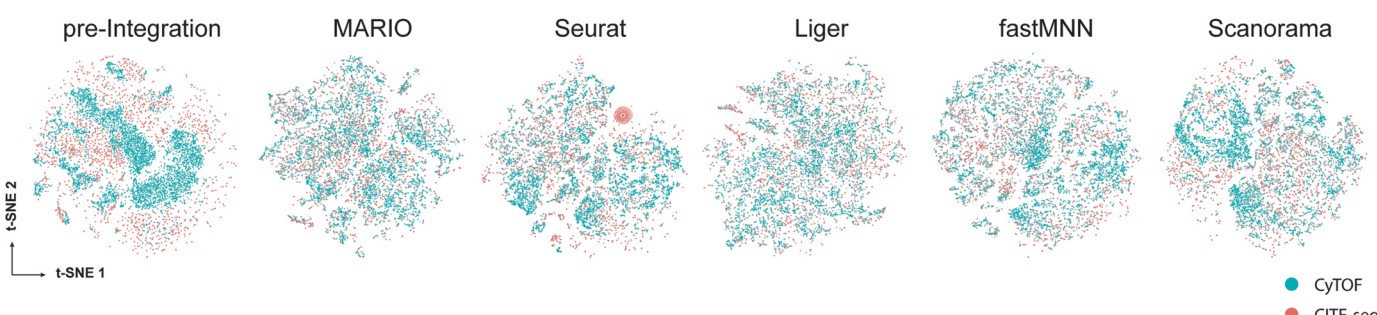

**Extended Data Fig. 7 | Performance of matching and integration on CODEX and CITE-seq murine spleen cells. (A)** Performance of matching and integration during the sequential dropping of shared protein features. **(B)** Testing algorithm stringency between different methods. Increasing amounts of random spike-in noise was added to the data, and the matching accuracy and proportion of cells matched to X were quantified. **(C)** Testing algorithm stringency among different methods. Single-cell types in Y were completely removed before matching to X. The proportion of cells belonging to the deleted cell type in matched X cells was used to calculate the erroneous avoidance score. **(D)** t-SNE plots visualizing pre-integration and post-integration results with different methods.

**A**

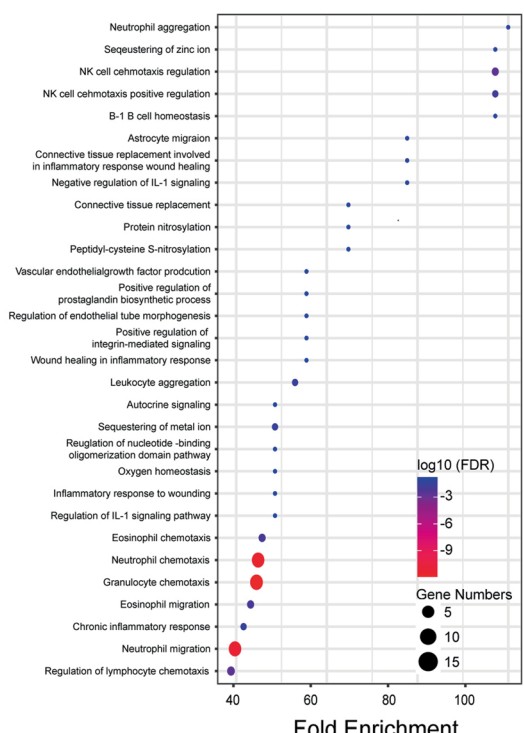
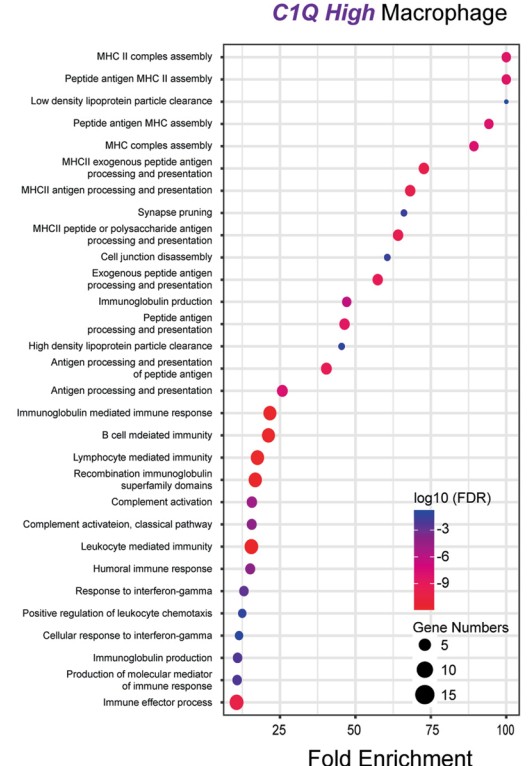

**B** Top Differentially Expressed Genes

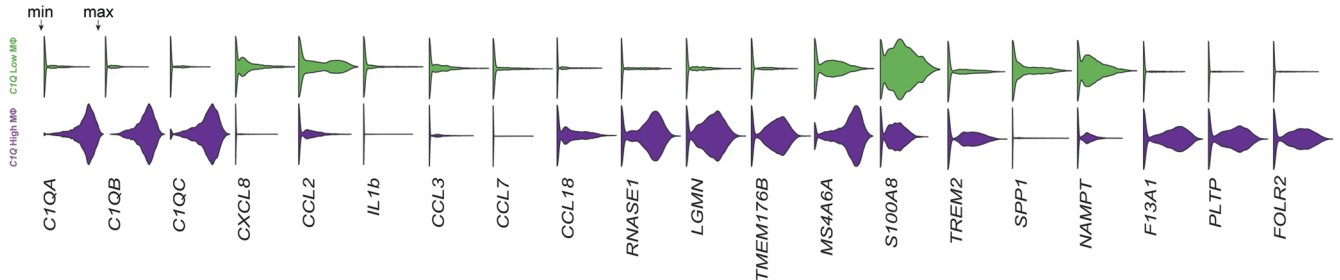

**C** Differentially Expressed ISGs

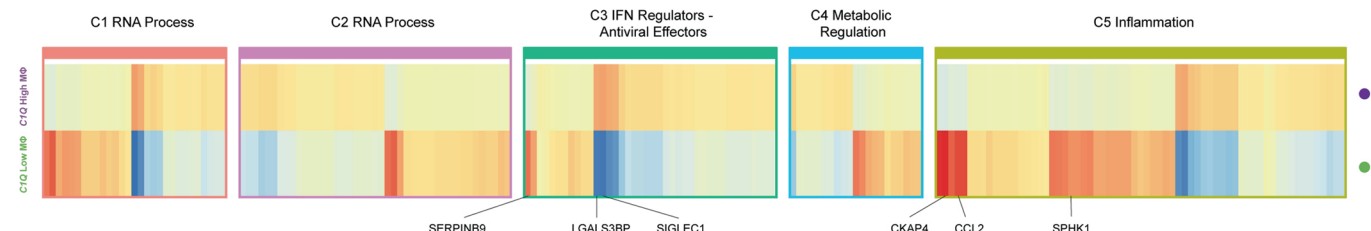

**Extended Data Fig. 8 | MARIO analysis on COVID-19 lung tissue and BALF cells. (A)** GO term analysis for transcriptional programs enriched in *C1Q* low (left) and high macrophages (right). **(B)** Violin plots of selected genes from the top 50 differentially expressed genes (p-adjust < 0.05) for *C1Q* low (green) or *C1Q* high (magenta) macrophages. **(C)** A heatmap representation of differentially expressed ISGs among *C1Q* low (up) or *C1Q* low macrophages (down). Genes are categorized into 5 previously described classes of biological pathways (see supplementary notes).

**A** Representative CODEX Cores Images

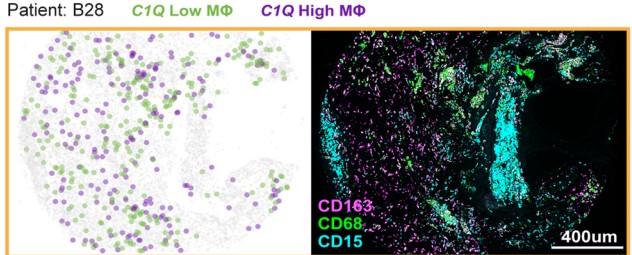

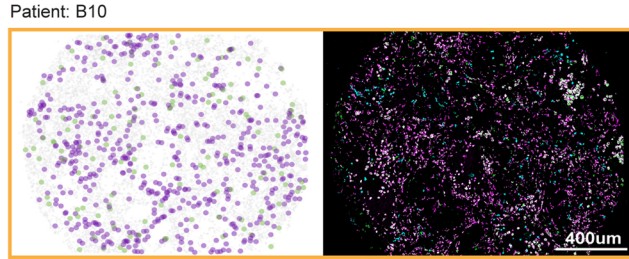

**B** Cell - Cell Interaction

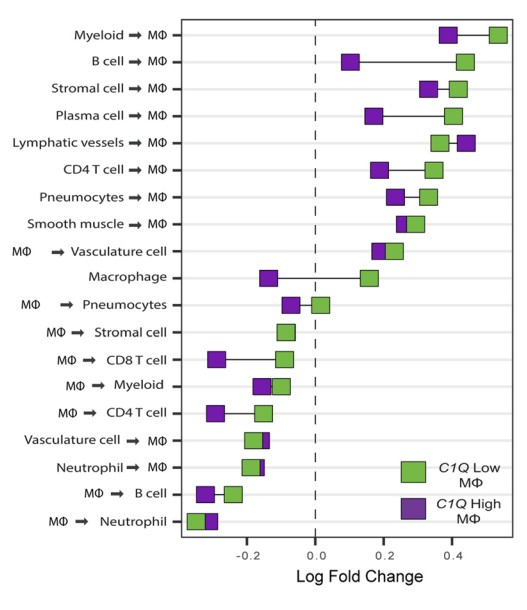

**C** Anchor Analysis

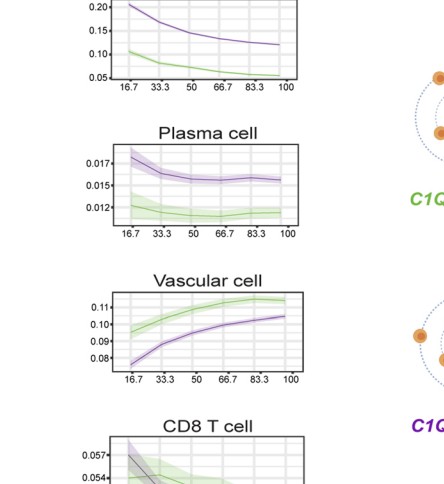

**D** Representative PANINI Cores Images

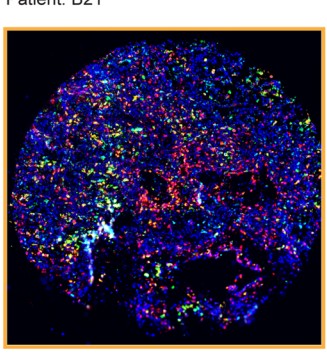

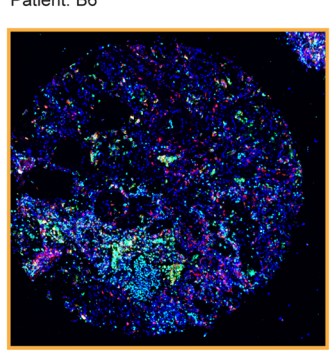

**E** PANINI - CODEX Spatial Correlation Baseline

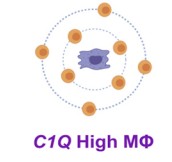

**Extended Data Fig. 9 | MARIO analysis on COVID-19 lung tissue and BALF cells. (A)** Additional representative CODEX images of COVID-19 lung tissue cores for patients with *C1Q* low (green) and high (magenta) macrophage locations. CD163, CD68, and CD15 antibody staining are shown on the right of each image. **(B)** The pairwise cell distances between *C1Q* high low (green) or (magenta) macrophages to other cell types, as an enrichment over the permutated background distribution. Only interactions that passed a statistical test (p < 0.05) for both macrophage subgroups conditions are shown. Squares that are toward the left indicate interactions that are closer than expected, and those toward the right indicate interactions that are further apart than expected. **(C)** Anchor plots of average cell type fractions around *C1Q* low (green) or *C1Q* high (magenta)

macrophages. The thick colored lines represent the means, and lighter regions around these lines depict the 95% confidence interval. The macrophages are anchored at 0 μm, and the plot ends at a 100 μm radial distance from the anchored macrophages. **(D)** Representative images of COVID-19 lung tissue cores in the PANINI validation experiment, stained with *C1QA*, CD68, CD15 and Hoechst. All cores (n = 76) were considered during quantitative anlysis. **(E)** Spatial correlation of cell density in each 10 x 10 region of the same tissue core between CODEX experiment and PANINI validation to determine the baseline correlation between the tissue sections for CODEX and PANINI (P-value and Correlation calculated via two-sided Spearman Ranked Test), lighter regions around the line depict the 95% confidence interval.

## Accuracy of Initial Matching with Different Distance Matrices

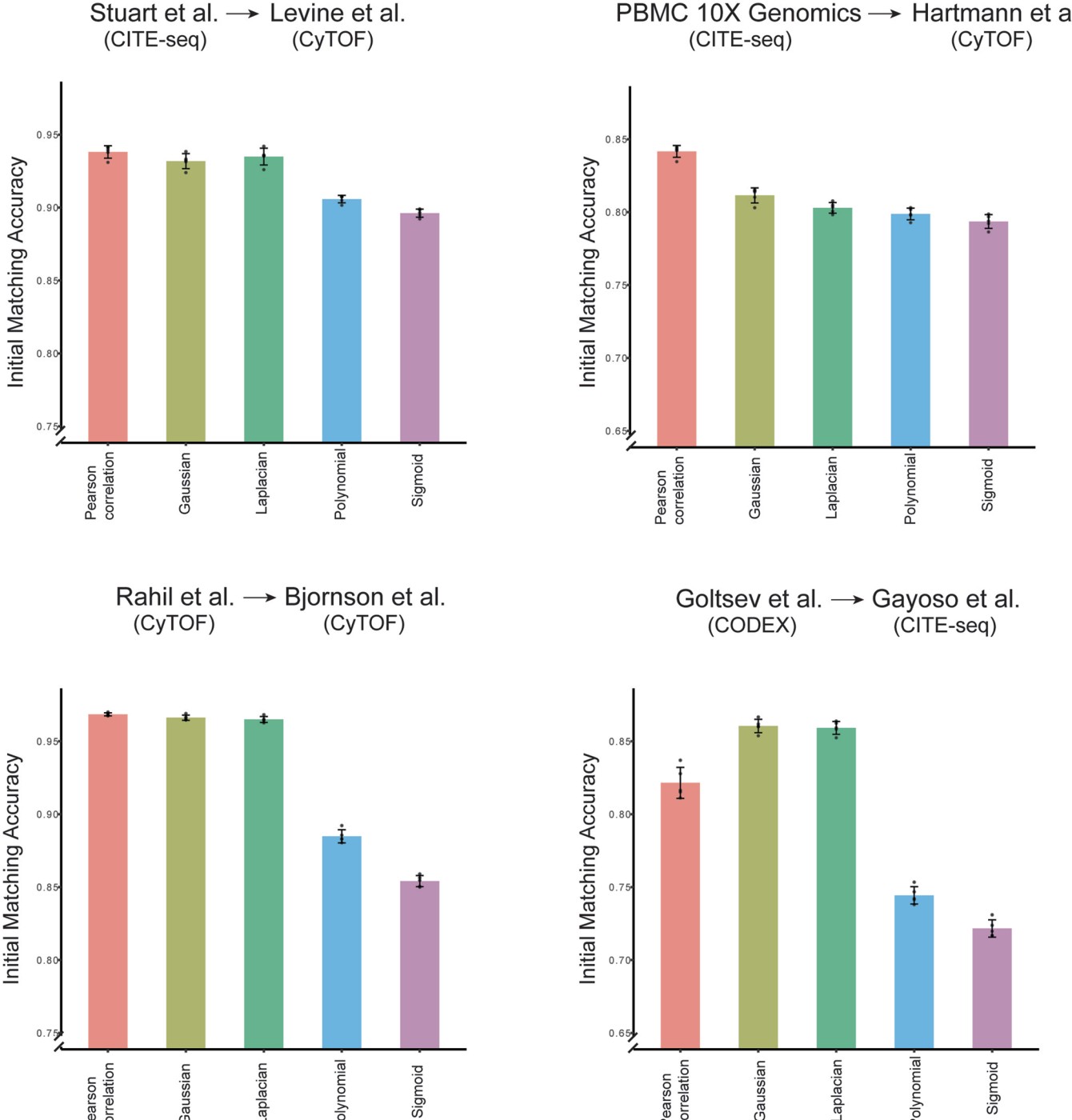

**Extended Data Fig. 10 | MARIO distance matrix construction method benchmarking.** The initial matching accuracy (mean ± sd, n = 5 batches) by MARIO using distance matrix constructed by different methods: pearson correlation, distance correlation; non-linear kernels: gaussian; laplacian; polynomial and sigmoid, on different datasets presented in the manuscript.

# nature research

# Reporting Summary

Nature Research wishes to improve the reproducibility of the work that we publish. This form provides structure for consistency and transparency in reporting. For further information on Nature Research policies, see our Editorial Policies and the Editorial Policy Checklist.

## Statistics

For all statistical analyses, confirm that the following items are present in the figure legend, table legend, main text, or Methods section.

| n/a | Confirmed | |
|---|---|---|
| ☐ | ☒ | The exact sample size (*n*) for each experimental group/condition, given as a discrete number and unit of measurement |
| ☐ | ☒ | A statement on whether measurements were taken from distinct samples or whether the same sample was measured repeatedly |
| ☐ | ☒ | The statistical test(s) used AND whether they are one- or two-sided <br> *Only common tests should be described solely by name; describe more complex techniques in the Methods section.* |
| ☒ | ☐ | A description of all covariates tested |
| ☐ | ☒ | A description of any assumptions or corrections, such as tests of normality and adjustment for multiple comparisons |
| ☐ | ☒ | A full description of the statistical parameters including central tendency (e.g. means) or other basic estimates (e.g. regression coefficient) AND variation (e.g. standard deviation) or associated estimates of uncertainty (e.g. confidence intervals) |
| ☐ | ☒ | For null hypothesis testing, the test statistic (e.g. *F*, *t*, *r*) with confidence intervals, effect sizes, degrees of freedom and *P* value noted <br> *Give P values as exact values whenever suitable.* |
| ☒ | ☐ | For Bayesian analysis, information on the choice of priors and Markov chain Monte Carlo settings |
| ☒ | ☐ | For hierarchical and complex designs, identification of the appropriate level for tests and full reporting of outcomes |
| ☐ | ☒ | Estimates of effect sizes (e.g. Cohen's *d*, Pearson's *r*), indicating how they were calculated |

*Our web collection on statistics for biologists contains articles on many of the points above.*

## Software and code

Policy information about availability of computer code

| | |
|---|---|
| Data collection | All code used to collect and pre-process the data used in this study is deposited to : https://github.com/shuxiaoc/mario-py <br> For the PANINI experimental validation of COVID tissues presented in Figure 5, images were collected via Keyence BZ series imaging software (2018 Ver), and stitching was performed with BZ-X Analyzer (2018 Ver). |
| Data analysis | All code used in data analysis in this study is deposited to : https://github.com/shuxiaoc/mario-py, code related to the this round of revision is submitted. <br> R packages used: <br> Seurat 3.2.3 <br> SeuratData <br> Seurat 4.1.1 (for rpca related) <br> rliger 1.0.0 <br> batchelor 1.2.4 <br> flowCore 1.52.1 <br> Rtsne 0.16 <br> dplyr 1.0.7 <br> data.table 1.14.2 <br> ggplot2 3.3.5 <br> plyr 1.8.7 <br> stats 3.6.3 <br> factoextra 1.0.7 <br> RColorBrewer 1.1-3 <br> gplots 3.1.3 <br> doSNOW 1.0.19 <br> deldir 1.0-6 |

```
foreach 1.5.2
tidyverse 1.3.1
mltools 0.3.5
parallel 1.32.1
patchwork 1.1.1
Pigengene 1.12.0
matrixStats 0.62.0
ggrepel 0.9.1
reshape2 1.4.4
Symsim 0.0.0.9000
ape 5.6-2

python package used:
scanorama 1.7.2
anndata 0.8.0
scanpy 1.9.1
spaotsc 0.2
numpy 1.20.1
pandas 1.2.1
scipy 1.6.2
scikit-learn 0.23.2
matplotlib 3.3.4
seaborn 0.11.1
```

For manuscripts utilizing custom algorithms or software that are central to the research but not yet described in published literature, software must be made available to editors and reviewers. We strongly encourage code deposition in a community repository (e.g. GitHub). See the Nature Research guidelines for submitting code & software for further information.

# Data

Policy information about availability of data

All manuscripts must include a data availability statement. This statement should provide the following information, where applicable:
- Accession codes, unique identifiers, or web links for publicly available datasets
- A list of figures that have associated raw data
- A description of any restrictions on data availability

Publicly available datasets:

Levine et al. Human BMC CYTOF: https://github.com/lmweber/benchmark-data-Levine-32-dim
Stuart et al. Human BMC CITE-seq (From R package SeuratData, "bmcite"): https://satijalab.org/seurat/articles/weighted_nearest_neighbor_analysis.html
Zainab et al. Human H1N1 challenged whole blood CYTOF: flow repository FR-FCM-Z2NZ
Bjornson et al. Human and non-human-primate whole blood CYTOF: flow repository FRFCM-Z2ZY
Goltsev et al. Murine Spleen CODEX: https://data.mendeley.com/datasets/zjnpwh8m5b/1 (Raw images per request from Nolan Lab)
Gayoso et al. Murine Spleen CITE-seq: https://github.com/YosefLab/totalVI_reproducibility/tree/master/data
COVID-19 Cell Atlas. COVID-19 patient BALF CITE-seq (VIB/Ghent): https://www.covid19cellatlas.org/index.patient.html
Hartmann et al. Human PBMC CyTOF: flow repository FR-FCM-Z249 :HD06_run1
10x Genomics. Human PBMC CITE-seq: https://support.10xgenomics.com/single-cell-gene-expression/datasets/3.0.2/5k_pbmc_protein_v3?

All data mentioned above are also available at: https://github.com/shuxiaoc/mario-py

Data generated in this study:

Goltsev et al. COVID-19 patient Lung tissue CODEX dataset: Subset of data used in this study is deposit at: https://github.com/shuxiaoc/mario-py. Separate manuscript in preparation, full dataset will release after manuscript submission.

Simulated data generated with Symsim (Zhang et al. 2019) for ground truth analysis: method and parameters described in the Material & Methods section. Code used submitted.

Figures with associated data list:
Deposited at: https://github.com/shuxiaoc/mario-py/blob/main/Manuscript_Archive_Code/data/readme.md

# Field-specific reporting

Please select the one below that is the best fit for your research. If you are not sure, read the appropriate sections before making your selection.

☒ Life sciences    ☐ Behavioural & social sciences    ☐ Ecological, evolutionary & environmental sciences

For a reference copy of the document with all sections, see nature.com/documents/nr-reporting-summary-flat.pdf

# Life sciences study design

All studies must disclose on these points even when the disclosure is negative.

| | |
|---|---|
| Sample size | For computational benchmarking of MARIO and other methods, we selected sufficient number of cells that is manageable in terms of time and computational power for all methods, generally ranging from 5k cells to 80k cells, dependent on the scenario. We the cell numbers are sufficient here since it is the same size or larger than most of the datasets from the same modality. Additionally, the occurrence of lower frequent cell types are still sufficiently represented during testing. Further increase of the cell number for testing will not change the conclusion presented in the manuscript.<br><br>For experiments related to COVID-19 CODEX data, all tissue cores acquired from the collection site was used in this analysis. |
| Data exclusions | No sample was excluded in this study. |
| Replication | A total of 13 different datasets (6 matching & integration cases) were described in this manuscript, confirming the effectiveness of the method. |
| Randomization | Randomization was performed if cells were subsampled from the original dataset. For COVID-19 related experiments presented in the manuscript, no randomization was performed we do not have multiple experimental groups or conditions. |
| Blinding | All data presented in the manuscript was analyzed with standardized quantitative algorithms and no qualitative measurements would be affected by observer bias.<br>When conducting and performing analysis of the COVID-19 ISH validation experiments, the researcher was blinded with the tissue core information intended for validation. |

# Reporting for specific materials, systems and methods

We require information from authors about some types of materials, experimental systems and methods used in many studies. Here, indicate whether each material, system or method listed is relevant to your study. If you are not sure if a list item applies to your research, read the appropriate section before selecting a response.

## Materials & experimental systems

| n/a | Involved in the study |
|---|---|
| ☐ | ☒ Antibodies |
| ☒ | ☐ Eukaryotic cell lines |
| ☒ | ☐ Palaeontology and archaeology |
| ☒ | ☐ Animals and other organisms |
| ☐ | ☒ Human research participants |
| ☒ | ☐ Clinical data |
| ☒ | ☐ Dual use research of concern |

## Methods

| n/a | Involved in the study |
|---|---|
| ☒ | ☐ ChIP-seq |
| ☒ | ☐ Flow cytometry |
| ☒ | ☐ MRI-based neuroimaging |

# Antibodies

| | |
|---|---|
| Antibodies used | The antibody information is described in the Material and Methods section of the manuscript.<br>primary<br>anti-CD15 (1:100 dilution, clone: MC480, Biolegend, 125602)<br>anti-CD68 (1:100 dilution, clone: D4B9C, Cell Signaling Technology, 76437T)<br><br>secondary<br>Anti- Mouse-Cy7 (1:250, Biolegend, 405315)<br>Anti-Rabbit- Alexa647 (1:250, Thermo Fisher Scientific, A-21245) |
| Validation | All primary and secondary antibody used in this study has been validated by the manufacturer.<br><br>CD15:<br>Verified reactivity in:<br>Human, Mouse;<br>Verified application:<br>FC - Quality tested<br>IP, WB, IHC-F, IHC-P - Reported in the literature, not verified in house<br>Validation studies listed by vendor:<br>Solter D and Knowles BB. 1978. Proc. Natl. Acad. Sci. USA. 75:5565. (IHC, IP, WB)<br>Tempest N, et al. 2018. Hum Reprod. 6:e00392. PubMed<br>Wu W, et al. 2017. Sci Rep. 7:44481. PubMed<br><br>CD68:<br>Verified reactivity in: |

Human;
Verified application:
IHC, IF, Flow
Validation studies listed by vendor (first two papers):
Wang, Qirui, et al. "Vascular niche IL-6 induces alternative macrophage activation in glioblastoma through HIF-2α." Nature communications 9.1 (2018): 1-15.
Mullen, Peter J., et al. "SARS-CoV-2 infection rewires host cell metabolism and is potentially susceptible to mTORC1 inhibition." Nature communications 12.1 (2021): 1-10.

# Human research participants

Policy information about <u>studies involving human research participants</u>

| | |
|---|---|
| Population characteristics | *Describe the covariate-relevant population characteristics of the human research participants (e.g. age, gender, genotypic information, past and current diagnosis and treatment categories). If you filled out the behavioural & social sciences study design questions and have nothing to add here, write "See above."* |
| Recruitment | *Describe how participants were recruited. Outline any potential self-selection bias or other biases that may be present and how these are likely to impact results.* |
| Ethics oversight | *Identify the organization(s) that approved the study protocol.* |

Note that full information on the approval of the study protocol must also be provided in the manuscript.

