## [Peer Review File. · Nature Methods]

Peer Review Information

Manuscript Title: Robust Single-cell Matching and Multi-modal Analysis Using Shared and Distinct Features

Corresponding author name(s): Zongming Ma, Garry P Nolan, Sizun Jiang

Editorial Notes: n/a

Reviewer Comments & Decisions:

Decision Letter, initial version:

Dear Professor Jiang,

Your Article entitled "Robust Single-cell Matching and Multi-modal Analysis Using Shared and Distinct Features Reveals Orchestrated Immune Responses" has now been seen by 3 reviewers, whose comments are attached. While they find your work of potential interest and impact, they have raised serious concerns which in our view are sufficiently important that they preclude publication of the work in Nature Methods, at least in its present form.

As you will see, the reviewers have several concerns about the technical validity of MARIO as well as the lack of benchmarking against existing tools in the paper. Should further experimental data allow you to fully address these criticisms we would be willing to look at a revised manuscript (unless, of course, something similar has by then been accepted at Nature Methods or appeared elsewhere). This includes submission or publication of a portion of this work somewhere else. We hope you understand that until we have read the revised paper in its entirety we cannot promise that it will be sent back for peer-review.

If you are interested in revising this manuscript for submission to Nature Methods in the future, please contact me to discuss your appeal before making any revisions. Otherwise, we hope that you find the reviewers' comments helpful when preparing your paper for submission elsewhere.

Sincerely,

Madhura

Madhura Mukhopadhyay, PhD
Associate Editor
Nature Methods

Reviewers' Comments:

Reviewer #1:

Remarks to the Author:

Zhu et al. report a method called MARIO (Matching with pARtial Overlap) to integrate single-cell multi-modal data sets that share only a limited number of features and can span different species. The approach is novel and given the growing number of data sets of diverse modalities, algorithms that enable the integration of such data are of high value. The authors apply MARIO to integrate CyTOF, CITEseq, and CODEX data sets in several different scenarios spanning healthy unperturbed cells, stimulated cells, and specimens from COVID patients, as well as across species and show several metrics to compare MARIO to previously published methods. However, more in-depth information is needed for the reader to properly assess whether MARIO truly outperforms previously published methods. Furthermore, benchmarking this method on ground-truth data sets would be highly beneficial to distinguish technical differences introduced by the different modalities from biological differences. Lastly, the examples of unknown biology being discovered with the help of MARIO could be presented in a more convincing way and it is unclear what concrete finding the authors refer to when stating that MARIO 'reveals orchestrated immune responses' in the title and abstract of their manuscript. Below are suggestions that would greatly improve the value of the manuscript.

Major remarks

- The authors only call high-level cell types in their data sets. In many instances the data sets already look sufficiently aligned pre-integration to call these very coarse cell types (e.g. Fig2A) and extract the biology exemplified (e.g. Figure 2 and 3). A finer level of clustering would be needed to assess the performance of the MARIO integration and yield benefit from transferring information across data sets. Similarly, the tSNEs comparing MARIO to other methods are not very informative without knowing the sub-cluster annotation of the cells. Of note, being able to infer highly granular transcriptional information from a limited protein panel would be conceptually exciting since it would suggest that subtle changes in protein expression might capture much more information than appreciated. However, this cannot be concluded from the coarse cell types shown.
- The test to drop features and assess the performance of MARIO compared to previous methods using multiple metrics is great. However, the same metric seems to perform quite variable across data sets, e.g. ARI F1 score shows major improvements in some, but not other data sets. It is therefore unclear

how reliable this benchmarking approach is. Are the results more consistent if features are dropped in a systematic way assessing all possible combinations instead of the alphabetic one-by-one drop?

- While it is nice to see MARIO applied to various settings, it would be highly beneficial for 1-2 data sets to provide more in-depth, and step-by-step information about what MARIO is doing rather than just the high-level tSNEs and metrics shown in the paper. E.g. what components are extracted from the data set using singular value decomposition? Given that the number of joint features is small, why is it beneficial to pick components and not use all components? How does the performance of MARIO change when changing such parameters? How does the integration actually look like when dropping features (instead of just showing the metrics)?

- Similarly, it would be important to evaluate the method on a real ground-truth data set where the same samples is triaged for measurements of different modalities (e.g. CyTOF and CITEseq. The data set should be designed to have some clearly distinct cell states within the same cell type, e.g. by stimulating myeloid cells with different cytokines, and perfectly matching highly granular annotations. In this ground-truth data set, how does MARIO perform if specific cell states are dropped from one modality, are differences in highly granular cell states preserved or does MARIO over-integrate the data sets? How does MARIO perform when features are dropped, noise is added etc.? Of note, figure S7 showing IL4 versus influenza stimulated blood cells suggests that MARIO might over-integrate data sets since clear differences between IL4 and influenza treated samples would be expected. Furthermore, pSTAT1 would be expected to be downstream of IFN (triggered by influenza), not downstream IL4, which is opposite of what is shown here.

Other remarks

- Error avoidance score (page 20): If a cell type from data set X does not exist in the data set Y, you don't want any of those cells in X matched to something in Y, so a should ideally be 0 and the score low, not high. Please clarify.

- Figure 2: CD34 and CD38 are listed as shared features in the text and used in the dropping of features approach in Figure 2C, but show up as CyTOF-specific features in Figure 2F. Please clarify.

- Figure S3A does not show B cells in the legend.

- The text (line 450, page 6) refers to Figure 3E and F which do not exist in Figure 3.

- Figure S5: It is unclear to me how the integration helps here. pSTAT1 and Ki-67 levels could just be compared within a data set between stimulated and unstimulated cells to determine the impact of the stimulation in the various settings. I would also argue that the effect is still visible in Figure S5D pre-integration.

- CiteSeq / CODEX integration (Figure 4, S9, S10): Since there are a lot more CODEX cells than CITEseq cells and therefore a pairwise matching would leave many CODEX cells without partner, how is the single cell information transferred to the spatial location? Is for each CODEX cell the closest matching CITEseq cell used? In this case, what happens to cell types/states that are only present in the CODEX data?

- The gating on the CODEX data in Figure S9D does not look super convincing. In order to show the power of integrating data sets, it might be better to instead use granular clustering information based on the scRNAseq data and locate those cells in the tissue using the integrated CODEX data.
- Shouldn't the MARIO-integrated tSNEs in Figures 4A and S9C be the same?
- Why is Figure S10C not shown for B cells?
- Figure S10D: There is a funny round red cluster in the Seurat integration tSNE. Is there something special about these cells?
- COVID data set and C1Q myeloid cell finding: Based on the CODEX data alone, would you have been able to distinguish C1Q low versus high cells? If not, can you show data on this to convince the reader that the integration of the data sets is indeed beneficial here?
- C1Q low versus high cells could represent monocytes versus more mature myeloid cells. Do the C1Q low cells express monocyte markers? Which markers in Figure 5B are indicative of M1? Only CD36 looks higher in C1Q low cells in the CODEX data, which is thought to be higher in M2 rather than M1.
- Looking at the axes of the plots, I would argue that the percentages in PANINI versus MARIO are actually quite different (5I and J), even though indeed nicely correlated.
- The correlation of C1Q myeloid cells to patient BMI feels a bit out of place.
- Figure S11D is missing the legend, but if blue is low and red is high, C1Q high cells are the ones with higher levels for CCL2, while MX1 does not seem to be different between the two populations of cells. This does not match the results stated in lines 649-652.
- S11F: If this figure shows pairwise distances, what is the difference between Myeloid => Mq and Mq => Myeloid? Lymphatic vessels are the only feature that seems to clearly be closer to C1Q low than C1Q high. Are there a lot more C1Q high than low myeloid cells in the tissue and is this part of the explanation regarding the shorter distances?

Reviewer #2:

Remarks to the Author:

This paper proposes a matching algorithm for single-cell multiomics data. The new method takes into account both shared and distinct features, and contains a filtering step to avoid sub-optimal matching. While the paper is generally well written and easy to read, I have some major and minor concerns as listed below.

Major concerns

1. The proposed method MARIO is very similar to Seurat v4s WNN method (Hao et al., 2021). Firstly, the cell matching procedure seems to be the 1-NN algorithm with the "cross-data distance matrix" as distance metric. The joint regularized filtering is similar to anchor filtering. So my first comment is that, how does this algorithm perform compared to Seurat's CCA (proposed in Seurat v3 (Stuart et al., 2019), and also available in package v4) or RPCA approach? RPCA can be found at

(https://satijalab.org/seurat/articles/integration_large_datasets.html) In other words, the authors only compared it with mutual nearest neighbor (mNN) methods but don't compare with CCA methods.

2. This is related to the first concern. The cross data distance matrix is generated by computing the Pearson correlation. This is much weaker compared to the distance matrix proposed in Hao et al. (2021) and can only capture linear correlation. There is a vast literature on independence measurements, such as HSIC, distance correlation, etc, that can capture both linear and non-linear associations. If the authors find their approach is inferior to Seurat V4, I suggest changing the Pearson correlation to something more advanced.

3. There are many results regarding the metric "matching accuracy", but there is no clear description on how to compute it for benchmarking methods. Some methods like Seurat's WNN may not have pairs for every cell. Furthermore, when different methods have different numbers of cell pairs or anchors, is it still reasonable to use "matching accuracy" as the metric? For example, if we filter out more "bad" anchors for Seurat's WNN method, would the "matching accuracy" of Seurat get improved? The authors need to make it clear and justify the use of evaluation metric.

Minor concerns:

1. UMAP plot should also be shown since it is more accurate compared to t-SNE.
2. Figure 3: Typo in caption; numbers of color bar in 3C are illegible.

REFERENCE

Hao, Y., Hao, S., Andersen-Nissen, E., Mauck III, W.M., Zheng, S., Butler, A., Lee, M.J., Wilk, A.J., Darby, C., Zager, M., et al. (2021). "Integrated analysis of multimodal single-cell data." *Cell*, 184(13), 3573–3587.

Stuart, T., Butler, A., Hoffman, P., Hafemeister, C., Papalexi, E., Mauck III, W.M., Hao, Y., Stoeckius, M., Smibert, P., and Satija, R. (2019). "Comprehensive integration of single-cell data." *Cell*, 177(7), 1888–1902.

Reviewer #3:

Remarks to the Author:

Zhu et al. proposed a new method, called MARIO, for integrating multi-modal single-cell proteomics data. The main idea is to utilize both shared features as well as non-shared features for matching cells. The authors demonstrated the efficacy of MARIO on several datasets and highlighted some interesting discoveries from integrating CODEX and CITE-seq from COVID-19 patients.

Overall, I found the paper interesting and the method potentially useful. The presentation in general is clear, although it can be improved. The methods are described in sufficient detail and software is deposited in github.

Regarding novelty & significant: Although none of the components of the proposed methods is novel, putting them together for proteomics single-cell data integration appears to be new. However, the proposed method is primarily based on heuristics. Although this is not necessary negative, the method itself does not provide significant conceptual advances.

I have some major concerns listed as follows:

1. The authors claim MARIO is “globally optimized”. However, in the paper, MARIO consists of several separate, sequential steps: a) initial matching with shared features; b) refined matching based on CCA of cells with aligned X & Y features from initial matching; and c) interpolating between initial and refined matchings. I don’t see a principled objective function that MARIO tries to optimize. Instead, it consists of several heuristic steps. In step c), the method aims to find the best convex combination in terms of the mean of the top k sample canonical correlations. The authors need to provide a justification on several issues:

a) Is the mean of the top k sample canonical correlations viewed as your global objective? If so, why is a good metric for integrating single-cell data?

b). Why consider only convex combinations? Is the resolution derived from interpolation globally optimal?

c). What’s the rationale for setting k=10? Are the results sensitive to the choice of k?

2. Methods on genomic data integration are not thoroughly reviewed. Methods based on mNN represent only one integration method, by no means the only one. In fact, existing work has demonstrated that mNN is suboptimal. It is commonly used primarily because of its simplicity. For a more systematic comparison, methods based on deep learning models, optimal transport, and others, should be reviewed and compared. Although these methods may not be proposed specifically for integrating proteomics data, with minor modifications, they can. Some of the references in this regard include:

Lin, Y., Wu, TY., Wan, S. et al. scJoint integrates atlas-scale single-cell RNA-seq and ATAC-seq data with transfer learning. *Nat Biotechnol* 40, 703–710 (2022). <https://doi.org/10.1038/s41587-021-01161-6>
Cang, Z., Nie, Q. Inferring spatial and signaling relationships between cells from single cell transcriptomic data. *Nat Commun* 11, 2084 (2020). <https://doi.org/10.1038/s41467-020-15968-5>

Gayoso, A., Steier, Z., Lopez, R. et al. Joint probabilistic modeling of single-cell multi-omic data with totalVI. *Nat Methods* 18, 272–282 (2021). <https://doi.org/10.1038/s41592-020-01050-x>

3. Since MARIO consists of multiple steps with multiple hyperparameters to tune for each step, there is a question of how robust MARIO is? How sensitive is MARIO wrt these parameters. I suggest authors include experiments of running MARIO on integrating sc-RNA-seq and spatial transcriptomics data. Several methods have been proposed for this benchmark. It would be interesting to see how MARIO performs compared to the state-of-the-art methods. This also serves the purpose of testing the robustness of MARIO by gradually removing shared features.

4. I also have serious concern regarding the assumption of injective mapping used in MARIO. It is clear that the cross-modal mapping is neither injective nor surjective due to the different ways how the samples are assayed. In addition, the distribution of cells across modality may be different. In this regard, the optimal transport approach is much better in handling such issues. The authors should discuss these limitations and show how it may impact the results across different samples.

5. Another issue is the linearity assumption used throughout the paper. Even for shared features, simple Pearson correlation may not be adequate to capture the similarity of cells since data are assayed with different technologies. CCA is similarly impacted by this issue. How to address this issue?

Minor comments:

1. Writing style: no need to repeat results of tables in the main text. It is fine to summarize key findings, but not necessary to restate what's already presented in the tables, making the text harder to read.

2. The Methods section needs to be cleaned up. Focus on describing new components of your methods. No need to describe in detail some of the standard methods such as PCA, CCA, which can be found in standard textbooks. Also I don't think it's a good idea to mix software with methods. Software can be described separately in the user manual. The methods should be dedicated to explain the method itself.

3. "The signals within these limited shared features alone are typically not sufficient to produce high-quality and interpretable pairwise cell matching results." To justify this statement, it would be interesting to compare matching results based on shared features and non-shared features separately.

4. The results often show comparable results when compared to Seurat, which also adopts CCA for matching. The author should discuss how CCA is applied differently between Seurat and MARIO.

In the introduction, the authors state that "Unfortunately, application of this approach to single cell...magnitudes smaller than those in single-cell sequencing datasets." Can authors elaborate on this and perhaps show some results to support the statement?

5. As shown in FigS1, compared with other MNN based methods, the overall scores do not consistently surpass other competitors except ARI F1 score. The same phenomenon is observed in FigS4A, the authors need to give some explanation on the inconsistent results.

The pipeline drawn in the Figure 1B is a little bit vague to understand, authors need to give more technical details on the matching and testing procedures.

6. The matching accuracy is evaluated at the cell type level, which depends on the quality of data annotation. Authors should discuss this limitation and discuss how to evaluate the methods on higher resolutions.

Author Rebuttal, first revision:

Updated version Reviewers' Comments:

Reviewer #1:

Remarks to the Author:

Zhu et al. report a method called MARIO (Matching with pARtial Overlap) to integrate single-cell multi-modal data sets that share only a limited number of features and can span different species. The approach is novel and given the growing number of data sets of diverse modalities, algorithms that enable the integration of such data are of high value. The authors apply MARIO to integrate CyTOF, CITEseq, and CODEX data sets in several different scenarios spanning healthy unperturbed cells, stimulated cells, and specimens from COVID patients, as well as across species and show several metrics to compare MARIO to previously published methods. However, more in-depth information is needed for the reader to properly assess whether MARIO truly outperforms previously published methods. Furthermore, benchmarking this method on ground-truth data sets would be highly beneficial to distinguish technical differences introduced by the different modalities from biological differences. Lastly, the examples of unknown biology being discovered with the help of MARIO could be presented in a more convincing way and it is unclear what concrete finding the authors refer to when stating that MARIO 'reveals orchestrated immune responses' in the title and abstract of their manuscript. Below are suggestions that would greatly improve the value of the manuscript.

Major remarks

- The authors only call high-level cell types in their data sets. In many instances the data sets already look sufficiently aligned pre-integration to call these very coarse cell types (e.g. Fig2A) and extract the biology exemplified (e.g. Figure 2 and 3). A finer level of clustering would be needed to assess the

performance of the MARIO integration and yield benefit from transferring information across data sets. Similarly, the tSNEs comparing MARIO to other methods are not very informative without knowing the sub-cluster annotation of the cells. Of note, being able to infer highly granular transcriptional information from a limited protein panel would be conceptually exciting since it would suggest that subtle changes in protein expression might capture much more information than appreciated. However, this cannot be concluded from the coarse cell types shown.

We thank the reviewer for raising this point, and benchmarking the matching accuracy with a higher granularity cell type annotation can further showcase the performance of MARIO. We performed a more detailed benchmarking analysis presented in the paper with a higher granularity level of cell types on the following datasets that further sub clustering is possible (Level 1: previous annotation, 6-7 cell types; Level 2: new annotation, 9-12 cell types):

Legend: **A**, part of Figure S2, cell type matching accuracy (level1 and 2) on bone marrow CITE-seq / CYTOF matching. **B**, part of Figure S8, cell type matching accuracy (level1 and 2) on cross species CYTOF matching (H1N1 and *Irf-gamma*). **C**, part of Figure S11, cell type matching accuracy (level1 and 2) on cross species CYTOF matching (H1N1 and *IL4*).

Additionally for these datasets, we updated the t-SNE plots of MARIO and other methods compared, with sub-cluster annotation information:

Legend: **A**, part of Figure S2, t-SNE plots on bone marrow CITE-seq / CYTOF integration. **B**, part of Figure S8, t-SNE plots on cross species CYTOF integration (H1N1 and *lfn-gamma*). **C**, part of Figure S11, t-SNE plots on cross species CYTOF integration (H1N1 and IL4).

We have also performed additional benchmarking on simulated **high granularity** data, a point that we will further elaborate below in later comments related to “real ground-truth dataset”.

- The test to drop features and assess the performance of MARIO compared to previous methods using multiple metrics is great. However, the same metric seems to perform quite variable across data sets, e.g. ARI F1 score shows major improvements in some, but not other data sets. It is therefore unclear

how reliable this benchmarking approach is. Are the results more consistent if features are dropped in a systematic way assessing all possible combinations instead of the alphabetic one-by-one drop?

We thank the reviewer for raising a very important point. Indeed, these metrics (Silhouette, Structure alignment, ARI, Mixing) are not perfect and may be of limited use when benchmarking cross-modality integration tasks compared to single-cell sequencing batch corrections methods. Intuitively, the reason why those metrics are so variable is because of their limited *differentiation power* when applied to cross-modality integration tasks.

For example, many of those metrics benchmark how well the two modalities are “well mixed” after integration, in the sense that those metrics will look good when certain statistical tests cannot distinguish which modality a dataset comes from. However, such statistical tests can be easily deceived: i.e. when the number of features is much larger than the number of samples, running CCA on two completely unrelated datasets will give perfect mixing due to overparameterization, an example of *over-integration* (a more extreme case is to simply attach to each cell a random vector from the same distribution to achieve perfect mixing). Within our cell matching frameworks, benchmarking those metrics serves as a “sanity check”: a method is clearly bad if it fails to achieve a reasonable score under any of these measures. However, a method is not necessarily good even if it is nearly perfect under all of these measures. If a method fails miserably on those metrics, we would not include that in our benchmarking pipeline. Within our cell matching frameworks, the key task is to find for each cell in X a matched cell in Y, such that they share similar biological states. Consequently, matching accuracy, matching proportions, and error avoidance scores are more robust and interpretable metrics for the scientific questions investigated in the current manuscript.

We fully agree with the reviewer’s point that assessing the sequence of feature dropping is important. While the reviewer had originally suggested a combinatorial drop test, that would have been a total of 8! Combinations (for 40320 plots per score), which we were regrettably unable to perform and present in an informative manner.

Therefore, on top of the alphabetical dropping we presented, we benchmarked dropping features in the sequence of their importance level (dropping less important ones first). This setup is closer to a real-world scenario, since when designing antibodies panels, the more “important” antibodies will be retained while the less biologically vital ones will be gradually dropped by their ranked importance, to be replaced with other markers based on their relevance to the study. We designed and implemented a random forest model with permutation feature importance test to evaluate the importance score of each antibody (detail can be found in Material & Methods section lines 1478 - 1490), and reperformed the ranked drop test now presented in the manuscript:

Legend: part of Figure S3. Benchmarking performed as previously but instead feature dropping in sequence of importance score, benchmarking on bone marrow CITE-seq / CYTOF matching.

Legend: part of Figure S9. Benchmarking performed as previously but instead feature dropping in sequence of importance score, benchmarking on cross species (H1N1-IJng) CYTOF matching.

Legend: part of Figure S14. Benchmarking performed as previously but instead feature dropping in sequence of importance score, benchmarking on murine spleen CODEX / CITE-seq matching.

- While it is nice to see MARIO applied to various settings, it would be highly beneficial for 1-2 data sets to provide more in-depth, and step-by-step information about what MARIO is doing rather than just the high-level tSNEs and metrics shown in the paper. E.g. what components are extracted from the data set using singular value decomposition? Given that the number of joint features is small, why is it beneficial to pick components and not use all components? How does the performance of MARIO change when changing such parameters? How does the integration actually look like when dropping features (instead of just showing the metrics)?

We fully agree with the reviewer's suggestion, and this effort will be vital for future users of MARIO. The parameters used in MARIO for each dataset have been detailed in the Material & Methods section. In addition, we encourage the reviewer to inspect our online tutorial, which walks through the reader on how to choose such parameters to achieve best performance (on our Github repository: eg. <https://github.com/shuxiaoc/mario-py/blob/main/tutorials/mario-py-tutorial-BM.ipynb>). The readers will also be able to recreate our analysis and figures for this paper using the python notebooks and code provided on our Github repository for reproducibility.

To further showcase the robustness of the choice of parameters used in MARIO, we now benchmark comprehensively MARIO parameters used (`n_components_ovlp`; `n_components_all`; `n_cancor`; `n_components_filter`; `bad_prop`) on different datasets. In summary, although choosing different

parameters (eg. number of components for initial matching) could affect the outcome of the matching to some extent, MARIO generally is highly robust with respect to different tuning parameter choices:

Legend: part of Figures S17. Benchmarking parameter used in MARIO, on bone marrow CITE-seq / CYTOF matching. Y axis, final matching accuracy by cell type annotation from MARIO (left) or percent of cells matched by MARIO (right); X axis number for the specific parameter used during MARIO run (from up to down: number of components

used during initial matching; number of components used during refined match; top numbers of k used to decide interpolation; number of clusters used during filtering; proportion of bad pairs assumed during filtering.)

Legend: part of Figures S17. Benchmarking parameter used in MARIO, on bone marrow cross species CYTOF matching. Y axis, final matching accuracy by cell type annotation from MARIO (left) or percent of cells matched by MARIO (right); X axis number for the specific parameter used during MARIO run (from up to down: number of

components used during initial matching; number of components used during refined match; top numbers of k used to decide interpolation; number of clusters used during filtering; proportion of bad pairs assumed during filtering.)

In terms of reviewer's question on how integration actually look like when dropping features, we now present a new set supplementary figures, recording the t-SNE plots for all the methods during each feature drop for the bmc and cross-species data:

Legend: part of Figures S2. t-SNE plots visualizing post-integration results with different methods, during each sequential protein feature drop step (alphabetical), for bone marrow CITE-seq / CYTOF matching.

Legend: part of Figures S3. t-SNE plots visualizing post-integration results with different methods, during each sequential protein feature drop step (importance), for bone marrow CITE-seq / CYTOF matching.

Legend: part of Figures S8. t-SNE plots visualizing post-integration results with different methods, during each sequential protein feature drop step (alphabetical), for cross species (H1N1-Iflng) CYTOF matching.

Legend: part of Figures S9. t-SNE plots visualizing post-integration results with different methods, during each sequential protein feature drop step (importance), for cross species (H1N1-Iflng) CYTOF matching.

- Similarly, it would be important to evaluate the method on a real ground-truth data set where the same samples is triaged for measurements of different modalities (e.g. CYTOF and CITEseq. The data set should be designed to have some clearly distinct cell states within the same cell type, e.g. by stimulating myeloid cells with different cytokines, and perfectly matching highly granular annotations. In this ground-truth data set, how does MARIO perform if specific cell states are dropped from one modality, are differences in highly granular cell states preserved or does MARIO over-integrate the data sets? How does MARIO perform when features are dropped, noise is added etc.? Of note, figure S7 showing IL4 versus influenza stimulated blood cells suggests that MARIO might over-integrate data sets since clear differences between IL4 and influenza treated samples would be expected. Furthermore, pSTAT1 would be expected to be downstream of IFN (triggered by influenza), not downstream IL4, which is opposite of what is shown here.

We thank the reviewer for this suggestion. Unfortunately, we were unable to obtain high-quality matched datasets suitable for these comparisons, and the generation of these data is beyond the scope of this manuscript. To fully address the reviewer's question, we leveraged upon simulated datasets where such requirements can be achieved quickly and reproducibly: We utilized Symim (Zhang et al., 2019), a single cell data simulation algorithm routinely used in benchmarking tasks (Cao et al., 2021), with parameters used previously to simulated proteomic data types (Kim et al., 2020). A total of 20 populations were simulated for high granularity, containing shared and distinct features mimicking real world antibody panel setups (20 shared, 20 non-shared). In summary, in this synthetic dataset, MARIO consistently outperformed all other benchmarked methods:

Simulated using Symsim (Zhang et al. Nat Comm 2019)

Legend: part of Figures S1. Symsim simulation setup.

Legend: part of Figures S1. Matching accuracy of MARIO and other methods on simulated high granularity ground truth data.

Legend: part of Figures S1. Difference in matching percentage of each sub-cluster for other methods compared to MARIO. MARIO generally performed better for all 20 cell types based on the diagonal color. Black indicates better performance for MARIO.

We thank the reviewer for very astute questions on the differences in pSTAT1 response within monocytic populations between the H1N1, IFN gamma and IL-4 treatments. We inspected the original data, and consistent with the reviewer’s point, observed much higher pSTAT1 response in Monocytes in IFN gamma compared to IL-4 treatment (both datasets from Bjornson-Hooper et al. 2022). During our re-inspection of the H1N1 treated dataset (from Rahil et al. 2019), we indeed observed that pSTAT1 staining was systematically lower in all batches from this dataset (spanning different donors and date), while other markers stained did not reflect this (Figure below).

We believe this is what caused the relative “dull” difference between influenza A virus challenge and IL-4 in terms of pSTAT1 response when both compared to that dataset:

Legend: figure for comment response. Marker levels across different datasets of CYTOF-cross species.

We are thankful for the reviewer in catching this oversight related to data quality. We have now removed the part of this study related to pSTAT1 and the H1N1 dataset.

In the updated figures, we now show the differential expression with the pSTAT3 expression. As shown in the previous figure, the staining quality of pSTAT3 was generally consistent between datasets. Despite the observed overall higher expression of pSTAT3 in the influenza group, NK cells from patients with H1N1 infections had consistently lower pSTAT3 expression compared to the IFN-gamma-stimulated NK cells.

Legend: part of Figures 3. Expression of markers on MARIO integrated embedding (H1N1- IFNgamma cross-species CyTOF).

We appreciate the reviewer’s point on potential over-integration between H1N1 and IL-4, and over-integration is indeed a huge challenge for the entire field. In this specific case, while the cells between the two datasets were aligned well in the same embedding space, showing very similar patterns (as this is one of **goals of integration**), differences between the two datasets are still preserved. For example, IL-4 stimulates high expression of STAT6 across multiple cell types, and the disparity compared to H1N1 can be observed:

Legend: figure for comment response. pSTAT6 expression in H1N1 and IL-4 cross-species CyTOF datasets.

Ref:

Zhang, X., Xu, C. and Yosef, N., 2019. Simulating multiple faceted variability in single cell RNA sequencing. *Nature communications*, 10(1), pp.1-16.

Cao, Y., Yang, P. and Yang, J.Y.H., 2021. A benchmark study of simulation methods for single-cell RNA sequencing data. *Nature Communications*, 12(1), pp.1-12.

Kim, H.J., Lin, Y., Geddes, T.A., Yang, J.Y.H. and Yang, P., 2020. CiteFuse enables multi-modal analysis of CITE-seq data. *Bioinformatics*, 36(14), pp.4137-4143.

Bjornson-Hooper, Z.B., Fragiadakis, G.K., Spitzer, M.H., Chen, H., Madhiredy, D., Hu, K., Lundsten, K., McIlwain, D.R. and Nolan, G.P., 2022. A comprehensive atlas of immunological differences between humans, mice, and non-human primates. *Frontiers in immunology*, 13.

Rahil, Z., Leylek, R., Schürch, C.M., Chen, H., Bjornson-Hooper, Z., Christensen, S.R., Gherardini, P.F., Bhate, S.S., Spitzer, M.H., Fragiadakis, G.K. and Mukherjee, N., 2020. Landscape of coordinated immune responses to H1N1 challenge in humans. *The Journal of clinical investigation*, 130(11), pp.5800-5816.

Other remarks

- Error avoidance score (page 20): If a cell type from data set X does not exist in the data set Y, you don't want any of those cells in X matched to something in Y, so a should ideally be 0 and the score low, not high. Please clarify.

Thank you for the suggestion. For consistency amongst all benchmarking figures such that a higher score means "better", we had reversed calculations for percentage (amount of cells should not be matched / amount of cells erroneously matched) in the Error Avoidance Score, thus when the score is high, less cells have been matched erroneously. The detail is described in the Material and Methods section as follows: "Error avoidance score measures the performance of the quality control process and is specific to the benchmarking scenario 3 (intentionally dropping cell types). For each cell type dropped, the corresponding error avoidance score is defined as V_a/b , where a is the number of cells in X that are of that type and have survived the quality control process (i.e., a match involving that cell type has occurred), and b is the total number of cells of that type X. The higher value of this score indicates that erroneous matching towards deleted cells types has been avoided more."

- Figure 2: CD34 and CD38 are listed as shared features in the text and used in the dropping of features approach in Figure 2C, but show up as CyTOF-specific features in Figure 2F. Please clarify.

We thank the reviewer for identifying the error. CD34 and CD38 are indeed shared features, but were excluded from the heatmap due to an unfortunate oversight. We have now updated Figure 2.

Legend: part of Figures 2. Updated heatmap with previously missing CD38 and CD34.

- Figure S3A does not show B cells in the legend.

We thank the reviewer for identifying the error. We have now added the B cell legend (the figure is now updated as Figure S5).

Legend: part of Figures S5. Updated figure with B cell legend.

- The text (line 450, page 6) refers to Figure 3E and F which do not exist in Figure 3.

We thank the reviewer for identifying this error. We have now updated the text to refer to the correct figures:

“We also observed the upregulation of pSTAT3 in the NK cell population within human and NHP samples treated with IFN gamma compared to human subjects challenged with influenza, although overall pSTAT3 PBMC expression was higher in the influenza group (Figure 3C)”

- Figure S5: It is unclear to me how the integration helps here. pSTAT1 and Ki-67 levels could just be compared within a data set between stimulated and unstimulated cells to determine the impact of the stimulation in the various settings. I would also argue that the effect is still visible in Figure S5D pre-integration.

We thank the reviewer for raising this point, and we agree that such a comparison may not necessarily require integration, particularly for qualitative comparisons. That said, they serve as great positive-controls to show that important information (such as relative measures of markers between species) is retained during the matching and integration process, and further allow users to directly visually assess functional changes across cell types after MARIO integration.

- CiteSeq / CODEX integration (Figure 4, S9, S10): Since there are a lot more CODEX cells than CITEseq cells and therefore a pairwise matching would leave many CODEX cells without partner, how is the single cell information transferred to the spatial location? Is for each CODEX cell the closest matching CITEseq cell used? In this case, what happens to cell types/states that are only present in the CODEX data?

The reviewer raises an astute point. Indeed there are much more CODEX cells (48,332) than CITE-seq cells (7,601) in this case. To accommodate this data size difference, we ran MARIO in a total of 32 batches (this has been described in the Material & Methods section inserted below), thus in each batch roughly 1,500 CODEX cells were matched to 7,601 CITE-seq cells. This batched matching will ensure that the majority of CODEX cells will result in a pairwise match with its closest CITE-seq partner, and thus transferable RNA information. In the methods section:

“and CODEX cells subsequently matched against CITE-seq cells, with MARIO parameters: $n_components_ovlp = 20$, $n_components_all = 15$, $sparsity = 1000$, $bad_prop = 0.05$, $n_batch = 32$, $knn = 15$.”

With regards to the reviewer’s point on the cell types that may only be present in one modality, we full appreciate this concern, which has been a huge driving factor behind our motivation to devise the “Joint Regularized Filtering” step which, by design, cleans out the matching pairs that are not from the same cell type / cell state. However, if the specific cell types / cell states are vastly unbalanced between datasets, for example in this mouse-spleen case the macrophage frequency is roughly 20 times lower in CITE-seq (likely due to the harsh tissue dissociation process), it could lead to lower model performance, compared to other cell types. This can be observed in Figure S12, where the macrophage matching accuracy is 73%, while other cell types with high accuracy.

Legend: part of Figures S12. Showing Macrophage matching accuracy.

These types of issue can not be fully resolved by algorithm design due to experimental or sample variation, we have thus added this to MARIO’s limitation in the discussion section as follows:

“Second, the prerequisite of performing such matching across datasets is that these datasets should be from very similar, if not the same, biological sample and biological status. Therefore, if certain cell types or cell state is missing in one modality, the matching and integration performance could potentially be worsen”.

- The gating on the CODEX data in Figure S9D does not look super convincing. In order to show the power of integrating data sets, it might be better to instead use granular clustering information based on the scRNAseq data and locate those cells in the tissue using the integrated CODEX data.

We thank the reviewer for raising this question, and have performed the analysis as suggested. Using the CITE-seq transcriptome information, we now locate the matched CITE-seq cell and visualized them on the CODEX data, and colored the spatial location based on their original CITE-seq annotation from Gayoso et al., 2021 (which is based on scRNA-seq information). The result is shown as follow:

Legend: part of Figures S12. A pseudo-colored murine spleen section colored by MARIO label transferred annotation from the CITE-seq dataset annotation.

We also updated the text in line 594 - 600 with:

“To further evaluate the matching outcome of refined cell types by MARIO, we sought to investigate the spatial location of matched B cell subtypes from the CITE-seq dataset onto the spleen. Indeed, the B cell subtypes were originally annotated by transcriptome information from Gayaso et al located into corresponding spatial niches (Figure S12F).”

- Shouldn't the MARIO-integrated tSNEs in Figures 4A and S9C be the same?

We thank the reviewer for raising this question. The tSNE plots shown in Figure S9C were based on CODEX protein only, thus different from the MARIO-integrated tSNE (which is based on CCA on post-processed matched pairs). We have now revised the figure legend (Figure S12) stating that t-SNE is calculated from CODEX protein but not MARIO integration values, to prevent confusion for future readers. The updated figure and legend are as follows:

Legend updated as:

“t-SNE plots (calculated from CODEX protein alone) of MARIO integrated murine spleen CITE-seq and CODEX cells, overlaid with matched CODEX protein and CITE-seq RNA expression levels.”

- Why is Figure S10C not shown for B cells?

We thank the reviewer for raising this question. In this specific case (CITE-seq CODEX murine spleen), B cells account for more than 50% of the total cells for both datasets. Therefore, if B cells are completely deleted in one modality, we reasoned such a situation is too extreme (since cells will be > 50% different), and matching will be difficult to be rigorously performed and accuracy measured on datasets that are highly dissimilar. This is in line with our results and explanation above related to the lower macrophage matching accuracy in this dataset due to imbalanced macrophage numbers (see above). Therefore we did not test the removal of B cells here.

- Figure S10D: There is a funny round red cluster in the Seurat integration tSNE. Is there something special about these cells?

Great observation. Indeed, these cells originated from the ones that Seurat failed to successfully integrate and embed (Seurat produced “NA” for them). For the transparency of our analysis, we replaced the NAs with 0s during tSNE calculation, causing these cells to aggregate into funny shapes without any meaningful interpretation available. We have noted this further in the Material and Methods section, as well as in our online code on Github to reproduce the analysis. In the methods section, we updated:

“In some rare cases certain methods produced NAs in the integrated values for limited number of cells, which were replaced as 0 for downstream analysis.”

- COVID data set and C1Q myeloid cell finding: Based on the CODEX data alone, would you have been able to distinguish C1Q low versus high cells? If not, can you show data on this to convince the reader that the integration of the data sets is indeed beneficial here?

We thank the reviewer for this question. We respectfully would like to point out that given the discovery of C1Q macrophages here using MARIO in this study, it might seem “obvious” why no one has observed it before (and indeed, this link has not been shown to date to our knowledge and reading of the literature).

First, without the *a priori* inclusion of C1Q as a CODEX staining marker, we would not be able to observe any C1Q-related differences in the cells *in situ*.

That said, while it appears that C1Q high macrophages are enriched in some M2 markers (such as CD163), it is not necessarily representative that all M2 macrophages are C1Q high (conversely, CD68 is also higher in C1Q high, while CD36 is lower as the reviewer points out). This suggests that the canonical M1/M2 separation is insufficient to fully stratify between C1Q high and low macrophages, and only possible with the integration of both CODEX and CITE-seq data with MARIO.

To further support our findings, we visualized UMAPs of the macrophages from CODEX and their associated protein expression profile, along with the matched CITE-seq C1Q expression levels. Note: The dimension reduction was performed with only CODEX protein features. In summary, using traditional methods, one would find it difficult to differentiate between the C1Q expressing macrophage population using solely canonical macrophage markers, including the classically defined M1 and M2 markers:

CODEX Macrophage Protein Expression (From CODEX):

CODEX Macrophage Matched C1Q Expression:

Legend: Figures S15. Umap reduction of CODEX macrophages, overlaid with CODEX protein or matched CITE-seq RNA expression levels.

We also updated the text in line 653-654:

“and such stratification is challenging without using MARIO matching (Figure S15).”

- C1Q low versus high cells could represent monocytes versus more mature myeloid cells. Do the C1Q low cells express monocyte markers? Which markers in Figure 5B are indicative of M1? Only CD36 looks higher in C1Q low cells in the CODEX data, which is thought to be higher in M2 rather than M1.

We appreciate this very valuable feedback by the reviewer, and believe this question is addressed in part by the response above, wherein the C1Q separation of macrophages are beyond just a binary M1/M2 classification. Such a discussion and detailed stratification of monocytes and mature myeloid cells may be beyond the scope of this manuscript, but we have now include it in the discussion and point readers to other resources that may be more suitable (Galán et al. 2015, Nahrendorf et al. 2016):

“These findings suggest the complex functionalities carried out by sub populations of myeloid cells in the tissue are beyond the classical binarization of M1/M2 macrophages in COVID-19 pathology.”

References

Chávez-Galán, Leslie, et al. "Much more than M1 and M2 macrophages, there are also CD169+ and TCR+ macrophages." *Frontiers in immunology* 6 (2015): 263.

Nahrendorf, Matthias, and Filip K. Swirski. "Abandoning M1/M2 for a network model of macrophage function." *Circulation research* 119.3 (2016): 414-417.

- Looking at the axes of the plots, I would argue that the percentages in PANINI versus MARIO are actually quite different (5I and J), even though indeed nicely correlated.

We agree with the reviewer that the PANINI versus MARIO do not show exactly the same percentages in the tissue cores. This could be due to various reasons unrelated to the matching performance and quality of MARIO, but rather the modality used.

We would like to point out that the *C1Q* expression level is evaluated using two different modalities

1. MARIO matching using data from CITE-seq: single cell sequencing from BALF cells, matched to CODEX cells). We used *C1QA*, *C1QB* and *C1QC* levels to categorize C1Q high/low macrophages.
2. Validation using PANINI: In-situ Hybridization with enzymatic amplification in FFPE tissues for *C1QA* alone, without protease treatment to retain the CD68 and CD15 epitopes.
3. The actual tissue section for the CODEX data (used for MARIO) and the PANINI validation data were not adjacent, but rather many sections away. This can also cause changes to the cells and tissue architecture present as we cut deeper into the tissue block. Generally, tissue blocks are trimmed between section sessions, and thus the two tissue sections used here are likely several mm apart, explaining in part the variation.

As the reviewer is aware, the values retrieved using *in situ hybridization* and sequencing can result in quite different numbers, even when measuring the same target in the same cell (Olivia, et al. 2015).

The following figure is from the Arjun Raj Lab (Raj lab blog post, 2015): showcasing absolute count difference between sequencing and FISH measurements in cells, with a nice correlation as pointed out by the reviewer. The absolute count from sequencing will be affected by sample preservation, library preparation, sequencing depth and more (Williams et al. 2014).

In our experiment, we did not perform RNA spot counting due to 1) challenges in highly autofluorescent FFPE tissues, 2) nature of sections of tissues versus whole cells, 3) highly amplified signals from PANINI and 4) limited resolution of microscopy data (20X versus 60X-100X generally used for FISH spot counting) to adhere to CODEX imaging parameters. As such, there may be more potential variation introduced due to the tissue type. Instead, we directly used the intensity of the *C1Q* RNA signals as a correlation for the quantity of *C1Q* transcripts present. For further references to technical variabilities related to FISH, we point the reviewer to the following reference Young et al. 2020.

In summary, while the percentage difference from the two methods is expected, while the nicely correlated result showcased the robustness of MARIO matching.

References

RNA-seq vs. RNA FISH for 26 genes, RajLab blog post 2015,
<http://rajlaboratory.blogspot.com/2015/06/rna-seq-vs-rna-fish-for-26-genes.html>

Padovan-Merhar, Olivia, et al. "Single mammalian cells compensate for differences in cellular volume and DNA copy number through independent global transcriptional mechanisms." *Molecular cell* 58.2 (2015): 339-352.

Williams, Alexander G., et al. "RNA-seq data: challenges in and recommendations for experimental design and analysis." *Current protocols in human genetics* 83.1 (2014): 11-13.

Young, Alexander P., Daniel J. Jackson, and Russell C. Wyeth. "A technical review and guide to RNA fluorescence in situ hybridization." *PeerJ* 8 (2020): e8806.

- The correlation of C1Q myeloid cells to patient BMI feels a bit out of place.

We appreciate the reviewer's suggestion and have expanded the sentence to better integrate it into the manuscript:

"Motivated to identify clinical correlates associated with complement activation of macrophages in patients, we observed a positive correlation between the abundance of C1Q Low macrophages and patient body mass index (BMI; Figure S16A). Given that low serum C1Q levels have been reported in patients with severe COVID-19 (65), future studies should explore whether C1Q dysregulation can explain the positive association between obesity and risk for COVID-19-related hospitalization and death (66)."

- Figure S11D is missing the legend, but if blue is low and red is high, C1Q high cells are the ones with higher levels for CCL2, while MX1 does not seem to be different between the two populations of cells. This does not match the results stated in lines 649-652.

We sincerely thank the reviewer for identifying the inconsistency here. We have updated the Figure (now Figure S16), where the C1Q high / Low group was originally mislabeled in reverse. We have also removed MX1 since it is not different between the two populations. We now include the updated Figure and text as follows:

D Differentially Expressed ISGs

"Our results suggest that in C1Q Low macrophages several previously described genes (including SERPINB9, CKAP4, CCL2 and SPHK1) (73–76) that encode proteins reported to directly inhibit SARS-CoV-2 replication and entry are upregulated, but the failure to regulate and dampen this innate response paves the way towards unchecked host immune responses and collateral tissue damage, while C1Q High

macrophages have elevated complement cascade (eg. LGALS3BP (77)), and genes previously found correlating to mild rather than severe COVID symptoms (eg. SIGLEC1 (78))”

- S11F: If this figure shows pairwise distances, what is the difference between Myeloid => Mq and Mq => Myeloid? Lymphatic vessels are the only feature that seems to clearly be closer to C1Q low than C1Q high. Are there a lot more C1Q high than low myeloid cells in the tissue and is this part of the explanation regarding the shorter distances?

We thank the reviewer for this question and apologize for lack of explanation. This method was proposed in our previous paper, and is essentially a permutation test with directionality (Jiang et al 2022).

To better convey the idea, here is an illustration:

Example case of why Interaction score is directional:
ie. Green Cell Abundant, Red Cell Rare

As such, due to the directionality that we can observe and statistically test in these pairwise distances, Myeloid => Mq is not equivalent to Mq => Myeloid. This method also takes into account the frequency of cell types being tested, and thus will not be biased due to higher or lower frequencies of e.g. myeloid cells in the tissue.

We have now expanded our Materials and Methods to better reflect this:

“To establish a baseline distribution of the distances, cells were randomly assigned to existing XY positions, for 1000 permutations. The baseline distribution of the distance was then compared to the observed distances using a Wilcoxon test (two-sided). The log₂ fold enrichment of observed mean over expected mean for each interaction type was plotted for interactions with a p-value < 0.05. The test results also includes the interactions in both directions (eg. Myeloid => T and T => Myeloid).”

References

Jiang, Sizun, et al. "Combined protein and nucleic acid imaging reveals virus-dependent B cell and macrophage immunosuppression of tissue microenvironments." *Immunity* (2022).

Reviewer #2:

Remarks to the Author:

This paper proposes a matching algorithm for single-cell multiomics data. The new method takes into account both shared and distinct features, and contains a filtering step to avoid sub-optimal matching. While the paper is generally well written and easy to read, I have some major and minor concerns as listed below.

Major concerns

1. The proposed method MARIO is very similar to Seurat v4s WNN method (Hao et al., 2021). Firstly, the cell matching procedure seems to be the 1-NN algorithm with the “cross-data distance matrix” as distance metric. The joint regularized filtering is similar to anchor filtering. So my first comment is that, how does this algorithm perform compared to Seurat’s CCA (proposed in Seurat v3 (Stuart et al., 2019), and also available in package v4) or RPCA approach? RPCA can be found at (https://satijalab.org/seurat/articles/integration_large_datasets.html) In other words, the authors only compared it with mutual nearest neighbor (mNN) methods but don’t compare with CCA methods.

We thank the reviewer for suggesting the comparison between MARIO and Seurat series. The newest flavor of Seurat method is Seurat V4 WNN: If the reference data contains multi omics information (eg. from CITE-seq, with both RNA and Protein data from the same cell), both modality will be used to construct the embedding, and then the embedding is used in the mNN approach for finding anchors, etc. Since our focus in this paper is general proteomic datasets, not just multi omics ones, the process to construct embeddings from multi omics datasets was not used in the reported benchmarking results, while the matching process we benchmarked against is the same as that used in Seurat V4. Although the goals of MARIO and Seurat series are partially in line with each other, we respectfully disagree with the reviewer's judgment that the methods are similar.

From an information-theoretic perspective, MARIO utilizes both the *explicit* signals in the shared features and *hidden* signals in the non-shared features, whereas Seurat only uses the former. Such a distinction is one of the key reasons why MARIO outperforms Seurat (and other methods using NN-based methods for matching) on proteomics datasets according to our now expanded benchmarking results. The capability of MARIO utilizing non-shared features is the result of a series of algorithmic inventions, which we respectfully refer the reviewer to in the references below.

Regarding Reviewer #2's concern about similarities with 1-NN and anchor filtering:

- The cell matching procedure (empowered by linear assignment) is fundamentally different from the 1-NN algorithm, in that linear assignment seeks to find a global correspondence matrix such that the sum of distances in matched pairs is minimized, whereas 1-NN performs a local search for each cell, and hence is greedy in nature. When the dimension of the dataset is relatively high (as is the case for single-cell datasets), 1-NN can be highly unstable due to the curse of dimensionality, whereas linear assignment is provably stable. We have submitted a sister paper, which includes support of this statement with full mathematical rigor (Chen et al. 2022 (arXiv preprint)). This is also supported by other papers in machine learning literature (e.g., Collier and Dalalyan 2016).
- Although both the joint regularized filtering and anchor filtering act as quality control steps, the two algorithms rely on very different assumptions. Joint regularized filtering assumes a matched pair is suspicious if they do not belong to the same cell population. On the other hand, anchor filtering declares a matched pair is suspicious if they are far away in the original (and high-dimensional) feature space (Stuart et al 2019). Consequently, anchor filtering is vulnerable to the curse of dimensionality as the metric structure in the original feature space can be noisy and potentially misleading. In contrast, joint regularized filtering operates in the "cluster space" where the original features are denoised by a regularized k-means algorithm and are thus more robust. The robustness of joint regularized filtering can also be seen from the benchmarking results, which reveals that this method can retain more cells while achieving higher matching accuracy, at sametime avoiding erroneous matching when cell types are imbalanced.
- For more mathematical detail and robustness related to our proposed filtering steps, please refer to our newly published theoretic paper (Chen et al. 2022 (Annals of Statistics)).

We appreciate and agree with the reviewer's suggestion on comparing our method with different flavors of Seurat. To fully answer reviewer's concern on this, we update the benchmarking with three type of Seurat V4 (PCA, CCA and RPCA) on 6 different datasets in the manuscript: simulated ground truth, bone marrow CITE-seq/CyTOF, cross-species CyTOF (2 of them), PBMC CITE-seq/CyTOF, and murine spleen CODEX / CITE-seq. In summary, the CCA and RPCA version did not show significant difference compared

to the default PCA versions, and the PCA version generally performs better in all cases tested. Results are shown below:

Simulated using Symsim (Zhang et al. Nat Comm 2019)

Mario (Accu 81.54%)

Seurat (PCA) (Accu 57.31%)

Seurat (CCA) (Accu 52.26%)

Seurat (RPCA) (Accu 53.07%)

Scanorama (Accu 63.30%)

FastMNN (Accu 60.57%)

Seurat (PCA) minus MARIO

Seurat (CCA) minus MARIO

Seurat (RPCA) minus MARIO

Scanorama minus MARIO

FastMNN minus MARIO

Legend: Figures S1. Matching accuracy MARIO compared to other methods, on simulated high granularity ground truth data. Simulation performed with Symsim, with parameters used to mimic epitome data as previously described.

Legend: part of Figures S2. Benchmarking of MARIO performance on bone marrow CITE-seq / CYTOF dataset, including Seurat PCA/CCA/RPCA.

Legend: part of Figures S6. Benchmarking of MARIO performance on PBMC CITE-seq / CYTOF dataset, including Seurat PCA/CCA/RPCA.

Legend: part of Figures S8. Benchmarking of MARIO performance on cross species CYTOF dataset (H1N1-Ifng), including Seurat PCA/CCA/RPCA.

Legend: part of Figures S11. Benchmarking of MARIO performance on cross species CYTOF dataset (H1N1-IL4), including Seurat PCA/CCA/RPCA.

Legend: part of Figures S13. Benchmarking of MARIO performance on murine spleen CODEX/ CITE-seq dataset, including Seurat PCA/CCA/RPCA.

References

Chen, Shuxiao, Sizun Jiang, Zongming Ma, Garry P. Nolan, and Bokai Zhu. "One-Way Matching of Datasets with Low Rank Signals." arXiv preprint arXiv:2204.13858 (2022).

Collier, Olivier, and Arnak S. Dalalyan. "Minimax rates in permutation estimation for feature matching." The Journal of Machine Learning Research 17, no. 1 (2016): 162-192.

Stuart, Tim, Andrew Butler, Paul Hoffman, Christoph Hafemeister, Efthymia Papalexi, William M. Mauck III, Yuhan Hao, Marlon Stoeckius, Peter Smibert, and Rahul Satija. "Comprehensive integration of single-cell data." Cell 177, no. 7 (2019): 1888-1902.

Chen, Shuxiao, Sifan Liu, and Zongming Ma. "Global and individualized community detection in inhomogeneous multilayer networks." Annals of Statistics (2022+).

2. This is related to the first concern. The cross data distance matrix is generated by computing the Pearson correlation. This is much weaker compared to the distance matrix proposed in Hao et al. (2021) and can only capture linear correlation. There is a vast literature on independence measurements, such as HSIC, distance correlation, etc, that can capture both linear and non-linear associations. If the authors

find their approach is inferior to Seurat V4, I suggest changing the Pearson correlation to something more advanced.

We thank the reviewer for this discussion on the design of the distance matrix. Indeed, if the cross data distance is generated by Pearson correlation on the *raw data*, then only linear structures can be captured. However, MARIO calculates the Pearson correlation on the *pre-processed* and *denoised* data (depending on the data types, eg arcsine transformed for CYTOF, or natural-log transformed CITE-seq counts), and the non-linear relationships across modalities can be captured in the pre-processing steps. The reason for such a pipeline (potentially non-linear pre-processing followed by linear Pearson correlation calculation) is two-fold.

1. It is computationally efficient as pair-wise Pearson correlation calculations can be vectorized
2. It provides highly accurate results according to various metrics and biological validations.

That said, we appreciate the point the reviewer has raised and it is a valuable suggestion. While as shown in the manuscript the matching performance by MARIO already outperforms Seurat (V3/V4 having the same mNN matching process), incorporating non-linear distance measures could be a potentially impactful future research direction for these methods. Hence for this revision, we performed benchmarking of MARIO initial matching accuracy, with Pearson correlation, distance correlation and non-linear kernelized distances, including radial basis function kernel (Gaussian kernel), polynomial kernel, and Laplacian kernel (detail described in material & methods). In summary, in the 4 datasets we benchmarked, pairwise correlation based distance matrix generally performs the best except one dataset. Among other kernels, Laplacian kernel generally performs better, and thus we have now updated the MARIO package with a flag to use Pearson correlation or Laplacian kernel to construct the distance matrix, per user's choice.

Legend: Figures S19. The initial matching accuracy by MARIO using distance matrix constructed by different methods: pearson correlation, distance correlation; non-linear kernels: gaussian; laplacian; polynomial and sigmoid, on different datasets presented in the manuscript.

We have also updated the discussion section regarding this point, in line 827 - 835:

“Fifth, while the distance matrix constructed in MARIO (by Pearson correlation) is computationally efficient and generally produces better matching outcome compared to more complicated distance matrices (eg. distance correlation or non-linear kernels) (Figure S19), to better accommodate specific requirements from future users, we supplied the option to use non-linear kernels (Laplacian) instead of Pearson to construct the distance matrix, per user's choice.”

3. There are many results regarding the metric “matching accuracy”, but there is no clear description on how to compute it for benchmarking methods. Some methods like Seurat’s WNN may not have pairs for every cell. Furthermore, when different methods have different numbers of cell pairs or anchors, is it still reasonable to use “matching accuracy” as the metric? For example, if we filter out more “bad” anchors for Seurat’s WNN method, would the “matching accuracy” of Seurat get improved? The authors need to make it clear and justify the use of evaluation metric.

We fully agree with the reviewer’s concern on the “matching accuracy” metric, as different methods will indeed produce different numbers of matched pairs, based on their own filtering scheme. For the matching accuracy presented in the paper, they are only calculated from cells that had been matched by the corresponding methods after their internal matching/ filtering steps (amount of cells matched correctly / amount of cells matched). We have the exact same concern on the issue the reviewers raised, and that is why for all the “matching accuracy” metric in the manuscript, they are coupled with a “proportion of cell matched” metric, thus also taking the filtering strength into account.

From all scenarios we evaluated, MARIO was consistently able to achieve a better “matching accuracy”, along with more or comparable amount of cells being matched with “proportion matched” (Figure 2C, Figure S2A, S6A, S8A, S11A, S13A). Moreover, we followed the reviewer’s suggestion on whether filtering based on Seurat’s anchor score can increase its accuracy. Based on our benchmark, where we filtered the Seurat matching result with Seurat’s Anchor scores (filter out cells with scores in lower quantile eg. 0%, 10% ... 50%), we still show superior performance by MARIO:

Legend: Figure for response. MARIO compared to Seurat matching with different filtering strength based on Anchor Scores.

Maintaining the number of cells being matched is vital, since certainly we could increase the filtering strength in eg. Seurat to achieve better cell type level accuracy (in extreme case filtering all cells but leaving one to get 100% accuracy), but such action could severely disrupt downstream analysis (eg. transferring RNA counts onto spatial experimental data), as this would lead to information loss in cell-specific granularity, decrease in statistical efficacy, and eventually result in non-informative integration which will not be relevant for biological studies.

Thus, we respectfully stand by the two combined metrics we used in the manuscript (“matching accuracy” and “proportion matched”) as a strong indicator that MARIO outperforms other methods.

Minor concerns:

1. UMAP plot should also be shown since it is more accurate compared to t-SNE.

We thank the reviewer for this suggestion and have updated the manuscript accordingly.

Legend: (A) part of Figures S2, UMAP plots for integration of bone marrow dataset. (B) part of Figures S6, UMAP plots for integration of PBMC dataset. (C) part of Figures S8, UMAP plots for integration of cross-species dataset (H1N1-Ifng).

Legend: (D) part of Figures S11, UMAP plots for integration of cross species dataset (H1N1-IL4). (E) part of Figures S13, UMAP plots for integration of murine spleen dataset.

2. Figure 3: Typo in caption; numbers of color bar in 3C are illegible.

We thank the reviewer for identifying the error. We have corrected the typo and made the color bar larger.

Reviewer #3:

Remarks to the Author:

Zhu et al. proposed a new method, called MARIO, for integrating multi-modal single-cell proteomics data. The main idea is to utilize both shared features as well as non-shared features for matching cells. The authors demonstrated the efficacy of MARIO on several datasets and highlighted some interesting discoveries from integrating CODEX and CITE-seq from COVID-19 patients.

Overall, I found the paper interesting and the method potentially useful. The presentation in general is clear, although it can be improved. The methods are described in sufficient detail and software is deposited in github.

Regarding novelty & significant: Although none of the components of the proposed methods is novel, putting them together for proteomics single-cell data integration appears to be new. However, the

proposed method is primarily based on heuristics. Although this is not necessary negative, the method itself does not provide significant conceptual advances.

I have some major concerns listed as follows:

1. The authors claim MARIO is “globally optimized”. However, in the paper, MARIO consists of several separate, sequential steps: a) initial matching with shared features; b) refined matching based on CCA of cells with aligned X & Y features from initial matching; and c) interpolating between initial and refined matchings. I don’t see a principled objective function that MARIO tries to optimize. Instead, it consists of several heuristic steps. In step c), the method aims to find the best convex combination in terms of the mean of the top k sample canonical correlations. The authors need to provide a justification on several issues:

We apologize for the confusion for the term “globally optimized”. Although MARIO consists of several sequential steps, there is indeed a single well-defined objective function it tries to optimize.

Following upon our notation in the Materials & Methods section, let X and Y be the two data matrices with rows corresponding to cells and columns corresponding to features. Without loss of generality, we assume that X has at most as many rows as Y. In other words, there are more or as many cells in Y compared with X. Suppose there are n rows in X and m rows in Y, then $n \leq m$. MARIO is an algorithm aimed at solving the following optimization problem:

$$\begin{aligned} & \text{maximize } \text{Tr}(A^T X^T H \Pi Y B) \\ & \text{subject to } \Pi \in S(n, m), \\ & \quad A^T X^T X A = I, \text{ and } B^T Y^T Y B = I. \end{aligned}$$

Here $H = I_m - \frac{1}{m} \mathbf{1}_m \mathbf{1}_m^T$ is the centering matrix, and $S(n, m)$ is the collection of all binary n-by-m matrices such that there are (m-n) zero columns and each of the remaining m columns has one and only entry equal to one, and each row has one and only one entry equal to one. That is, MARIO aims at simultaneously finding the cell-cell correspondence matrix Π and two linear transformations A and B such that after projecting the data matrices X and Y to a common latent space using A and B, and selecting a subset of rows of YB and matching them to the rows of XA in a one-to-one fashion, the trace inner-product between XA and $\Pi Y B$ is maximized. By the definition of $S(n, m)$, the matrix Π selects n rows of YB and then finds a bijection between the selected rows of YB and rows of XA.

Suppose both A and B are of rank k . The objective function of the optimization problem is a combination of the top k CCA objective function and the (unbalanced) linear assignment problem objective function:

1. When Π is given, solving for optimal A and B is simultaneously solving for top k canonical correlation loading vectors for the pair $(X, \Pi Y)$;
2. When A and B are given, solving for Π is exactly solving a linear assignment problem.

The interpolation between initial and refined matchings is a data-adaptive quality control step that allows users to backtrack/shrink towards the initial matching when the refined matching is not reliable (which could happen, e.g., when the non-shared features are extremely noisy).

We also respectfully disagree with the reviewer's comment that the method in MARIO is heuristic and provides no significant conceptual advances. We would like to point out that the matching algorithm and the joint filtering step in MARIO are in fact theoretically novel, and have resulted in two separate sister papers in high-dimensional statistics and machine learning (Chen et al. 2022 and Chen et al. 2022). The first paper also proves with mathematical rigor that for a general class of models, linear assignment provides information-theoretically optimal matching results when accuracy is measured by mismatch proportions. This theoretical underpinning gives us confidence at using linear assignment both in initial matching and in matching with canonical correlation scores in the refinement stage.

References

- Chen, Shuxiao, Sizun Jiang, Zongming Ma, Garry P. Nolan, and Bokai Zhu. "One-Way Matching of Datasets with Low Rank Signals." arXiv preprint arXiv:2204.13858 (2022).
- Chen, Shuxiao, Sifan Liu, and Zongming Ma. "Global and individualized community detection in inhomogeneous multilayer networks." *Annals of Statistics* (2022+).

a) Is the mean of the top k sample canonical correlations viewed as your global objective? If so, why is a good metric for integrating single-cell data?

We thank the reviewer for raising this important question. Mathematically, the sum of the top k sample canonical correlations (after alignment by Π) is indeed the global optimal objective value if the CCA latent dimension is set to k (note that maximizing the mean of top k canonical correlations is mathematically equivalent to minimizing the sum of squares in the latent space). For fix k , the sum and the mean of the top k sample canonical correlations are equivalent measures as they differ only by a scaling factor of k .

We believe it is a good metric for integrating single-cell data because intuitively, if we align cells of similar biological states from two datasets and project them into a common latent space, those latent

embeddings should stay close together. In the interpolation step, MARIO uses this metric as the proxy of matching quality and allows users to adaptively backtrack/shrink towards the initial matching when the refined matching becomes unreliable (eg. when unshared features are too noisy to improve matching accuracy). We have updated the text in line 214-217: *“This allows users to data-adaptively backtrack/shrink towards the initial matching when the refined matching becomes unreliable (eg. when unshared features are too noisy to improve matching accuracy).”*

b). Why consider only convex combinations? Is the resolution derived from interpolation globally optimal?

We thank the reviewer for opening a discussion on the convex combinations. We consider convex combinations in MARIO since it is computationally efficient and provides interpretable meanings on the relative reliability between the initial and the refined matchings.

For example, if the best weight is 0.3 for initial matching and 0.7 for refined matching, then we know that refined matching indeed extracts more hidden information from the non-shared features. However, the choice of using convex combination is not related to the global objective function MARIO tries to optimize.

As is discussed in the previous paragraph, convexity-based interpolation is a quality control step to make the whole MARIO pipeline more robust to extreme cases, e.g., when the non-shared features are extremely noisy and thus useless. The text is updated as described in the previous comment.

c). What’s the rationale for setting $k=10$? Are the results sensitive to the choice of k ?

We thank the reviewer for raising this point, and indeed the robustness of MARIO under different k is important. Our rationale of using top K is as follows: First, top canonical correlations are generally less noisy than the bulk ones (Bao et al 2019). On the other hand, including more canonical correlations (if they are accurate) can explain more cross-data variations. Our empirical evaluations indicate that $k=10$ achieves a reasonable tradeoff between the two competing goals in proteomics datasets tested.

The reviewer’s concern about how sensitive the choice of k is indeed valid. In the revision, we included new supplementary figures on the selection of top k and the corresponding algorithm performance. In summary, results suggest MARIO is not sensitive to the choice of k :

Legend: part of Figures S17. (A) Matching accuracy and proportion of cells matched for MARIO using different K, on bone marrow dataset. (B) Matching accuracy and proportion of cells matched for MARIO using different K, on cross species dataset (H1N1-Iflng).

References

Zhigang Bao, Jiang Hu, Guangming Pan, Wang Zhou. Canonical correlation coefficients of high-dimensional Gaussian vectors: Finite rank case. *Annals of Statistics* 47(1): 612-640 (2019). DOI: [10.1214/18-AOS1704](https://doi.org/10.1214/18-AOS1704)

2. Methods on genomic data integration are not thoroughly reviewed. Methods based on mNN represent only one integration method, by no means the only one. In fact, existing work has demonstrated that mNN is suboptimal. It is commonly used primarily because of its simplicity. For a more systematic comparison, methods based on deep learning models, optimal transport, and others, should be reviewed and compared. Although these methods may not be proposed specifically for integrating proteomics data, with minor modifications, they can. Some of the references in this regard include:

Lin, Y., Wu, TY., Wan, S. et al. scJoint integrates atlas-scale single-cell RNA-seq and ATAC-seq data with transfer learning. *Nat Biotechnol* 40, 703–710 (2022). <https://doi.org/10.1038/s41587-021-01161-6>

Cang, Z., Nie, Q. Inferring spatial and signaling relationships between cells from single cell transcriptomic data. Nat Commun 11, 2084 (2020). <https://doi.org/10.1038/s41467-020-15968-5>

Gayoso, A., Steier, Z., Lopez, R. et al. Joint probabilistic modeling of single-cell multi-omic data with totalVI. Nat Methods 18, 272–282 (2021). <https://doi.org/10.1038/s41592-020-01050-x>

We thank the reviewer for raising an important point. Indeed, compared to the proteomic field, genomic data integration is relatively better explored and studied by the field with numerous new tools published even in the last year. We apologize as our manuscript was not intended to be a comprehensive survey on genomic data integration, since the focus was on proteomic data integration. We agree with the reviewer that mNN definitely is not the only method used in the field, albeit being the most popular one (eg. Stuart et al. 2019, Hie et al. 2019, Haghverdi et al. 2018). There are other state-of-the-art methods available and showed significant improvement of performance in their respective field (ie, deep learning methods such as scJoint for scATAC->scRNA with very large number of shared features; optimal transport methods such as SpaOTsc for resolving spatial information from scRNA-seq data/cell-cell interaction).

These methods are usually designed for very specific tasks and lack optimization towards proteomic type of data, with potentially substantial modification and work needed to achieve reasonable results. For example, when using scJoint on proteomics datasets, we observed significant underperformance compared to all methods we previously benchmarked, while > 30 pre-training parameter combinations were tested: in the BMC CITE-seq/CYTOF dataset ~ 25% accuracy was achieved, while other methods can generally achieve > 85% accuracy; in the Cross-species CyTOF dataset ~ 60% accuracy was achieved, while other methods can generally achieve > 80%.

This is likely due to the limited feature numbers available in proteomic datasets, with 10-20 available compared to >10k features intended to be used in scJoint (scRNA & scATAC). Moreover, we observed over-fitting in the proteomic case, for example in the cross-species dataset, all the cells matched to neutrophil cells in the other dataset, which is the most abundant cell type in both datasets, thus achieving relatively lower loss while not producing biologically relevant matching results. We have decided **not to include scJoint** in benchmarking comparison as we do not want to criticize it unfairly on an integration task that it was not designed for.

The other type of matching method (Optimal transport eg. SpaOTsc) that the reviewer pointed out is indeed a promising way for matching cells across modalities. We therefore also benchmarked SpaOTsc, where we switched the originally required spatial distance matrix to a feature distance matrix for the second modality, in order to adapt the method to our scenario. The produced mapping matrix (gamma) was used to find the best match (largest gamma value) of each cell in modality 1 to cells in modality 2. We benchmarked SpaOTsc in four different datasets presented in the manuscript. In summary, while Optimal

transport can outperform mNN based methods in some cases, MARIO still consistently outperforms in all scenarios tested.

Legend: Figures S20. Matching accuracy by MARIO and SpaOTsc (optimal transport) across different datasets presented in the manuscript.

We have updated the discussion section in regard of other methods for potential matching purposes, in line 837 - 844: “And lastly, linear assignment (MARIO) and mNN (eg. Seurat, Scanorama, fastMNN and more) are not the only methods capable of matching cells across modalities. While not specifically designed for such task, we also tested SpaOTsc that utilized Optimal Transport to achieve matching of cells (Figure S20). Such methods also have potential and are of interest for future researchers in this field.”

References

Stuart, Tim, et al. "Comprehensive integration of single-cell data." *Cell* 177.7 (2019): 1888-1902.

Hie, Brian, Bryan Bryson, and Bonnie Berger. "Efficient integration of heterogeneous single-cell transcriptomes using Scanorama." *Nature biotechnology* 37.6 (2019): 685-691.

Haghverdi, Laleh, et al. "Batch effects in single-cell RNA-sequencing data are corrected by matching mutual nearest neighbors." *Nature biotechnology* 36.5 (2018): 421-427.

3. Since MARIO consists of multiple steps with multiple hyperparameters to tune for each step, there is a question of how robust MARIO is? How sensitive is MARIO wrt these parameters. I suggest authors include experiments of running MARIO on integrating sc-RNA-seq and spatial transcriptomics data. Several methods have been proposed for this benchmark. It would be interesting to see how MARIO performs compared to the state-of-the-art methods. This also serves the purpose of testing the robustness of MARIO by gradually removing shared features.

We fully agree with the reviewer that robustness of the method is vital. In line with the reviewer's suggestions, we produced new set of supplementary figures, benchmarking comprehensively MARIO parameters used (`n_components_ovlp`; `n_components_all`; `n_cancor`; `n_components_filter`; `bad_prop`) on different datasets (The BMC CITE-seq/CYTOF dataset (Figure 2) and Cross species CYTOF dataset (Figure 3)).

Legend: part of Figures S17. Benchmarking parameter used in MARIO, on bone marrow CITE-seq / CyTOF matching. Y axis, final matching accuracy by cell type annotation from MARIO (left) or percent of cells matched by MARIO (right); X axis number for the specific parameter used during MARIO run (from up to down: number of components

used during initial matching; number of components used during refined match; top numbers of k used to decide interpolation; number of clusters used during filtering; proportion of bad pairs assumed during filtering.)

Legend: part of Figures S17. Benchmarking parameter used in MARIO, on bone marrow cross species CYTOF matching. Y axis, final matching accuracy by cell type annotation from MARIO (left) or percent of cells matched by MARIO (right); X axis number for the specific parameter used during MARIO run (from up to down: number of

components used during initial matching; number of components used during refined match; top numbers of k used to decide interpolation; number of clusters used during filtering; proportion of bad pairs assumed during filtering.)

We appreciate the reviewer’s suggestion related to testing our method on genomic type datasets (scRNA-seq data and single cell spatial transcriptomics data). Although this type of data is not the primary focus of this manuscript, we included here new benchmarking analysis for MARIO using data from scRNA-seq (Mouse prefrontal cortex–SMART-seq2) and STARMAP (mouse cortex brain Visual_160):

Mouse prefrontal cortex (visual_160) – STARmap (Wang et al.,2018a)

Mouse prefrontal cortex – SMART-seq2 (The Allen Institute for Brain Science)

*For FastMNN only matched 10 cells (~1%)

Legend: Figure for response. Benchmarking MARIO on STARmap and scRNA-seq data.

To further follow along reviewer’s comment on benchmarking dataset types, we continued to benchmark MARIO on simulated ground truth datasets: We utilized Symim (Zhang et al., 2019) , a single cell data simulation algorithm routinely used in benchmarking tasks (Cao et al., 2021), with parameters used previously to simulated epitome data types (Kim et al., 2020). A total of 20 populations were simulated as high granularity data, with shared and distinct features mimicking real world antibody panel setups (20 shared, 20 unshared each dataset). In summary, MARIO achieved outstanding accuracy in this groundtruth high granularity dataset, compared to other methods:

Legend: part of Figures S1. Symsim simulation setup.

Mario (Accu 81.54%)

Seurat (PCA) (Accu 57.31%)

FastMNN (Accu 60.57%)

Seurat (CCA) (Accu 52.26%)

Seurat (RPCA) (Accu 53.07%)

Scanorama (Accu 63.30%)

Legend: part of Figures S1. Matching accuracy of MARIO and other methods on simulated high granularity ground truth data.

Legend: part of Figures S1. Matching accuracy of each sub-cluster for other methods compared to MARIO.

4. I also have serious concern regarding the assumption of injective mapping used in MARIO. It is clear that the cross-modal mapping is neither injective nor surjective due to the different ways how the samples are assayed. In addition, the distribution of cells across modality may be different. In this regard, the optimal transport approach is much better in handling such issues. The authors should discuss these limitations and show how it may impact the results across different samples.

We thank the reviewer for raising this excellent point. Indeed, if MARIO is applied to the raw data matrix X and Y without any pre-processing or post-processing, then the performance would clearly be sub-optimal due to the injectivity constraint. However, practically implementing MARIO involves two key steps to accommodate potential heterogeneity across modalities (e.g., the true matching is neither injective nor surjective, the distribution of cell types has mismatches, etc.):

1. We apply the core MARIO matching algorithm to *batches* of data. In short, we cut X into small batches and Y into large batches, and we apply the algorithm to the pair (X_b, Y_b) where the number of cells in X_b is smaller than the number of cells in Y_b (usually 3x~5x smaller, detailed further in Material and Methods for each dataset). In this way, we ensure the mapping is approximately injective within each batch. Theoretically, MARIO can give accurate matching as long as each cell in X_b has at least one cell in Y_b with a similar biological state. For better clarification on this point, we will expand the description on this aspect in the manuscript.
2. Joint regularized filtering. In this filtering step, we fit an L_0 -regularized k-means model and claim a matched pair is spurious when their estimated cell types do not match. Such a filtering procedure helps reduce the number of spurious matches resulting from cell type heterogeneity. In fact, we have provided benchmarking results when we manually make cell type compositions across modalities increasingly imbalanced, and we have confirmed that the joint regularized filtering can identify and discard spurious matches (see benchmarking related to “error avoidance scores”). For detailed mathematical proof of the joint regularized filtering step, we refer the reviewer to our newly published theory paper (Chen et al 2022). We have also updated the discussion section about this potential limitation when cell type is imbalance across modalities, in line 808-814: *“Second, the prerequisite of performing such matching across datasets is that these datasets should be from very similar, if not the same, biological sample and biological status. Therefore, if certain cell types or cell states are missing in one modality, the matching and integration performance could potentially be worsen.”*

References

Chen, Shuxiao, Sifan Liu, and Zongming Ma. "Global and individualized community detection in inhomogeneous multilayer networks." *Annals of Statistics* (2022+).

5. Another issue is the linearity assumption used throughout the paper. Even for shared features, simple Pearson correlation may not be adequate to capture the similarity of cells since data are assayed with different technologies. CCA is similarly impacted by this issue. How to address this issue?

We thank the reviewer for this discussion on the design of the distance matrix. Indeed, if the cross data distance is generated by Pearson correlation on the *raw data*, then only linear structures can be captured. However, MARIO calculates the Pearson correlation on the *pre-processed* and *denoised* data (depending on the data types, eg arcsine transformed for CyTOF, or natural-log transformed CITE-seq counts), and the non-linear relationships across modalities can be captured in the pre-processing steps. The reason for

such a pipeline (potentially non-linear pre-processing followed by linear Pearson correlation calculation) is two-fold.

1. It is computationally efficient as pair-wise Pearson correlation calculations can be vectorized
2. It provides highly accurate results according to various metrics and biological validations.

That said, we appreciate the point the reviewer has raised and it is a valuable suggestion. Incorporating non-linear distance measures could be a potentially impactful future research direction for these methods. Hence for this revision, we performed benchmarking of MARIO initial matching accuracy, with Pearson correlation, distance correlation, and non-linear kernelized distances, including radius basis function kernel (Gaussian kernel), polynomial kernel, and Laplacian kernel (detail described in material & methods). In summary, in the 4 datasets we benchmarked, pairwise correlation based distance matrix generally performs the best except one dataset. Among other kernels, Laplacian kernel generally performs better, and thus we will update the MARIO package with a flag to use Pearson correlation or Laplacian kernel to construct the distance matrix, per user's choice.

Legend: Figures S18. The initial matching accuracy by MARIO using distance matrix constructed by different methods: pearson correlation, distance correlation; non-linear kernels: gaussian; laplacian; polynomial and sigmoid, on different datasets presented in the manuscript.

We have also updated the text in line 829 - 837: *"Fifth, while the distance matrix constructed in MARIO (by Pearson correlation) is computationally efficient and generally produces better matching outcome compared to more complicated distance matrices (eg. distance correlation or non-linear kernels) (Figure S19), to better accommodate specific requirements from future users, we supplied the option to use non-linear kernels (Laplacian) instead of Pearson to construct the distance matrix, per user's choice."*

Minor comments:

1. Writing style: no need to repeat results of tables in the main text. It is fine to summarize key findings, but not necessary to restate what's already presented in the tables, making the text harder to read. We thank the reviewer for this suggestion. We have made the reporting of results in the manuscript more concise.

2. The Methods section needs to be cleaned up. Focus on describing new components of your methods. No need to describe in detail some of the standard methods such as PCA, CCA, which can be found in standard textbooks. Also I don't think it's a good idea to mix software with methods. Software can be described separately in the user manual. The methods should be dedicated to explain the method itself. We thank the reviewer for this suggestion, and we have cleaned up the Material & Methods section and moved out the basic definitions, and denotation related to the software package.

3. "The signals within these limited shared features alone are typically not sufficient to produce high-quality and interpretable pairwise cell matching results." To justify this statement, it would be interesting to compare matching results based on shared features and non-shared features separately. We appreciate the reviewer for suggesting this interesting comparison. However, based on the nature of our method implemented in MARIO, we will not be able to perform such analysis, since we will need an initial matching step (thus using the shared features) performed before we can implement the refined matching step (using unshared features). Thus, shared features and unshared features matching cannot be separately analyzed.

4. The results often show comparable results when compared to Seurat, which also adopts CCA for matching. The author should discuss how CCA is applied differently between Seurat and MARIO. In the introduction, the authors state that "Unfortunately, application of this approach to single cell...magnitudes smaller than those in single-cell sequencing datasets." Can authors elaborate on this and perhaps show some results to support the statement?

We apologize for the confusion and lack of explanation. In the revised manuscript, we will include more details on the distinction between MARIO CCA and Seurat CCA. From a high level, Seurat applies a

simplified version of CCA (which was referred as “diagonal CCA” in their manuscript) to the **rows of the datasets** (i.e., *the dimension corresponding to cells*), whereas MARIO applies classical CCA (based on partial least squares per Scikit-Learn’s implementation) to the **columns of the datasets** (i.g., *the dimension corresponding to protein features*). Seurat CCA is used to get joint embeddings of the shared parts of the two datasets, whereas MARIO CCA is adopted to extract the hidden information in the non-shared parts of the two datasets. Although both methods used CCA, however, the places where CCA is used are fundamentally different.

For the statement “Unfortunately, application of this approach to single cell...magnitudes smaller than those in single-cell sequencing datasets” we made in the introduction section, supporting results can be seen by the lower matching accuracy generated by other methods compared to MARIO in all 6 datasets (Figure S1, Figure 2C and S2A, Figure S6A, Figure S8A, Figure S11A, Figure S13A), and such underperformance was further exacerbated by antibody dropping tests during benchmarking. For example, when 8 antibodies were dropped from the original dataset in Figure 2C and Figure S2A, the matching accuracy produced by other methods was only ~70% , which is far from optimal and relevant to meaningful biology, why MARIO is > 90% accurate.

In single-cell sequencing datasets, the information to separate fine-graded cell subpopulations is sufficient, meanwhile there will be numerous shared features (typically ranging from 1k-10k). This makes cell matching a relatively easy task for two reasons:

1. When restricted to shared features, the cross-modality distance is straightforward to compute
2. When the separation among cell types is sufficiently clear, local searches like nearest neighbors can give reliable matching results.

However, when integrating proteomics datasets, the number of shared features is very limited (typically 10-30). Thus, we require both a more accurate matching algorithm (linear assignment rather than nearest-neighbor), and also to further augment the signal for matching by exploiting the correlation between shared and unshared features.

5. As shown in FigS1, compared with other MNN based methods, the overall scores do not consistently surpass other competitors except ARI F1 score. The same phenomenon is observed in FigS4A, the authors need to give some explanation on the inconsistent results.

The pipeline drawn in the Figure 1B is a little bit vague to understand, authors need to give more technical details on the matching and testing procedures.

We thank the reviewer for raising a very important point. Indeed, these metrics (Silhouette, Structure alignment, ARI, Mixing) are not perfect and may be of limited use when benchmarking cross-modality

integration tasks compared to single-cell sequencing batch corrections methods. Intuitively, the reason why those metrics are so variable is because of their limited *differentiation power* when applied to cross-modality integration tasks.

For example, many of those metrics benchmark how well the two modalities are “well mixed” after integration, in the sense that those metrics will look good when certain statistical tests cannot distinguish which modality a dataset comes from. However, such statistical tests can be easily deceived: i.e. when the number of features is much larger than the number of samples, running CCA on two completely unrelated datasets will give perfect mixing due to overparameterization, an example of *over-integration* (a more extreme case is to simply attach to each cell a random vector from the same distribution to achieve perfect mixing). Within our cell matching frameworks, benchmarking those metrics serves as a “sanity check”: a method is clearly bad if it fails to achieve a reasonable score under any of these measures. However, a method is not necessarily good even if it is nearly perfect under all of these measures. If a method fails miserably on those metrics, we would not include that in our benchmarking pipeline. Within our cell matching frameworks, the key task is to find for each cell in X a matched cell in Y, such that they share similar biological states. Consequently, matching accuracy, matching proportions, and error avoidance scores are more robust and interpretable metrics for the scientific questions investigated in the current manuscript.

We thank the reviewer for the point on Fig 1B, which was originally intended for a brief guide for readers. In discussions with the handling editor, we will expand upon this figure with the Springer Nature art team should this manuscript be accepted.

6. The matching accuracy is evaluated at the cell type level, which depends on the quality of data annotation. Authors should discuss this limitation and discuss how to evaluate the methods on higher resolutions.

We thank the reviewer for raising this point, and we agree that although MARIO has already outperformed other matching methods in the current cell type annotation level, benchmarking the matching accuracy with higher granularity cell type annotation could further showcase the true-outperforming capabilities of MARIO. Therefore, we performed a more detailed benchmarking analysis presented in the paper with a higher granularity level of cell types, with the following datasets:

Legend: A, part of Figure S2, cell type matching accuracy (level1 and 2) on bone marrow CITE-seq / CYTOF matching. B, part of Figure S8, cell type matching accuracy (level1 and 2) on cross species CYTOF matching (H1N1 and Ifn-

gamma). **C**, part of Figure S11, cell type matching accuracy (level1 and 2) on cross species CYTOF matching (H1N1 and IL4).

Moreover, we have benchmarked the MARIO matching performance on the high granularity ground-truth simulated dataset (by Symsim, with 20 cell types) as shown in previous responses, where we showcased superior accuracy of MARIO compared to other methods. Figures and text related to benchmarking higher resolution were updated accordingly.

Decision Letter, first revision:

Dear Sizun,

Thank you for submitting your response to our queries on your revised manuscript "Robust Single-cell Matching and Multi-modal Analysis Using Shared and Distinct Features Reveals Orchestrated Immune Responses" (NMETH-A47859C).

I am delighted to inform you that we'll be happy in principle to publish it in Nature Methods, pending minor revisions to satisfy the referees' final requests and to comply with our editorial and formatting guidelines. We are now performing detailed checks on your paper and will send you a checklist detailing our editorial and formatting requirements in about a week. Please do not upload the final materials and make any revisions until you receive this additional information from us.

Thank you again for your interest in Nature Methods Please do not hesitate to contact me if you have any questions.

Sincerely,
Madhura

Madhura Mukhopadhyay, PhD
Senior Editor
Nature Methods

TRANSPARENT PEER REVIEW

Nature Methods offers a transparent peer review option for new original research manuscripts submitted from 17th February 2021. We encourage increased transparency in peer review by publishing the reviewer comments, author rebuttal letters and editorial decision letters if the authors agree. Such peer review material is made available as a supplementary peer review file. Please state in the cover letter 'I wish to participate in transparent peer review' if you want to opt in, or 'I do not wish to participate in transparent peer review' if you don't. Failure to state your preference will result in delays in accepting your manuscript for publication.

ORCID

Reviewer #1 (Remarks to the Author):

In their revised manuscript and thorough rebuttal letter, the authors provide considerably more information to help the reader evaluate the performance of MARIO against other methods and include a simulated data set to further benchmark the different methods. The authors also (1) benchmarked MARIO with more granular and therefore biologically more useful cell type annotations, (2) corrected

errors that previously made some of the biology confusing, such as the former mislabeling of C1Q high versus low in what is now Figure S16, (3) tracked down an issue with low pSTAT1 staining quality in one of the data sets that accounted for an erroneous conclusion, and (4) more convincingly showed how MARIO can assist in transferring CITE-seq transcriptome annotations to CODEX data (Figure S12). Technical issues such as the variable staining quality in different tissues / data sets are common traps when performing integrative analyses, which might be worth to point out in the discussion of the limitations. The C1Q macrophage findings and whether the integration of the lung autopsy and BALF data is biologically reasonable remain difficult to evaluate in the absence of more information such as the 50 CODEX and 250 CITE-seq markers. In sum, it is still a bit unclear what the 'orchestrated immune responses' are that are promised in the title, but the presentation of MARIO as a method and the examples what to use it for are much improved.

Minor point: There are still some minor errors that should be fixed such as in the legend of Figure 3 where it says 'pSTAT1' instead of 'pSTAT3'.

Reviewer #2 (Remarks to the Author):

The authors were very responsive and they have successfully addressed my comments. As a reviewer I posed several questions based on my understanding of MARIO. I have one minor comment remaining. The authors successfully responded to my questions, and in some cases the point was clear in the original text. However, in other cases I don't believe the text was clear and the authors didn't revise the manuscript to ensure others wouldn't misunderstand. The Methods are extremely clear. On the other hand, while the text can't be technical, Figure 1 and the associated text is confusing. What is being presented in Fig 1a, right quadrant?

Reviewer #3 (Remarks to the Author):

I thank the authors for taking efforts to address my previous comments. I think the revised paper is improved with more clarifications on technical details as well as underlying assumptions. Although I am generally pleased with the responses, I have the following additional comments:

1. I believe the method is of value to the proteomics community. But my previous comment regarding the novelty and significance of the methods remains. It is a useful method, showing some improvements compared to the previous methods on the datasets the authors benchmarked. But the method fundamentally relies on linear assumptions, based on classical CCA. In the abstract, the authors wrote "MARIO thus provides an analytical framework for unified analysis of single-cell data for a

comprehensive understanding of the underlying biological system.” I would strongly disagree with this characterization. Firstly, it is only developed for and only demonstrated for proteomics. Secondly, several advanced integration methods are proposed recently (for other data modalities) and are not compared with. I would suggest authors tone down their language and present the work as practically useful for proteomics data integration, instead of making unsupported grand statements.

2. To address the linear assumption limitation, the authors argue for the value of preprocessing and denoising. It may address it partially, but it doesn't solve the problem since how the preprocessing is done depends on your assumptions.

3. In the updated analysis, the authors present results on different choices of parameter settings. However, I'm puzzled that most of the time the results do not change et al – the lines are mostly flat in Figure S17. If the results do not change over a small range of parameters, that would argue for the robustness of the algorithm. But if the results are flat over a large range, what does this mean?

4. I appreciate the author for clarifying the objective function of MARIO. The authors include discuss in the main manuscript, and also discuss the pros and cons of using the mean of the top k sample canonical correlation, as well as its biological relevance.

5. It's clear that the multi-step algorithm presented in the paper doesn't guarantee to find the globally optimal solution for the objective function. The authors should remove such a statement.

6. In the abstract, “Current available tools ... generally rely upon large number of shared features across datasets for mutual Nearest Neighbor (mNN) matching”. This is a narrow view of the current state of the arts. Should change it.

7. For my previous minor points 5 & 6:

The authors mentioned in their paper that they have benchmarked with LIGER in many of their analyses, however, we think the authors need to specify if they are using LIGER's latest implementation of UINMF1, which is designed to perform a similar task as MARIO.

Kriebel, A. R. & Welch, J. D. UINMF performs mosaic integration of single-cell multi-omic dataset using nonnegative matrix factorization. Nat Commun 13, 780 (2022).

Final Decision Letter:

Dear Sizun,

I am pleased to inform you that your Article, "Robust Single-cell Matching and Multi-modal Analysis Using Shared and Distinct Features", has now been accepted for publication in Nature Methods. Your paper is tentatively scheduled for publication in our January print issue, and will be published online

prior to that. The received and accepted dates will be 25th Mar 2022 and 31 Oct 2022. This note is intended to let you know what to expect from us over the next month or so, and to let you know where to address any further questions.

Over the next few weeks, your paper will be copyedited to ensure that it conforms to Nature Methods style. Once your paper is typeset, you will receive an email with a link to choose the appropriate publishing options for your paper and our Author Services team will be in touch regarding any additional information that may be required.

Your paper will now be copyedited to ensure that it conforms to Nature Methods style. Once proofs are generated, they will be sent to you electronically and you will be asked to send a corrected version within 24 hours. It is extremely important that you let us know now whether you will be difficult to contact over the next month. If this is the case, we ask that you send us the contact information (email, phone and fax) of someone who will be able to check the proofs and deal with any last-minute problems.

If, when you receive your proof, you cannot meet the deadline, please inform us at rjsproduction@springernature.com immediately.

Once your manuscript is typeset and you have completed the appropriate grant of rights, you will receive a link to your electronic proof via email with a request to make any corrections within 48 hours. If, when you receive your proof, you cannot meet this deadline, please inform us at rjsproduction@springernature.com immediately.

Once your paper has been scheduled for online publication, the Nature press office will be in touch to confirm the details.

Content is published online weekly on Mondays and Thursdays, and the embargo is set at 16:00 London time (GMT)/11:00 am US Eastern time (EST) on the day of publication. If you need to know the exact publication date or when the news embargo will be lifted, please contact our press office after you have

submitted your proof corrections. Now is the time to inform your Public Relations or Press Office about your paper, as they might be interested in promoting its publication. This will allow them time to prepare an accurate and satisfactory press release. Include your manuscript tracking number NMETH-A47859D and the name of the journal, which they will need when they contact our office.

About one week before your paper is published online, we shall be distributing a press release to news organizations worldwide, which may include details of your work. We are happy for your institution or funding agency to prepare its own press release, but it must mention the embargo date and Nature Methods. Our Press Office will contact you closer to the time of publication, but if you or your Press Office have any inquiries in the meantime, please contact press@nature.com.

If you are active on Twitter, please e-mail me your and your coauthors' Twitter handles so that we may tag you when the paper is published.

Please note that *Nature Methods* is a Transformative Journal (TJ). Authors may publish their research with us through the traditional subscription access route or make their paper immediately open access through payment of an article-processing charge (APC). Authors will not be required to make a final decision about access to their article until it has been accepted. [Find out more about Transformative Journals](https://www.springernature.com/gp/open-research/transformative-journals)

To assist our authors in disseminating their research to the broader community, our SharedIt initiative provides you with a unique shareable link that will allow anyone (with or without a subscription) to read the published article. Recipients of the link with a subscription will also be able to download and print

the PDF. As soon as your article is published, you will receive an automated email with your shareable link.

Please note that you and your coauthors may order reprints and single copies of the issue containing your article through Nature Portfolio 's reprint website, which is located at <http://www.nature.com/reprints/author-reprints.html>. If there are any questions about reprints please send an email to author-reprints@nature.com and someone will assist you.

Best regards,
Madhura